# A Probabilistic Framework for LLM-Based Model Discovery

**Stefan Wahl** [1 2]  **Raphaela Schenk** [3]  **Ali Farnoud** [3]  **Jakob H. Macke** [1 2 4]  **Daniel Gedon** [1 2]

## Abstract

Automated methods for discovering mechanistic simulator models from observational data offer a promising path toward accelerating scientific progress. Such methods often take the form of agentic-style iterative workflows that repeatedly propose and revise candidate models by imitating human discovery processes. However, existing LLM-based approaches typically implement such workflows via hand-crafted heuristic procedures, without an explicit probabilistic formulation. We recast model discovery as probabilistic inference, i.e., as sampling from an unknown distribution over mechanistic models capable of explaining the data. This perspective provides a unified way to reason about model proposal, refinement, and selection within a single inference framework. As a concrete instantiation of this view, we introduce ModelSMC, an algorithm based on Sequential Monte Carlo sampling. ModelSMC represents candidate models as particles which are iteratively proposed and refined by an LLM, and weighted using likelihood-based criteria. Experiments on real-world scientific systems illustrate that this formulation discovers models with interpretable mechanisms and improves posterior predictive checks. More broadly, this perspective provides a probabilistic lens for understanding and developing LLM-based approaches to model discovery.

## 1. Introduction

Scientific discoveries—from the expanding universe (Hubble, 1929) to gravitational waves (Abbott et al., 2016) and neuronal excitability (Hodgkin & Huxley, 1952)—often rely on formal models that integrate theory with data to enable interpretation and prediction. As scientific problems grow in experimental and conceptual complexity, computational simulators that encode explicit mechanistic hypotheses have become central tools for discovery (Frigg & Reiss, 2009; Winsberg, 2019; Lavin et al., 2022).

Simulators are only approximations of reality, and their fidelity depends on how accurately they capture the structure of the underlying system (Box & Draper, 1987). Constructing scientific simulators that are simultaneously interpretable, tractable, and realistic can be challenging. This modeling process requires substantial domain expertise and iterative manual refinement, rendering it labor-intensive and difficult to scale.

A range of methods have been proposed to partially automate model discovery, but existing approaches typically trade off expressive model classes against mechanistic interpretability or require substantial manual specification (Udrescu & Tegmark, 2020; Donnelly et al., 2024; Schröder & Macke, 2024). LLMs offer a qualitatively different point in this design space: trained on extensive corpora of scientific text and source code, they can generate and revise executable simulator implementations directly from textual problem formulations and feedback (Austin et al., 2021; Chen, 2021).

However, most LLM-based discovery systems are defined operationally, i.e., via prompts, agent roles, and interaction protocols, rather than as inference procedures for an explicit target distribution. As a result, these approaches (i) do not explicitly state the underlying inference problem, which hinders interpretability, obscures failure modes, and prevents principled extensions or connections to existing theory; (ii) do not naturally admit theoretical analysis or convergence properties; and (iii) rely on loosely specified agent roles and interactions, making it difficult to reason formally about the necessity or sufficiency of components.

Here, we formulate automated model discovery as a probabilistic inference problem. Given observational data $\boldsymbol{x}_o$, the goal is to sample from the unknown distribution $p(m|\boldsymbol{x}_o)$ over mechanistically interpretable models $m$—here, simulator programs implemented in code—capable of explaining the data $\boldsymbol{x}_o$. Building on this formulation, we introduce

---

[1]Machine Learning in Science, University of Tübingen, Tübingen, Germany [2]Tübingen AI Center, Tübingen, Germany [3]Boehringer Ingelheim, Biberach, Germany [4]Department Empirical Inference, Max Planck Institute for Intelligent Systems, Tübingen, Germany. Correspondence to: <{firstname.lastname}@uni-tuebingen.de>.

*Proceedings of the 43rd International Conference on Machine Learning*, Seoul, South Korea. PMLR 306, 2026. Copyright 2026 by the author(s).

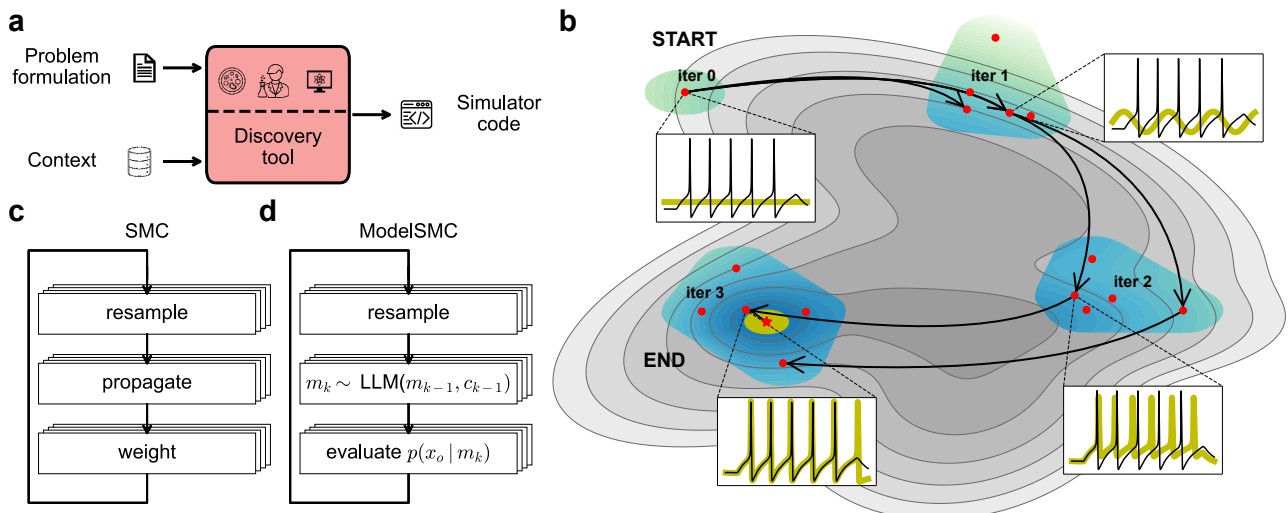

*Figure 1.* **Overview of ModelSMC for automated LLM-based model discovery. (a)** Given a textual problem formulation and context data, we infer a simulator model implemented in code. **(b)** ModelSMC iteratively refines an initial model to sample high-density regions of the model posterior $p(m|\boldsymbol{x}_o)$, approaching the unknown data-generating process (red star) in high-density regions. **(c)** ModelSMC is inspired by SMC, which approximates evolving distributions via weighted particles by iterative resampling, propagation, and weighting. **(d)** In ModelSMC, models are propagated by LLM sampling and weighted by likelihood evaluation.

ModelSMC, an approach based on Sequential Monte Carlo (SMC) methods, which approximate latent distributions using collections of weighted particles (Johansen, 2009). In ModelSMC, candidate simulator models are represented as particles whose code is iteratively proposed and modified by an LLM, evaluated using likelihood-based criteria, and resampled to focus on promising (i.e., high-marginal likelihood) regions of the model space (Fig. 1).

Formulating model discovery as probabilistic inference within an SMC framework closely mirrors the iterative workflows employed by human experts, yielding conceptual and practical advantages: First, it provides an explicit mathematical formulation of the refinement process, clarifying assumptions and failure modes. Second, under standard SMC conditions, it enables principled guarantees on stability and convergence. Third, it allows formal reasoning about the role and necessity of individual system components, facilitating systematic ablations and extensions. Our main contributions can be summarized as follows:

- We cast automated LLM-based discovery of models as probabilistic inference over executable programs.

- We propose ModelSMC, an SMC-style population method with LLM-based proposals of programs.

- We demonstrate competitive performance on synthetic tasks and show that ModelSMC discovers accurate and mechanistically interpretable models on real-world neuroscience and pharmacology datasets.

- We show how existing SMC theory can be used to analyze automated model discovery with LLMs.

## 2. Related Work

**Classic mechanistic model discovery.** Automated discovery of mechanistic models has traditionally focused on symbolic regression, equation discovery, and program synthesis, including genetic programming (Schmidt & Lipson, 2009; Udrescu & Tegmark, 2020; Kamienny et al., 2022) and sparse regression over predefined libraries of candidate terms (Brunton et al., 2016; Rudy et al., 2017), as well as grammar- or DSL-based program synthesis (Gulwani et al., 2017; Parisotto et al., 2017; Ellis et al., 2021). Other approaches combine mechanistic structure with statistical learning or physics-informed models (Raissi et al., 2019; Rackauckas et al., 2020; Karniadakis et al., 2021; Donnelly et al., 2024). These methods rely on carefully engineered search spaces and have so far focused on low-complexity models. In contrast, ModelSMC explores complex simulator programs spanning hundreds of lines of code in an open search space without hand-crafted restrictions.

**LLM-based methods.** More recently, several works have used LLMs for open-ended automated model discovery. One class of approaches combines LLMs with symbolic regression or optimization loops, where the LLM proposes candidate equations or programs and iterative evaluation or gradient-based refinement is used to improve fit to data (Ma et al., 2024; Li et al., 2024; Shojaee et al., 2025; Holt et al., 2025). Another line builds on evolutionary search, exemplified by FunSearch (Romera-Paredes et al., 2024; Castro et al., 2025) or the closed-source AlphaEvolve (Novikov et al., 2025). These frameworks have been applied to domain-specific problems in cognitive science (Rmus et al.,

2025; Jha et al., 2026), language modeling (Cheng et al., 2025), pharmaceutical applications (Holt et al., 2024), neuroscience (Tilbury et al., 2025), or combinatorial optimization (Liu et al., 2024). Model proposal, refinement, and selection are typically defined through heuristic system designs, such as fixed agent roles, optimization loops, or evolutionary operators. Selection in particular relies on general-purpose metrics such as MMD, MSE, or Wasserstein distance (Gretton et al., 2012; Arjovsky et al., 2017), rather than derived from an explicit mathematical objective.

Closest to our work are methods combining LLMs with Bayesian reasoning in scientific and probabilistic settings, such as Large Language Bayes (Domke, 2025) and AutoDiscovery (Agarwal et al., 2025). Large Language Bayes generates probabilistic programs from informal descriptions and performs approximate Bayesian inference via likelihood-weighted model averaging. AutoDiscovery explores nested hypotheses using Bayesian surprise within a Monte Carlo tree search, without explicitly defining a posterior over mechanistic models or maintaining a population of executable models. In contrast, we frame mechanistic model discovery as inference over models.

**Probabilistic model inference.** Probabilistic approaches to model discovery and comparison are traditionally grounded in Bayesian model selection and averaging, where candidate models are evaluated via marginal likelihoods or Bayes factors (Kass & Raftery, 1995; Morey et al., 2016), as well as Bayesian program induction (Lake et al., 2015). With simulators as tools for scientific inquiry, this has motivated simulation-based and likelihood-free inference methods, including synthetic likelihoods and neural density estimation (Cranmer et al., 2020; Deistler et al., 2025). It also includes SMC samplers for static or trans-dimensional targets, where explicit likelihood tempering bridges from prior to posterior to mitigate weight degeneracy (Del Moral et al., 2006; Neal, 2001). Recent work, such as simulation-based model inference (Schröder & Macke, 2024), extends this view to simultaneous model and parameter inference over graph-structured components using amortized inference, but assumes a fixed compositional prior without iterative refinement. In contrast, ModelSMC treats full model programs as latent variables and uses LLMs as proposals for open-ended model spaces, progressively concentrating on high-likelihood regions without an explicit tempering schedule.

# 3. Method

## 3.1. Background: Sequential Monte Carlo

SMC methods (Johansen, 2009; Naesseth et al., 2019) approximate a sequence of target probability distributions using a set of weighted samples. Consider a state space model

with latent states $x_{1:T}$ and observations $y_{1:T}$, with joint density $p(x_{1:T}, y_{1:T})$. The inference task is to approximate the filtering or target distribution $p(x_t|y_{1:t})$. At each time step $t$, SMC represents the distribution using a finite set of $N$ particles $\{x_t^i\}_{i=1}^N$ and associated weights $\{w_t^i\}_{i=1}^N$ that together define an empirical measure

$$p(x_t|y_{1:t}) \approx \sum_{i=1}^N w_t^i \, \delta_{x_t^i}, \qquad (1)$$

providing a discrete approximation of the target distribution. In practice $N$ is of the order $10^2$–$10^5$. Convergence results guarantee that, under mild conditions, the particle approximation converges to the true target distribution as $N \to \infty$ (Del Moral, 2004; Johansen, 2009).

As a specific instantiation of SMC, we consider a standard bootstrap particle filter with adaptive resampling (Doucet et al., 2001). Given particles at time $t - 1$, the update at time $t$ consists of the following steps:

1. **Resample.** Sample $N$ ancestor indices $a_t^i$ with systematic resampling (Kitagawa, 1996) if adaptive resampling is triggered. Otherwise copy all particles, i.e. $a_t^i = i$.

2. **Propagate.** Sample new particles from the transition model, $x_t^i \sim p(x_t|x_{t-1}^{a_t^i})$.

3. **Weight.** Assign unnormalized weights using the likelihood, $\tilde{w}_t^i = p(y_t|x_t^i)$, and normalize $w_t^i = \frac{\tilde{w}_t^i}{\sum_{j=1}^N \tilde{w}_t^j}$.

Repeating these steps for $t = 1, \ldots, T$ yields a particle-based approximation of the target distributions. SMC can also be used for static target distributions, i.e., $p(x_t) \equiv p(x)$ for all $t$, by defining an artificial sequence of intermediate target distributions that progressively transform a simple initial distribution into the desired target (e.g., via likelihood tempering) (Neal, 2001; Del Moral et al., 2006).

## 3.2. Problem Formulation

Scientific model discovery aims to infer a model $m$ that reproduces the observable behavior of a system of interest $m^*$. Information about $m^*$ is available only through an observed set $\boldsymbol{x}_o = \{(x_o^j, c_o^j)\}_{j=1}^M$, where each observation $x_o^j$ is generated by $m^*$ under context $c_o^j$. The context may include, e.g., initial conditions or experimental parameters. We assume the observed data pairs are independent and identically distributed (iid).

A model $m$ is an algorithm written in code that probabilistically maps context $c \in C$ and model parameters $\theta \in \Theta$ to data $x \in X$. The model parameters $\theta$ are unknown but fall within a prior distribution $p(\theta)$ (details in App. D.1). For notational simplicity, we abbreviate $m(c, \theta) \equiv m$.

From a probabilistic perspective, model discovery corresponds to inference over models conditioned on the observed data. The inference target is a distribution over models $p(m|\boldsymbol{x}_o)$ that assigns higher probability to models whose simulated behavior matches the observed behavior of $m^*$ across contexts. This distribution is defined implicitly via a likelihood and a prior: the likelihood $p(x_o|m)$ is induced by executing the simulator model defined by $m$ with latent parameters $\theta \sim p(\theta)$. The prior $p(m)$ corresponds to the marginal distribution over model implementations induced by an LLM-based generative process. The distribution $p(m|\boldsymbol{x}_o)$ is generally intractable due to model space size and the absence of an explicit likelihood.

### 3.3. SMC-Based Model Discovery with ModelSMC

ModelSMC approximates the posterior $p(m|\boldsymbol{x}_o)$ via a population of $N$ candidate models that is iteratively refined using SMC. Since the true system $m^*$ is fixed, the SMC time index corresponds to iterative refinement rather than temporal evolution. Unlike classical SMC, each particle corresponds to a candidate model rather than a dynamical state.

Each particle represents a candidate model and is associated with a weight reflecting its posterior relevance. In addition to the model implementation, particles store auxiliary context such as evaluation feedback. A particle consists of the tuple of model, weight and context $(m, w, c)$.

ModelSMC proceeds for $K$ iterations. In each iteration $k$, particles are resampled, propagated to generate new candidate models, and weighted according to their likelihood under the observed data. Each iteration consists of three steps: (i) resampling, (ii) propagation, and (iii) weighting. Algorithm 1 summarizes the complete procedure.

**Resampling.** The effective sample size (ESS, Eq. 2), a measure of how many particles carry meaningful weight, is computed for the particles in the population. If the ESS is below a threshold $\tau_{\text{ESS}}$, systematic resampling (Kitagawa, 1996) is applied to rejuvenate the population, i.e. to rebalance the population by discarding low-weight particles. Otherwise, the population remains unaltered.

**Propagation.** Each particle in the population is propagated via a mixture kernel (Eq. 4). With probability $\alpha$ the particle is cloned unchanged, and with probability $1 - \alpha$ we sample a new implementation by drawing from the LLM conditioned on (i) the models along its ancestry, (ii) their associated contextual feedback, and (iii) a fixed task-specific prompt $c_{\text{task}}$ encoding domain knowledge, implementation constraints, and evaluation criteria. Simulating and evaluating new models is computationally expensive; cloned particles avoid this cost entirely, motivating the mixture kernel. The number of particles in the population is constant by construction of the propagation scheme, while each itera-

---

**Algorithm 1** ModelSMC (for $i = 1, \ldots, N$)

**Input:** Number of particles $N$, number of iterations $K$, cloning probability $\alpha \in [0, 1]$, ESS threshold $\tau_{\text{ESS}} > 0$, observed data $\boldsymbol{x}_o = \{(x_o^j, c^j)\}_{j=1}^M$, task prompt $c_{\text{task}}$, initial particle and weight $(m_0, w_0 = 1)$.

1: **for** $k = 1, \ldots, K$ **do**
2:    ▷ **1. Resample**
    Compute effective sample size (ESS)

$$\text{ESS}_k = \frac{1}{\sum_{i=1}^N (w_k^i)^2}. \quad (2)$$

3:    Sample ancestor indices via systematic resampling $\mathcal{R}$.

$$a_k^i \sim \begin{cases} \delta_i & \text{if } \text{ESS}_k \geq \tau_{\text{ESS}}, \\ \mathcal{R}\left(\{w_{k-1}^j\}_{j=1}^N\right) & \text{else.} \end{cases} \quad (3)$$

4:    ▷ **2. Propagate**
5:    Propagate each particle $m_k^i$. Sample $z_i \sim \mathcal{U}[0,1]$, then

$$m_k^i \sim \begin{cases} \delta_{m_{k-1}^{a_k^i}}, & \text{if } z_i < \alpha, \\ p_{\text{LLM}}(m \mid m_{k-1}^{a_k^i}, c_{k-1}^{a_k^i}, c_{\text{task}}), & \text{else.} \end{cases} \quad (4)$$

6:    ▷ **3. Weight**
7:    Compute surrogate likelihood $p_\phi(x \mid m_k^i, \theta, c)$.
8:    Compute marginal likelihood for each $(x_o^j, c_o^j) \in \boldsymbol{x}_o$

$$p(x_o^j \mid m_k^i) = \int p(\theta)\, p_\phi(x_o^j \mid m_k^i, \theta, c_o^j)\, d\theta. \quad (5)$$

9:    Update weights $\tilde{w}_k^i$ and normalize to obtain $w_k^i$

$$\tilde{w}_k^i = \prod_{j=1}^M p(x_o^j \mid m_k^i). \quad (6)$$

10:    Generate feedback $c_k^i = \text{LLM}(m_k^i, w_k^i, c_{\text{prompt}})$.
11: **end for**
**Output:** Particles and weights $\{m_K^i, w_K^i\}_{i=1}^N$.

---

tion generates $N \cdot (1 - \alpha)$ new particles in expectation.

The context $c_{k-1}^a$ contains the information on which the LLM is conditioned in the generation of a new implementation. In addition to the implementations of the resampled particle's ancestors, it also contains LLM-based performance assessments of the ancestors and task-specific information (prompting details in Apps. D.2 and H). To control computational cost and prompt length, only a limited number of the most recent ancestry elements are included. The probability $\alpha$ controls the trade-off between exploration and exploitation: cloned particles require no new simulations or likelihood evaluations, preserving computational resources, while newly proposed particles explore the model space. Selection among clones is handled implicitly by resampling based on likelihood weights.

This iterative adaptation of the proposal plays a role analogous to likelihood tempering in classical SMC samplers (Del Moral et al., 2006): early iterations explore the model space broadly, while later iterations concentrate proposals around high-likelihood candidates as ancestry and feedback accumulate. Unlike explicit tempering, this annealing occurs in the proposal distribution rather than the target, and without formal variance-control guarantees.

**Weighting.** Each generated model $m_k^i$ is assigned a weight proportional to the likelihood of the observed data,

$$\tilde{w}_k^i = p(\boldsymbol{x}_o \mid m_k^i). \tag{7}$$

Assuming iid observed data, the likelihood factorizes across samples (Eq. 6). Weighting models by likelihood is consistent with the posterior $p(m|\boldsymbol{x}_o)$ as target (under idealized assumptions; derivation in App. B).

The likelihood $p(\boldsymbol{x}_o|m_k^i)$ induced by a model is generally intractable. To obtain an estimate of the likelihood, we can sample from the likelihood by evaluating the model at different values of $(\theta, c)$. Specifically, we sample a synthetic dataset $\theta \sim p(\theta)$, $c \sim p(c)$, $x \sim p(x|m_k^i(\theta, c))$ and train a conditional density estimator $p_\phi(x|m_k^i, \theta, c)$. This approach is also known as Neural Likelihood Estimation (Papamakarios et al., 2019, NLE). Given a trained likelihood surrogate, the marginal likelihood for each observed data instance is obtained by integrating out $\theta$ via Monte Carlo approximation (Eq. 5). We note that the trained likelihood estimation is an approximation of the true likelihood (details in App. D.4).

For NLE, we adapt a likelihood-estimation variant derived from neural posterior estimation with prior-data fitted networks (NPE-PFN) (Vetter et al., 2025). This approach uses the pretrained tabular foundation model TabPFN as a density estimator, enabling training-free likelihood estimation (Hollmann et al., 2025). Crucially, this eliminates the need for a separate likelihood estimation training loop at each iteration of ModelSMC, which is especially beneficial in model discovery where repeated likelihood evaluations are required. The foundation model is particularly efficient with small simulation budgets. Our likelihood estimation approach allows ModelSMC to generate non-differentiable models without any explicit parameter optimization.

ModelSMC differs from prior model discovery methods in two key aspects. First, the weighting step is derived from the model posterior $p(m|\boldsymbol{x}_o)$, providing a principled probabilistic objective rather than relying on alternative discrepancy measures such as MMD, MSE, or Wasserstein distance (Gretton et al., 2012; Arjovsky et al., 2017). Second, ModelSMC marginalizes over model parameters $\theta$ during discovery instead of optimizing point estimates, thereby reducing sensitivity to parameter uncertainty. For downstream use of discovered models, we perform parameter

inference using neural posterior estimation (NPE) (Lueckmann et al., 2017; Papamakarios & Murray, 2016) and NPE-PFN (Vetter et al., 2025) (details in App. D.3).

### 3.4. Theoretical Insights

Since our framework is based on SMC, we connect ModelSMC to classical SMC theory. Specifically, we apply a standard SMC consistency result (Chopin, 2004; Del Moral, 2004) to the model discovery setting. showing that an idealized ModelSMC sampler provides an asymptotically unbiased particle approximation of the model posterior $p(m|\boldsymbol{x}_o)$.

**Theorem 3.1** (Consistency of ModelSMC)**.** *Let $\pi(m) \propto p(\boldsymbol{x}_o|m)p(m)$ be a fixed target distribution over models, and let $q(m'|m)$ denote an idealized proposal kernel induced by the propagation step. Assume (i) support coverage, i.e., any $m$ with $\pi(m) > 0$ can be generated with non-zero probability; (ii) uniformly bounded importance weights; (iii) conditional independence of propagated particles given resampling; and (iv) exact likelihood evaluation, i.e., the importance weights are computed using the true marginal likelihood $p(\boldsymbol{x}_o|m)$. Let $\{(m^i, w^i)\}_{i=1}^N$ be the particle population at a fixed iteration. Then, for any bounded test function $\varphi$,*

$$\sum_{i=1}^N w^i \varphi(m^i) \xrightarrow{\mathbb{P}} \mathbb{E}_\pi[\varphi(m)] \quad as \quad N \to \infty, \tag{8}$$

*with asymptotic variance $\mathcal{O}(1/N)$. The result holds in the presence of (adaptive) resampling.*

The proof is provided in App. C. Theorem 3.1 characterizes an idealized reference sampler with exact likelihoods and a prior-matching proposal. The practical algorithm deviates from this setting through surrogate likelihoods and a context-conditioned LLM proposal. Thus, the theorem is best understood as a proof of concept that the probabilistic framing of model discovery admits formal analysis through Bayesian inference over models.

## 4. Experiments

### 4.1. Validation of ModelSMC for Model Discovery

We first validate the core probabilistic inference mechanism underlying ModelSMC in a controlled setting where the ground-truth model is known, and the model proposal is not affected by the LLM. We consider an LLM-free variant of ModelSMC on a finite model space with 20 candidates. Each candidate is a 10-dimensional Gaussian mixture models with varying numbers of components and dense covariances (details in App. F.1). In this setting, the LLM-based propagation kernel component is replaced by uniformly sampling a random model from the candidate set.

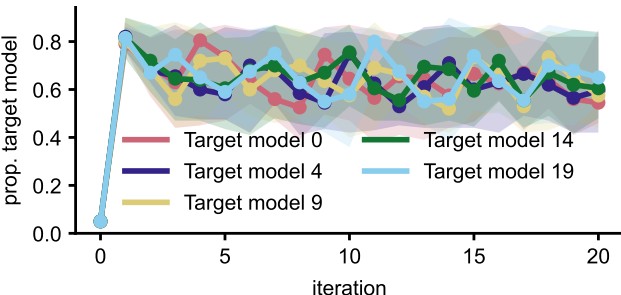

*Figure 2.* **Proportion of the target model in the population for LLM-free ModelSMC.** Median proportion of the target model in the population, averaged over ten random seeds, for five target models. Shaded regions depict the 100% confidence interval of the median, estimated using 10,000 bootstrap replicates.

*Table 1.* **Performance evaluation.** We evaluate the median metrics for the best particles over ten discovery runs. Results including confidence intervals are reported in Appendix Table G-1.

|  | FunSearch+ | ModelSMC | ModelSMC $N = 1$ |
|---|---|---|---|
|  | $-\log p(\boldsymbol{x}_o|m)$ ($\downarrow$) | | |
| SIR | -48794.12 | -45834.78 | -57028.36 |
| HH | 254.58 | 256.00 | 243.27 |
| Kidney | N/A | 42.85 | 36.54 |

Figure 2 shows that the target model dominates the particle population for all evaluated configurations. Furthermore, a characteristic oscillation of the proportion of the target model and a upper bound of approximately 80% can be observed. This is the expected behaviour for the evaluated propagation kernel and hyper parameter choice: resampling concentrates the population on the target model, while the random replacement of particles in the propagation step gradually reduces its proportion until the next resampling is triggered (details in App. G.1).

This experiment isolates the inference component of ModelSMC and shows that, in a finite model space, the algorithm concentrates posterior mass on the true data-generating process, behaving as a consistent posterior sampler over models. Thus, once the ground-truth model is proposed by the LLM, ModelSMC reliably identifies it. These results verify the inference procedure of ModelSMC.

### 4.2. Quantitative Results

We perform a quantitative evaluation of ModelSMC across three model discovery tasks: a synthetic epidemiological system with known ground-truth dynamics (SIR; App. F.2), a pharmacological simulator of kidney aldosterone regulation implemented in R (Sec. 4.3), and a neuroscience Hodgkin-Huxley (HH) neuron simulator (Sec. 4.4). ModelSMC targets model discovery as mechanistic equations for complex simulator models. We therefore compare against two baselines matching this scope: (i) FunSearch+ a modification of FunSearch (Romera-Paredes et al., 2024) that optimizes free model parameters and uses likelihood-based weights (Castro et al., 2025), and (ii) ModelSMC with a single particle ($N$=1) where the most recent particle is propagated ($\alpha = 0.0$), which removes the population-based inference of the full algorithm. Together, these two baselines represent the two dominant paradigms in LLM-based model discovery: evolutionary search (FunSearch+) and sequential non-probabilistic model exploration (Mod-

elSMC $N$=1), the latter being representative of related sequential methods (Holt et al., 2025; Ma et al., 2024; Li et al., 2024; Shojaee et al., 2025). Crucially, ModelSMC $N$=1 differs from ModelSMC only in removing the SMC framework, isolating the contribution of population-based probabilistic inference. FunSearch+ is Python-specific, which prevents direct application to the R-based kidney simulator. We do not compare with classical symbolic regression because it is typically limited to low-complexity functions (Schmidt & Lipson, 2009), and black-box predictive models, such as transformers, because they lack mechanistic interpretability.

Across all tasks, ModelSMC achieves comparable quantitative performance (Table 1). On both synthetic and real-world tasks, it attains competitive marginal and conditional likelihoods at comparable computational budgets (Fig. G-4), remaining on par with baseline methods. Unlike FunSearch+, ModelSMC handles any coding language.

These results indicate that framing model discovery as an inference problem does not compromise quantitative performance. Instead, ModelSMC discovers competitive mechanistic models while providing the probabilistic foundation that enables additional interpretability and analysis.

### 4.3. Pharmacological Kidney Model

We evaluate ModelSMC on a real-world quantitative systems pharmacology (QSP) problem. Specifically, we adapt a simulator model of plasma potassium regulation in the kidney mediated by aldosterone, a key steroid hormone (Maddah & Hallow, 2022). The original model is implemented in R using RxODE2 (Wang et al., 2016; R Core Team, 2025) and is derived from first principles of pharmacokinetics and pharmacodynamics, capturing renal excretion, tubular reabsorption, and hormonal feedback. The model's aldosterone mechanism is empirically fitted rather than biophysically derived (Maddah & Hallow, 2022).

We intentionally replace the aldosterone mechanism with a constant term (Fig. 3a, left), thereby introducing a targeted structural misspecification. ModelSMC is tasked with inferring the missing regulatory dynamics from the data. The inference problem is particularly challenging due to ex-

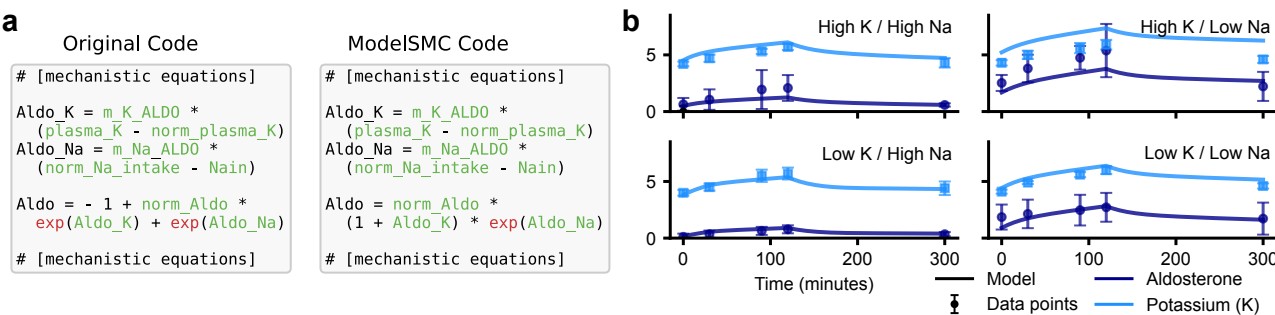

*Figure 3.* **Systems pharmacology kidney model with experimental data.** **(a)** Original code snippet in R for the aldosterone mechanism (left) and one instance of the inferred model code (right). **(b)** Posterior predictive for the code instance in (a) with real-world data points.

tremely limited data: four patient intake scenarios (high / low K $\times$ high / low Na), with a single measurement of plasma potassium and aldosterone at five time points per scenario (Dluhy et al., 1972) (details in App. F.3).

Across all seeds, ModelSMC assigns high posterior weight to a narrow family of aldosterone feedback mechanisms whose functional form closely matches the originally proposed model (Fig. 3a, right, for one model). Although individual particles differ in structural details, models encoding qualitatively different regulatory hypotheses receive negligible posterior weight. Posterior predictive simulations from high-probability models closely track observed plasma potassium and aldosterone measurements, despite the limited data (Fig. 3b).

From an inference perspective, this experiment illustrates strong posterior concentration over a mechanistically coherent class rather than the selection of a single best-fitting model. The absence of competing high-weight alternatives indicates that the available data are sufficient to rule out broad classes of hypotheses, diagnosing structural inadequacy rather than parameter uncertainty. Compared to methods producing a single point estimate, ModelSMC yields a posterior over low-complexity mechanisms that both generalize in posterior predictive checks and quantify uncertainty over plausible mechanisms.

### 4.4. Hodgkin-Huxley Model

Finally, we evaluate ModelSMC on a neuroscience simulator of neuronal membrane dynamics. Specifically, we consider the single-compartment Hodgkin-Huxley (HH) model of action potential generation (Hodgkin & Huxley, 1952; Pospischil et al., 2008), which describes membrane voltage dynamics via nonlinear ion-channel conductances and gating variables. As baseline, we consider the simulator from Gonçalves et al. (2020), consisting of eight biophysical parameters and explicit ion channels for sodium, potassium, and leak current. While this model provides a strong description of neuronal dynamics, it can exhibit system-

atic discrepancies when fitted to real recordings (Gao et al., 2023; Bernaerts et al., 2025). Observations consist of membrane voltage traces summarized by seven standard summary statistics (Gonçalves et al., 2020). We consider 10 real voltage recordings from the Allen Cell Types Database (Allen Institute for Brain Science, 2015).

In contrast to prior work focusing exclusively on parameter inference (Gonçalves et al., 2020; Vetter et al., 2025), we treat the HH model as a structural inference problem. Taking the baseline HH model, ModelSMC is tasked with inferring additional ion-channel mechanisms that improve model specification and reduce systematic mismatch to data. This setting is particularly challenging due to the limited number of real-world observations, the already high fidelity of the baseline model, and the need to detect subtle but consequential structural deficiencies rather than gross misspecification (details in App. F.4).

ModelSMC shows rapid posterior concentration while continuing to explore multiple high-likelihood structural variants over successive iterations (Fig. 4a). For a representative run, particle ancestry reveals a diverse sampling of the model space with distinct ion-channel extensions of the baseline HH model (Fig. 4a, opaque). Across runs, high posterior weight is consistently assigned to models introducing an M-type slow potassium current, also paired with a secondary channel such as persistent sodium ($I_{NaP}$), A-type, or HCN ($I_h$). Within these families, refinements such as a corrected potassium reversal potential ($E_K$) receive increased posterior weight and improve posterior predictive agreement with recorded voltage traces, particularly for spike-frequency adaptation (Fig. 4b,c).

Rather than identifying a single revised HH model, ModelSMC yields a posterior over mechanistically interpretable extensions, with comparable explanatory power. The posterior exposes residual non-identifiability in ion-channel composition while isolating robust conclusions, i.e., the extension with additional slow currents. Thus, posterior mass serves as a diagnostic: low-weight regions indicate gen-

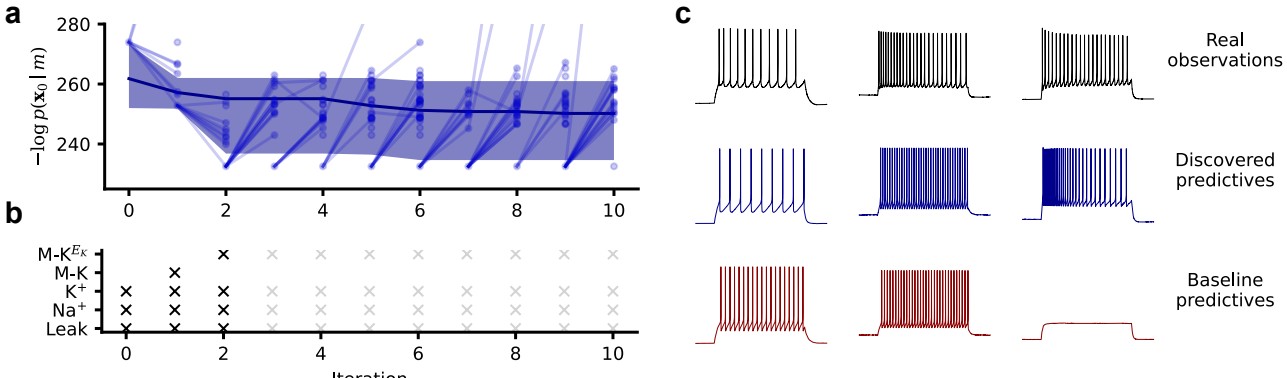

*Figure 4.* **Hodgkin–Huxley model with Allen cell types database data.** Results are shown for one representative random seed from the 10 runs, used consistently across panels. **(a)** ModelSMC convergence across runs, showing the mean (solid line) and 95% percentiles (shaded). With lower opacity, we overlay the full model ancestry of the selected run. **(b)** For the selected run, highlighting of inferred ion-channel mechanisms at successive improvement stages. **(c)** Posterior predictive voltage simulations for real observations from the Allen database, comparing the baseline HH model to the best model identified in the selected run.

uine structural mismatch, while clusters of high-probability models reveal symmetries and ambiguities that are invisible to single-model discovery.

### 4.5. Posterior Analysis

Particle weights in ModelSMC represent a posterior distribution, enabling evaluation of entire families of modelling hypotheses beyond single-model selection. We demonstrate this for the ion-channel extensions from Sec. 4.4, showing which structural choices are supported by the data and which are ruled out.

We apply a three-step LLM-assisted pipeline to all $1,445$ particles across 10 seeds. First, a stratified sample of $\approx 15\%$ of particles is used to extract free-form descriptions of the added channels, and consolidated into a two-level taxonomy of 11 structural subtypes across families ($I_M$, $I_{NaP}$, $I_h$, $I_A$, and multi-channel combinations). Second, all particles are classified against this taxonomy. Third, weights are compared across subtypes (details in App. F.5).

Among the added channels, $I_M$ variants (M-type K$^+$ currents) consistently achieve the lowest weights across subtypes (Fig. 5), while $I_M$ with a second channel including $I_{NaP}$, $I_h$, and $I_A$ variants receive substantially higher weights. The unextended base model itself receives comparably low weights, reflecting its already high fidelity, but is consistently outperformed by $I_M$ extensions. This ranking is stable across independent runs, with $I_M$ placing first in 9 out of 10 seeds (App. Fig. F-2), demonstrating that ModelSMC arrives at reproducible posterior conclusions rather than spreading mass uniformly across alternatives (details in App. F.5).

*Table 2.* **Ablation study summary on the Hodgkin-Huxley task.** Median $-\log p(\boldsymbol{x}_o|m)$ with 90% confidence intervals for each variant relative to the default (first row). The ablation results are based on 5 runs. Underlined cells indicate configurations outperforming the default. The confidence intervals are estimated using 10,000 bootstrap replicates.

| Axis | Ablation | $-\log p(\boldsymbol{x}_o|m)$ ($\downarrow$) |
|---|---|---|
| Default | | 256.00 [255.05, 262.03] |
| Weighting | MSE weights | 264.65 [250.90, 341.82] |
| Simulations | 1000 | 270.07 [250.34, 292.69] |
| | 500 | 263.89 [254.03, 284.50] |
| | 200 | 266.91 [264.11, 292.58] |
| Prompt | Reduced | 262.18 [249.96, 282.15] |
| Feedback | LLM + metrics | 254.82 [233.71, 262.89] |
| | Metrics only | 258.30 [245.49, 262.89] |
| Pool size | $N = 150, K = 5$ | 267.01 [260.03, 278.91] |
| | $N = 10, K = 75$ | 258.25 [254.10, 277.81] |
| | $N = 5, K = 150$ | 248.02 [237.97, 267.01] |
| LLM | GPT-5 mini | 254.49 [242.22, 256.04] |

### 4.6. Ablation Study

We evaluate the sensitivity of ModelSMC to its core design choices on the Hodgkin-Huxley task (Sec. 4.4), systematically varying six axes while holding all others fixed at the default configuration. We vary the *weighting scheme* (marginal likelihood vs. MSE) to test whether the principled Bayesian weights are necessary. We reduce the *simulation budget* for likelihood estimation (5,000 to 200–1,000) to understand the compute–quality trade-off. We simplify the *prompt* to assess how much prompt engineering contributes. We change the *feedback format* (LLM

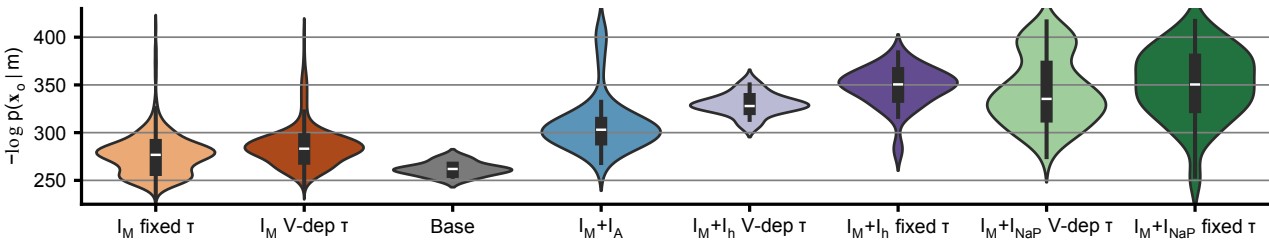

*Figure 5.* **Posterior mass analysis.** Distribution of weights per ion-channel subtype, pooled across all 1,445 particles and 10 seeds of the Hodgekin-Huxley experiment. Channel types are sorted and colored by family.

narrative, metrics, or both) to determine whether richer feedback improves LLM proposals. We vary *pool size* $N$ and number of iterations $K$ at a fixed total budget of $K \cdot (1-\alpha) \cdot N = 150$ new particles to study the depth-vs-diversity trade-off. Finally, we replace the *LLM backbone* with GPT-5 mini to verify robustness across models. Overall, no single variation causes a significant degradation in performance (Table 2), indicating that ModelSMC is robust to these design choices (full results in App. F.6).

## 5. Discussion

We formulated automated discovery of mechanistic models as Bayesian inference over programs. We introduced ModelSMC, an SMC-based algorithm that uses LLMs as proposal distributions. Unlike prior agentic systems, ModelSMC directly approximates the posterior $p(m|\boldsymbol{x}_o)$ by marginalizing over parameters. This approach produces a weighted population of models, revealing which hypotheses concentrate posterior mass, which remain uncertain, and which are non-identifiable. Across synthetic and real-world systems, this enables diagnosing which mechanistic hypotheses are strongly supported, ruled out, or remain as plausible alternatives under limited data.

A natural question is whether automated discovery systems generate genuinely novel scientific insights or primarily rediscover known mechanisms. We emphasize that our primary contribution is methodological: we introduce a general inference framework in which such questions can be meaningfully posed. Whether novel mechanisms emerge depends on the application, data, and modeling assumptions, and is orthogonal to the validity of the framework itself. Practically, ModelSMC applies to settings where candidate models can be executed and admit reliable likelihood approximation. In such domains, probabilistic reasoning over models is well-defined and informative.

**Limitations.** We introduce an inference framework for model discovery, instantiated by ModelSMC. Limitations arise at algorithmic and framework levels: First, ModelSMC is computationally expensive due to repeated simulation and likelihood estimation for each candidate model.

This could be alleviated by multi-fidelity simulation, adaptive particle budgets, or early-rejection strategies from SMC literature. Second, posterior weights rely on surrogate likelihoods, which may introduce bias under misspecification. Incorporating uncertainty-aware or robust likelihood estimation is an important direction. Third, the discovery process is bottlenecked by the LLM's ability to propose relevant model structures. Improvements in LLM capability, domain-specific pretraining, or retrieval-augmented generation will directly enhance discovery performance (Lewis et al., 2020). Finally, while posterior weights enable principled ranking and clustering of hypotheses (Sec. 4.5), models remain high-dimensional programs without an explicit geometry. Developing notions of similarity or clustering over model space would enable richer posterior analysis and enhance inference with semantic structure over program spaces.

**Conclusion.** By reframing LLM-driven model discovery as probabilistic inference, we provide a framework that unifies proposal, refinement, and selection under a single probabilistic objective. ModelSMC provides one concrete instantiation of this view, combining LLM-based proposals with likelihood-based weighting to approximate a posterior over mechanistic models. Crucially, this connection enables direct use of established Monte Carlo methodology, including particle rejuvenation moves (Gordon et al., 1993), SMC samplers for static targets (Del Moral et al., 2006), adaptive resampling schemes (Doucet et al., 2001; Johansen, 2009; Del Moral et al., 2012), and annealed or tempered objectives (Neal, 2001). More broadly, this perspective situates recent LLM-based discovery systems within a well-studied inference framework, providing a common language for comparison, analysis, and systematic development of future methods.

## Acknowledgements

We thank Manuel Gloeckler for feedback on the manuscript, members of the Mackelab for discussions, and Julian Schwab for domain expertise on the QSP experiment. We thank Anthropic for API credits provided through the AI for Science Program. This work was

funded by the German Research Foundation (DFG) under Germany's Excellence Strategy – EXC number 2064/1 – 390727645 and SFB 1233 'Robust Vision' (276693517), and the Boehringer Ingelheim AI & Data Science Fellowship Program. SW is a member of the International Max Planck Research School for Intelligent Systems (IMPRS-IS).

## Impact Statement

This paper presents work whose goal is to advance the field of Machine Learning. There are many potential societal consequences of our work, none of which we feel must be specifically highlighted here.

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

# Appendix Contents

## A. Software and Computational Resources

All methods are implemented in Python. For general computations, NumPy (Harris et al., 2020) and PyTorch (Paszke et al., 2019) are used. Plotting is performed using Matplotlib (Hunter, 2007). To store and handle experimental results, we use pandas (pandas development team, 2020). NLE and NPE are performed using SBI's (Boelts et al., 2025) implementation of the corresponding methods. In the kidney experiment (Sec. 4.3), the models are implemented in R (R Core Team, 2025). To parallelize the evaluation of multiple particles in ModelSMC, we use Ray (Moritz et al., 2018). LLMs are accessed via DSPy (Khattab et al., 2022; 2024), which automatically assembles prompts and returns structured outputs with named fields. Unless stated otherwise, we run all experiments with `Claude Sonnet 4.6` (Anthropic, 2026) as language model and have an ablation using `GPT-5-mini` (OpenAI, 2025). Within individual discovery runs, the same language model is used for propagation and feedback generation (App. D.2). Code is available at `https://github.com/mackelab/ModelSMC`.

NPE-PFN (App. D.3) and NLE-PFN (App. D.4) are run on an A100 GPU with 40GB of memory with associated CPUs. NLE and NPE are run on the CPUs only. The particle propagation and evaluation in ModelSMC can be parallelized. In our experiments, we run up to five particles in parallel. In contrast, the baseline method ModelSMC $N=1$ and FunSearch+ are run sequentially.

## B. Derivation of ModelSMC Weights from a Static Bayesian Target

This section derives the weighting rule used in ModelSMC from a Bayesian posterior over model programs. The derivation makes explicit how likelihood-based weights arise from an SMC construction, with an underlying static inference problem.

**Inference target.** Given observed data $\boldsymbol{x}_o = \{(x_o^j, c_o^j)\}_{j=1}^M$, the target posterior over models is

$$p(m \mid \boldsymbol{x}_o) \;\propto\; p(\boldsymbol{x}_o \mid m)\, p(m), \tag{B-1}$$

with marginal likelihood

$$p(\boldsymbol{x}_o \mid m) = \prod_{j=1}^M p(x_o^j \mid m), \qquad p(x_o^j \mid m) = \int p(\theta)\, p(x_o^j \mid m, \theta, c_o^j)\, d\theta. \tag{B-2}$$

The prior $p(m)$ is implicit and corresponds to the marginal distribution over models induced by the LLM-based generative process in the absence of conditioning on observed data.

For notational simplicity, we drop the explicit conditioning on the observed context $c_o^j$, including particle-specific feedback variables, for the remainder of the derivation. This corresponds to an idealized setting in which the model proposal distribution depends only on the previously generated model. In practice, the proposal is conditioned on additional contextual information, which leads to approximate rather than exact importance weights (see Remarks below at the end of this section).

**Artificial SMC target distribution.** To apply SMC, we introduce an artificial sequence of latent variables $(m_1, \ldots, m_K)$, where each $m_k$ represents a candidate model. We define the unnormalized target distribution on the extended space as

$$\tilde{\pi}(m_{1:K}) := p(m_{1:K}, \boldsymbol{x}_o) = \prod_{k=1}^K p(m_k \mid m_{1:k-1})\, p(\boldsymbol{x}_o \mid m_k). \tag{B-3}$$

The corresponding normalized target is

$$\pi(m_{1:K}) = \frac{\tilde{\pi}(m_{1:K})}{Z}, \qquad Z = p(\boldsymbol{x}_o)^K. \tag{B-4}$$

Marginalizing $\pi(m_{1:K})$ over $m_{1:k-1}$ yields the posterior over models under the LLM-induced prior, $p_{\text{LLM}}(m_k \mid \boldsymbol{x}_o)$ for every $k$.

Thus, the sequence is purely algorithmic and does not represent a temporal evolution of the system.

**Proposal distribution.** At iteration $k$, ModelSMC generates a new model using the same LLM-based generative mechanism. In practice, this proposal is conditioned on the previous model, particle-specific feedback, and task context, yielding

$$q_k(m_k \mid m_{1:k-1}) := p_{\text{LLM}}(m_k \mid m_{1:k-1}, c_{1:k-1}, c_{\text{task}}). \tag{B-5}$$

For the purpose of the derivation, we absorb all contextual variables into the conditioning and consider an idealized proposal that depends only on $m_{1:k-1}$.

**Incremental importance weights.** The standard SMC incremental weight is

$$\tilde{w}_k = \frac{\tilde{\pi}(m_{1:k})}{\tilde{\pi}(m_{1:k-1}) \, q_k(m_k \mid m_{1:k-1})}. \tag{B-6}$$

Using Eq. B-3, we obtain

$$\tilde{w}_k = \frac{\prod_{t=1}^{k} p(m_t \mid m_{1:t-1}) \, p(\boldsymbol{x}_o \mid m_t)}{\left( \prod_{t=1}^{k-1} p(m_t \mid m_{1:t-1}) \, p(\boldsymbol{x}_o \mid m_t) \right) q_k(m_k \mid m_{1:k-1})} \tag{B-7}$$

$$= \frac{p(m_k \mid m_{1:k-1}) \, p(\boldsymbol{x}_o \mid m_k)}{q_k(m_k \mid m_{1:k-1})}. \tag{B-8}$$

We refer to this as a prior-matching proposal when the proposal coincides with the conditional generative distribution defining the implicit model prior in an idealized setting, i.e.,

$$q_k(m_k \mid m_{1:k-1}) = p(m_k \mid m_{1:k-1}). \tag{B-9}$$

Under this assumption, the transition terms cancel exactly, yielding the idealized incremental weight

$$\tilde{w}_k = p(\boldsymbol{x}_o \mid m_k). \tag{B-10}$$

In practice, the proposal is conditioned on additional contextual information (e.g., data-dependent feedback, task context, or finite context windows), and the exact incremental weight would take the form

$$\tilde{w}_k^{\text{true}} = p(\boldsymbol{x}_o \mid m_k) \, \frac{p(m_k \mid m_{1:k-1})}{q_k(m_k \mid m_{1:k-1}, \boldsymbol{x}_o, c_{1:k-1})}. \tag{B-11}$$

Because this context-conditioned proposal density is implicit and intractable for large language models, ModelSMC omits the correction factor and uses $p(\boldsymbol{x}_o \mid m_k)$ as a practical, approximate importance weight.

**Resulting weighting rule.** Thus, under a prior-matching proposal, the incremental SMC weight at iteration $k$ is precisely the marginal likelihood of the newly proposed model. This yields the weighting rule used in ModelSMC as the likelihood

$$\tilde{w}(m) \propto p(\boldsymbol{x}_o \mid m). \tag{B-12}$$

**Remarks.**

- **Implicit model prior.** The prior $p(m)$ is implicit and defined by the marginal distribution over models induced by the LLM generative process. The proposal distribution corresponds to its conditional transitions. ModelSMC therefore performs inference under this implicit prior rather than an explicit analytic one.

- **Approximate prior matching.** The derivation assumes an idealized prior-matching proposal that ignores contextual conditioning. In practice, the LLM proposal depends on feedback and task-specific context, so the exact importance ratio for the SMC weight would additionally include a proposal correction term. Because this proposal density is implicit and intractable, ModelSMC uses likelihood-based weights as a practical approximation, corresponding to inference under an implicit LLM-induced prior.

- **Static target.** The sequence $(m_1, \ldots, m_K)$ is artificial and introduced solely to enable SMC; all marginals correspond to the same static posterior $p(m \mid \boldsymbol{x}_o)$.

- **Tempering extension.** The current formulation uses the full likelihood $p(\boldsymbol{x}_o \mid m_k)$ at every iteration. This can be generalized by introducing a schedule $\beta_1 \leq \cdots \leq \beta_K = 1$ and replacing the target at iteration $k$ with $p(\boldsymbol{x}_o \mid m_k)^{\beta_k}$, recovering a standard tempered SMC sampler (Del Moral et al., 2006). The existing temperature parameter $\tau$ (Eq. D-38) could be adapted to serve this role.

## C. Proof of ModelSMC Convergence

**Theorem C.1** (restated Theorem 3.1). *Let $\pi(m) \propto p(\boldsymbol{x}_o|m)p(m)$ be a fixed target distribution over models, and let $q(m'|m)$ denote an idealized proposal kernel induced by the propagation step. Assume (i) support coverage, i.e., any $m$ with $\pi(m) > 0$ can be generated with non-zero probability; (ii) uniformly bounded importance weights; (iii) conditional independence of propagated particles given resampling; and (iv) exact likelihood evaluation, i.e., the importance weights are computed using the true marginal likelihood $p(\boldsymbol{x}_o|m)$. Let $\{(m^i, w^i)\}_{i=1}^N$ be the particle population at a fixed iteration. Then, for any bounded test function $\varphi$,*

$$\sum_{i=1}^N w^i \varphi(m^i) \xrightarrow{\mathbb{P}} \mathbb{E}_\pi[\varphi(m)] \quad as \quad N \to \infty, \tag{C-13}$$

*with asymptotic variance $\mathcal{O}(1/N)$. The result holds in the presence of (adaptive) resampling.*

**Discussion of assumptions.**

- **Assumption (i)** (support coverage) requires that the proposal induced by the LLM assigns non-zero probability to any model with non-zero posterior mass. This assumption cannot be verified formally and depends critically on the expressive capacity and training of the underlying LLM. In practice, we assume that a large class of scientifically meaningful models, including those commonly used in the literature, their combinations, and moderate extrapolations thereof, lie within the effective support of the proposal distribution. The use of textual feedback and task-specific prompts allows the proposal to be steered toward relevant regions of model space, but does not guarantee full coverage. If this assumption is violated, ModelSMC converges to the posterior restricted to the subset of models reachable by the proposal, resulting in asymptotic bias.

- **Assumption (ii)** (uniformly bounded importance weights) ensures finite variance of the Monte Carlo estimator. While difficult to verify formally, it is encouraged in ModelSMC by finite surrogate likelihood outputs.

- **Assumption (iii)** is standard in SMC analyses and is satisfied by resampling followed by conditionally independent propagation (Del Moral, 2004). In ModelSMC, propagation additionally depends on particle context, so strict conditional independence is not fully met in practice. The theorem therefore characterizes an idealized setting that serves as a reference point for the algorithm's behavior.

- **Assumption (iv)** formalizes an idealized setting in which the marginal likelihood $p(\boldsymbol{x}_o|m)$ is available exactly. In ModelSMC, this likelihood is approximated using a surrogate density estimator using NLE (Papamakarios et al., 2019) or NLE-PFN (details in App. D.4). The theorem therefore characterizes the asymptotic behavior of an idealized sampler and serves as a reference point for the practical algorithm, as is standard in analyses of likelihood-free SMC methods (Del Moral et al., 2006).

The following argument establishes consistency for an idealized importance sampling or SMC scheme in which the proposal distribution $q_k$ is known and used to compute exact importance weights. In ModelSMC, the proposal is implicit and context-dependent, so the theorem applies to the corresponding idealized sampler that motivates the practical algorithm.

*Proof.* The proof follows classical importance sampling arguments and their extension to SMC samplers for static target distributions, adapted to the ModelSMC setting (Robert et al., 1999; Del Moral, 2004; Del Moral et al., 2006).

**Step 1: Target expectation and proposal identity.** Fix an iteration $k$. Conditional on the resampling step at iteration $k-1$, the propagation step induces a proposal distribution

$$q_k(m) := p_{\mathrm{LLM}}(m \mid m_{k-1}), \tag{C-14}$$

where $m_{k-1}$ denotes a resampled ancestor. By assumption (i), the support of $q_k$ contains the support of $\pi$.

For any bounded test function $\varphi$, we have

$$\mathbb{E}_\pi[\varphi(m)] = \int \varphi(m)\, \pi(m)\, dm = \int \varphi(m)\, \frac{\pi(m)}{q_k(m)}\, q_k(m)\, dm = \mathbb{E}_{q_k}\left[\frac{\pi(m)}{q_k(m)}\, \varphi(m)\right]. \tag{C-15}$$

**Step 2: Importance sampling estimator.** Let $m^1, \ldots, m^N$ be iid samples from $q_k(m)$. Define (unnormalized) importance weights

$$\tilde{w}^i = \frac{\pi(m^i)}{q_k(m^i)}, \qquad w^i = \frac{\tilde{w}^i}{\sum_{j=1}^N \tilde{w}^j}. \tag{C-16}$$

The corresponding estimator is

$$\hat{I}^N := \sum_{i=1}^N w^i \, \varphi(m^i). \tag{C-17}$$

By the strong law of large numbers,

$$\sum_{i=1}^N w^i \, \varphi(m^i) \xrightarrow{\text{a.s.}} \mathbb{E}_{q_k}\left[ \frac{\pi(m)}{q_k(m)} \varphi(m) \right] = \mathbb{E}_\pi[\varphi(m)], \tag{C-18}$$

$$\sum_{i=1}^N w^i \xrightarrow{\text{a.s.}} \mathbb{E}_{q_k}\left[ \frac{\pi(m)}{q_k(m)} \right] = \int \pi(m) \, dm = 1. \tag{C-19}$$

Hence $\hat{I}^N \to \mathbb{E}_\pi[\varphi(m)]$ almost surely.

**Step 3: Variance decomposition.** Define the random variables

$$Z^i := \frac{\pi(m^i)}{q_k(m^i)} \varphi(m^i), \qquad m^i \sim q_k \text{iid} \tag{C-20}$$

Then

$$\frac{1}{N} \sum_{i=1}^N Z^i \xrightarrow{\text{a.s.}} \mathbb{E}_\pi[\varphi(m)]. \tag{C-21}$$

Moreover,

$$\text{Var}\left( \hat{I}^N \right) = \frac{1}{N} \text{Var}(Z^1). \tag{C-22}$$

**Step 4: Variance goes to zero.** By Assumption (ii) (uniformly bounded importance weights) and boundedness of $\varphi$, there exists $C < \infty$ such that

$$\left| \frac{\pi(m)}{q_k(m)} \varphi(m) \right| \leq C \quad \text{for all } m. \tag{C-23}$$

Hence $\text{Var}(Z^1) \leq C^2$, and therefore

$$\text{Var}\left( \hat{I}^N \right) = \text{Var}\left( \frac{1}{N} \sum_{i=1}^N Z^i \right) \leq \frac{C^2}{N} = \mathcal{O}(1/N) \to 0. \tag{C-24}$$

**Step 5: Convergence in probability (Chebyshev).** For any $\varepsilon > 0$, Chebyshev's inequality yields

$$\mathbb{P}\left( \left| \hat{I}^N - \mathbb{E}_\pi[\varphi(m)] \right| \geq \varepsilon \right) \leq \frac{\text{Var}\left( \hat{I}^N \right)}{\varepsilon^2} \leq \frac{C^2}{N\varepsilon^2}, \tag{C-25}$$

which converges to zero as $N \to \infty$. This establishes convergence in probability of the importance sampling estimator to $\mathbb{E}_\pi[\varphi(m)]$.

**Step 6: Effect of resampling.** The resampling step replaces the weighted particle system by an equally weighted one whose empirical measure is unbiased. Under Assumption (iii) (conditional independence of propagated particles given resampling) and bounded importance weights, resampling does not change the limit of the estimator and introduces additional variance of order $\mathcal{O}(1/N)$ only. Therefore, resampling preserves convergence in probability and the Monte Carlo rate. When adaptive resampling is used, i.e., resampling is triggered only when $\mathrm{ESS} < \tau_{\mathrm{ESS}}$, the argument extends directly. On iterations where resampling is skipped, particles retain their current weights and ancestry, which is part of the path-space reduction in Step 7 without introducing additional bias.

**Step 7: Importance sampling vs. SMC (path-space argument).** ModelSMC performs SMC updates with incremental weights of the form

$$w_k^i = \frac{\pi_k(m_k^i)}{\pi_{k-1}(m_{k-1}^i)\, q_k(m_k^i \mid m_{k-1}^i)}, \tag{C-26}$$

where $\{\pi_k\}$ is an artificial sequence of target distributions whose marginal at iteration $k$ is $\pi$. Viewing SMC as importance sampling on the extended path space $(m_0, \ldots, m_k)$, the product of incremental weights along a trajectory equals the standard importance weight $\pi(m_k)/q_k(m_k)$ for the marginal distribution. Hence, analyzing the corresponding importance sampling estimator suffices to establish consistency of the SMC estimator.

**Step 8: Conclusion.** Combining the above steps, we obtain that

$$\sum_{i=1}^{N} w^i\, \varphi(m^i) \xrightarrow{\mathbb{P}} \mathbb{E}_\pi[\varphi(m)] \quad \text{as } N \to \infty, \tag{C-27}$$

with asymptotic variance $\mathcal{O}(1/N)$. Thus, ModelSMC provides a consistent Monte Carlo approximation of the posterior $p(m \mid \boldsymbol{x}_o)$. $\qquad\square$

**Remarks.**

- **Mixture proposal.** The mixture kernel in Algorithm 1 clones a fraction of particles while propagating the remainder via the LLM. Since cloned particles retain their previous weights and the overall kernel satisfies the support coverage assumption as $N \to \infty$, this does not affect the validity of the convergence argument.

- **Idealized setting.** The theorem characterizes the asymptotic behavior of an idealized ModelSMC sampler with exact importance weights. The implemented algorithm uses approximate weights due to implicit, context-conditioned LLM proposals, and should therefore be interpreted as an approximation to this idealized sampler.

- **Prompt engineering and proposal quality.** Prompt design affects the proposal distribution implicitly by shaping the inductive biases of the LLM. From the perspective of the theorem, prompt engineering serves as a practical mechanism to improve support coverage and reduce weight variance, analogous to hand-crafted proposal kernels in traditional Monte Carlo methods. It influences finite-sample efficiency but does not affect the asymptotic consistency result.

# D. Method Details

## D.1. Note on the Definition of Prior Distributions

The dimensionality of the parameter vectors $\theta$ and the interpretation of the individual entries may vary across models. Therefore, the prior $p(\theta)$ is model- and task-specific, and is defined by the scientist using domain knowledge to identify meaningful parameter ranges, defaulting to uniform priors over physically plausible bounds, or incorporating knowledge from literature. While this is the approach taken in our experiments, an alternative is to jointly generate models and priors in the discovery experiment. For example, Holt et al. (2025) synthesize the bounds of uniform priors as part of the models.

The experimental conditions $c$ (e.g., the stimulus current and initial voltage in Hodgkin-Huxley (Sec. 4.4) or the initial population sizes in the SIR task (App. F.2)) are observable by definition. Therefore, a categorical distribution over observed training values is always available as a minimal choice for $p(c)$. Additional information about the experimental setting can provide more sophisticated distributions.

## D.2. Prompting Strategy

In the propagation step of ModelSMC (Algorithm 1 and Sec. 3.3), an LLM is used as a conditional sampler for implementations of models, implicitly synthesizing new models. To obtain meaningful implementations, we condition the LLM on text-based conditions incorporating information about previously generated models and the task to solve.

Passing information about models generated earlier in the discovery process, especially their implementations, to the LLM, is a well-established strategy in LLM-based model discovery (Holt et al., 2025; Castro et al., 2025; Romera-Paredes et al., 2024; Agarwal et al., 2025). We follow this general scheme by providing the ancestry of implementations $m_{k-1}^{a_k^i}$ and the ancestry of performance feedback $c_{k-1}^{a_k^i}$ to the LLM. Both rely on the ancestry of the resampled particle. Injecting ancestral information in the propagation step enables the LLM to reason about the effect of changes to the models on their performance.

Furthermore, we pass a task-specific context $c_{\text{task}}$ to the LLM, which does not change over the course of the discovery run. The task-specific context $c_{\text{task}}$ contains all information about the task. By passing $c_{\text{task}}$ to the LLM, we aim at triggering the LLM's broad knowledge base to further improve its ability to provide implementations that solve the given task. ModelSMC structures $c_{\text{task}}$ into three parts:

1. **System description.** We include all prior information about the task to solve. This includes information about the initial solution provided to ModelSMC or the metrics used to evaluate the generated models. The description also contains information about the scientific domain and about the data used to test the synthesized models.

2. **Signature description.** We collect all information about the framework in which the generated models are evaluated. This includes details such as the interpretation and structure of the model's input arguments and outputs.

3. **Task description.** This part summarizes the aim of the discovery run and describes how the tasks should be approached (i.e., to reason from previous attempts, use coding standards, or specific coding instructions).

The different parts of $c_{\text{task}}$ can vary across different applications, depending on the amount of prior knowledge, the environment used to evaluate the code, the scientific domain, and the specifics of the task. For the full prompts, we refer to the accompanying code.

The generation of contextual feedback $c_k^i$ of a model $m_k^i$ in Algorithm 1 is performed after model simulation, and numerical evaluation. In a single LLM call, the system description, the simulator code, and the numerical evaluation metrics are jointly provided to the LLM, which is instructed to simultaneously diagnose failure modes and propose concrete improvements. Any exceptions arising during evaluation (invalid code, numerical instability, non-terminating simulations, or timeouts) are captured and included in the performance metrics string passed to this call. The LLM returns structured feedback comprising a concise main diagnosis and up to two prioritized issues, each annotated with a severity level, a code location, and a targeted fix suggestion. The resulting feedback is designed to guide subsequent LLM propagation steps by highlighting concrete directions for model revision.

During model simulation and evaluation, exceptions may arise due to invalid code, numerical instability, non-terminating simulations, or timeouts. Such exceptions are explicitly caught and recorded. When an exception occurs, a structured description of the failure mode (e.g., error messages, timeout events, or invalid outputs) is included in the first-stage diagnosis and propagated to the LLM as part of the contextual feedback. This allows models that failed to receive targeted corrective feedback to be revised if they are later resampled as ancestors, rather than being irreversibly discarded. Figure D-1 visualizes the structure of the prompts used in the propagation and feedback steps of ModelSMC. An instantiation of the prompts is provided in App. H.

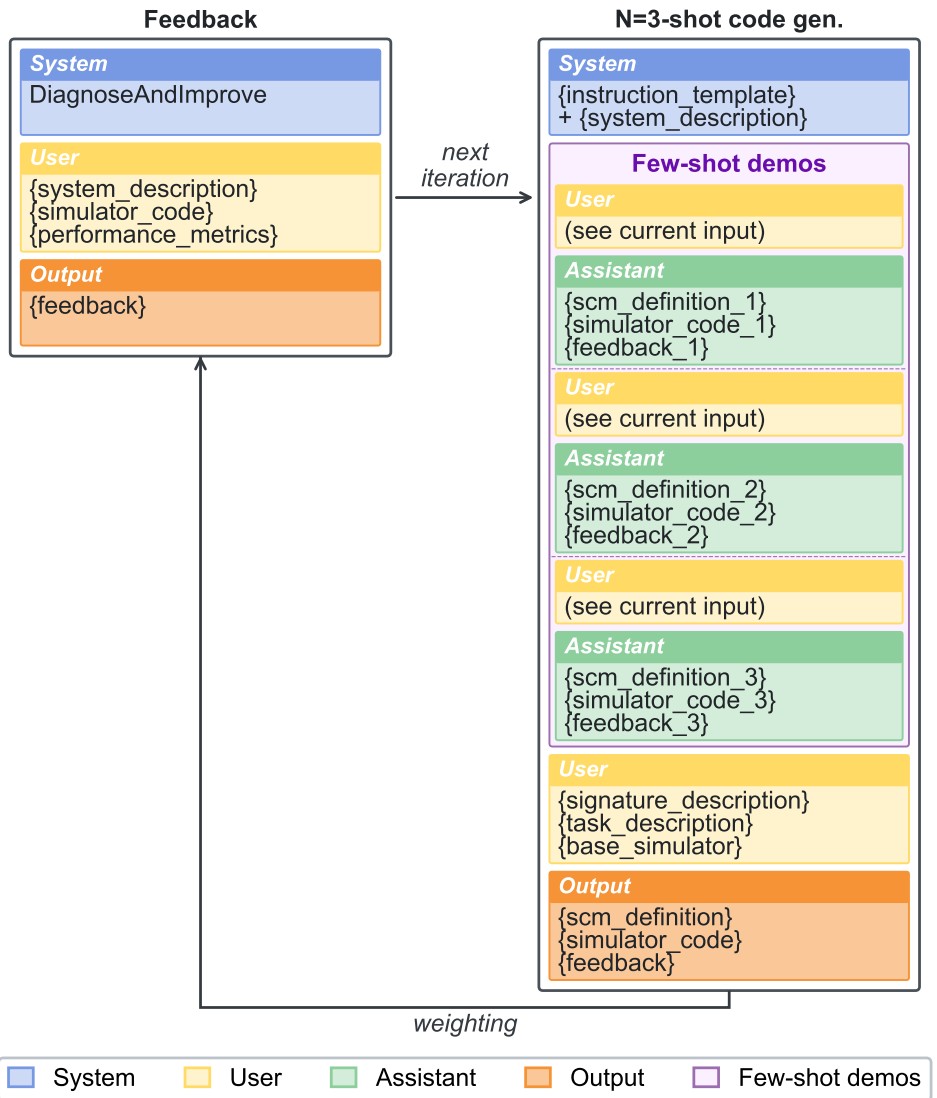

*Figure D-1.* **End-to-end prompting example for ModelSMC.** Based on the numeric performance metrics computed in the weighting step, the implementation of the model, general instructions, and task-specific information, we obtain LLM-based feedback about the model (**left**). Based on general instructions, task-specific information, few-shot examples consisting of the implementation of particles together with their feedback, the structural causal model, and the corresponding user prompt, we generate a new model implementation and the corresponding structural causal model in the propagation step. The propagation step also yields a text-based feedback field for consistency with the few-shot example structure, though this feedback is not used. To avoid duplicating static text, we do not include the signature description, task description, and base implementation in the few-shot example messages, but instead include references to the final user message (**right**).

### D.3. Parameter Estimation

ModelSMC does not require the value of the external parameters $\theta$ for the model discovery process. Since many downstream applications require parameter estimation, we estimate the parameter values $\hat{\theta}$ that best explain the observed data. For an observation $x_o$ and associated context $c_o$, we estimate $\hat{\theta}_o$ by maximizing the posterior $p(\theta|x, c, m)$ with respect to $\theta$ as

$$\hat{\theta}_o = \arg\max_{\theta \in \Theta} \log p\left(\theta \mid x_o, c_o, m\right). \tag{D-28}$$

To simplify notation, we drop the conditioning on the discovered model for the remainder of this section. The key ingredient for estimating $\hat{\theta}_o$ via Eq. D-28 is access to the posterior $p(\theta|x, c)$, which in practice is usually intractable. Therefore, we

use a surrogate model

$$p_\psi(\theta \mid x, c) \approx p(\theta \mid x, c). \tag{D-29}$$

One approach for obtaining $p_\psi$ is Neural Posterior Estimation (Lueckmann et al., 2017; Papamakarios & Murray, 2016, NPE) which trains a density estimator $p_\psi$ from scratch using a training set of $n_{\text{sim}}$ synthetic samples

$$\mathcal{D} = \{(\theta_j \sim p(\theta), c_j \sim p(c), x_j = m(c_j, \theta_j) \sim p(x \mid m(\theta, c))\}_{j=1}^{n_{\text{sim}}}. \tag{D-30}$$

Once $p_\psi$ is trained, the expression in Eq. D-28 is estimated by drawing $n_{\text{post}}$ samples from $p_\psi$ and selecting the sample that maximizes $p_\psi(\theta|x, c)$

$$\hat{\theta}_o = \arg\max_{\theta \in \Theta_o} \log p_\psi(\theta \mid x_o, c_o), \qquad \text{with} \quad \Theta_o = \{\theta_l \sim p_\psi(\theta \mid x_o, c_o)\}_{l=1}^{n_{\text{post}}}. \tag{D-31}$$

An alternative to NPE is NPE-PFN (Vetter et al., 2025). NPE-PFN employs TabPFN as a conditional density estimator. TabPFN (Hollmann et al., 2025) is a Prior-data Fitted Network (Müller et al., 2022) for tabular data, which provides one-dimensional density estimates and can be autoregressively extended for higher-dimensional densities. The key difference between NPE and NPE-PFN is that NPE-PFN does not train a new surrogate model but uses $\mathcal{D}$ (see, Eq. D-30) as an in-context dataset for the pretrained TabPFN model. Hence, NPE-PFN allows for training-free posterior estimation.

Following Vetter et al. (2025) and adapting to the case of additional external context $c_o$, the posterior can be decomposed into a product over the parameter dimensions as

$$p(\theta|x_o, c_o) \approx \prod_{j=1}^{d_\theta} q_\psi(\theta_j \mid \theta_{<j}, x_o, c_o, \mathcal{D}_{<j}). \tag{D-32}$$

For a $d_\theta$-dimensional parameter space, $\theta_j$ is the $j$-th component of $\theta$, and $\theta_{<j}$ the components of $\theta$ up to, but not including, the $j$-th. Analogously, $\mathcal{D}_{<j}$ is defined as $\mathcal{D}_{<j} = \{\theta_j^i, [\theta_{<j}^i, x^i, c^i] : (\theta^i, x^i, c^i) \in \mathcal{D}\}$. In the setting of NPE-PFN $q_\psi$ is a one-dimensional density estimate by TabPFN. To sample from the approximated posterior Eq. D-32, the individual dimensions of $\theta$ are sampled recursively.

The parameter estimation scheme Eq. D-28 used for NPE can also be applied with NPE-PFN. For high-dimensional parameter spaces, however, the recursive sampling and density estimation (Eq. D-32) for $n_{\text{post}}$ posterior samples can slow down the discovery process. To reduce the computational costs of the parameter estimation, we exploit NPE-PFNs factorization of the posterior Eq. D-32 and introduce a new strategy for estimating $\hat{\theta}_o$ which is based on temperature-scaling the posterior.

Temperature-scaling refers to exponentiating a density with an exponent of $\frac{1}{T}$, with the temperature $T > 0$. Applying this transformation to a density leaves the positions of the modes, i.e., the local maxima, unchanged. Furthermore, for $T \ll 1$ the temperature-scaled density becomes localized around the mode with the highest density value in the original density. Sampling from the temperature-scaled distribution accordingly produces samples that have a high density under the original distribution.

Sampling from the temperature-scaled posterior at $T \ll 1$ instead of the posterior itself can significantly reduce the necessary amount of posterior samples $n_{\text{post}}$ in Eq. D-31 as all samples are close to the posterior mode with the highest density value. Therefore, we temperature-scale the posterior to obtain

$$p(\theta|x_o, c_o)^{\frac{1}{T}} \approx \prod_{j=1}^{d_\theta} q_\psi(\theta_j \mid \theta_{<j}, x_o, c_o, \mathcal{D}_{<j})^{\frac{1}{T}}. \tag{D-33}$$

Crucially, Eq. D-33 only requires to power-scale the one-dimensional factors in the product rather than the full, potentially high-dimensional, posterior itself. In our experiments, we draw one sample from the temperature-scaled posterior and use it as $\hat{\theta}_o$. Drawing only one sample from the temperature-scaled posterior reduces the computational cost of the parameter estimation from $n_{\text{post}}$ sampling and density estimation steps to a single sampling step. Therefore, this parameter estimation approach exploits the factorization of the posterior in NPE-PFN and significantly reduces the computational costs of the parameter estimation.

### D.4. Estimation of Resampling Weights: Likelihood Estimation

Eq. 5 defines the resampling weight of a particle as the likelihood of the observed data $x_o$ under the model $m$, marginalized over the external parameters $\theta$. In practice, the likelihood $p(x|\theta, c, m)$ is intractable. Therefore, we use a surrogate model $p_\phi(x|\theta, c, m) \approx p(x|\theta, c, m)$ in the computation of Eq. 5. For simplicity, we drop the conditioning on $m$ for the remainder of this section. Throughout, Monte Carlo samples are drawn iid unless otherwise noted.

One approach to learn a surrogate model is to apply Neural Likelihood Estimation (Papamakarios et al., 2019, NLE). NLE uses the training set $\mathcal{D}$ (Eq. D-30) to train the surrogate model $p_\phi(x|\theta, c, m)$ via maximum likelihood training. An alternative to NLE is to adapt NPE-PFN (Vetter et al., 2025) to likelihood estimation. We denote this approach as NLE-PFN. Compared to Eq. D-32, NLE-PFN effectively interchanges $\theta$ and $x$, yielding

$$p(x|\theta, c) \approx \prod_{\ell=1}^{d_x} q_\psi\left(x_\ell \mid x_{<\ell}, \theta, c, \mathcal{B}_{<\ell}\right), \tag{D-34}$$

where $d_x$ denotes the dimensionality of the observation space, $x_\ell$ the $\ell$-th component of the observation, and $x_{<\ell}$ the components of $x$ up to, but not including, the $\ell$-th. The in-context dataset $\mathcal{B}$ is defined as $\mathcal{B}_{<j} = \left\{x_\ell^i, \left[x_{<\ell}^i, \theta^i, c^i\right] : \left(\theta^i, x^i, c^i\right) \in \mathcal{D}\right\}$. As in NPE-PFN, each factor in Eq. D-34 is estimated by a one-dimensional density estimate from TabPFN (Hollmann et al., 2025).

With the likelihood surrogate model $p_\phi(x \mid \theta, c, m)$, Eq. 5 can be estimated by the Monte Carlo estimate

$$p(x_o^j \mid m_k^i) \approx \frac{1}{B} \sum_{b=1}^{B} p(x_o^j \mid m_k^i, \theta_b, c_o^j), \qquad \theta_b \stackrel{\text{iid}}{\sim} p(\theta), \qquad \forall b = 1, ..., B. \tag{D-35}$$

If a (neural) posterior estimate $p_\psi(\theta \mid x)$ is available, $p(x_o^j \mid m_k^i)$ can also be approximated by using standard importance weighting and samples drawn from the posterior distribution rather than from the prior

$$p(x_o^j \mid m_k^i) \approx \frac{1}{B} \sum_{b=1}^{B} \frac{p_\psi\left(\theta_b \mid x_o^j\right)}{p(\theta_b)} p(x_o^j \mid m_k^i, \theta_b, c_o^j), \qquad \theta_b \stackrel{\text{iid}}{\sim} p_\psi\left(\theta \mid x_o^j\right), \qquad \forall b = 1, ..., B. \tag{D-36}$$

We find that for small $B$, Eq. D-36 has a lower variance than Eq. D-35. This reduced variance however, comes at the cost of requiring a pretrained NPE and additional density evaluations to compute the importance weights for each parameter sample.

For high-dimensional observations, we can use summary statistics $s : X \to \mathbb{R}^d$ with $d \ll d_x$ (Lueckmann et al., 2017). Assuming that $s(x)$ retains most of the relevant information about $x$, the approximation of the likelihood changes to

$$p(x|\theta, c) \approx p_\phi(s(x)|\theta, c). \tag{D-37}$$

Evaluating Eq. D-35 for each observed instance $(x_o^j, c_o^j) \in \boldsymbol{x}_o$ allows us to compute the unnormalized resampling weight $\tilde{w}_k^i$ (Eq. 6) of a model $m_k^i$. In practice, we use the slightly adapted definition Eq. D-38 of $\tilde{w}_k^i$ of the form

$$\tilde{w}_k^i \propto \exp\left(\frac{1}{\tau} \log\left(\prod_{j=1}^{M} p(x_o^j \mid m_k^i)\right)\right). \tag{D-38}$$

The temperature parameter $\tau$ in Eq. D-38 controls the bias of the resampling step. For $\tau \ll 1$, particles explaining the data well are favored, whereas for $\tau \gg 1$, the selection becomes more uniform across all particles. Unless stated otherwise, the temperature is set to $\tau = 1$ in all experiments. Importantly, the temperatures $T$ and $\tau$ used in Eq. D-33 and Eq. D-38 are two distinct hyperparameters.

## E. Baseline Methods

### E.1. FunSearch+

FunSearch (Romera-Paredes et al., 2024) is an LLM-based evolutionary method to evolve function implementations to improve their performance with respect to an evaluation function. The program database used by FunSearch to store

previously generated functions consists of several populations of functions, referred to as islands. Periodically, the worst-performing islands are discarded and reinitialized with a function from the remaining well-performing islands. When generating the prompt to synthesize a new function, FunSearch randomly selects an island and samples $k$ functions from this island with sampling probabilities based on their score and length. The selected functions are sorted by their scores and inserted into the program skeleton to provide ranked examples to the LLM. Based on this prompt, the LLM is tasked to implement a new function with the goal of generating a higher-scoring function. Newly generated functions are added to the same island from which the functions in the prompt were selected. Therefore, the islands evolve independently between reinitialization steps. A key difference between ModelSMC and FunSearch is that FunSearch selects the functions in the prompt based on the score but does not consider ancestral relations.

As originally proposed for problems from extremal combinatorics, FunSearch does not allow the use of external model parameters $\theta$. We therefore include the ability to use and estimate model parameters in FunSearch and denote this extended version as FunSearch+. Including external model parameters into FunSearch is conceptually inspired by Castro et al. (2025). Castro et al. (2025) applies gradient descent-based optimization on a likelihood-based criterion to estimate the values of the model parameters. The score of a discovered model is given by the validation performance of this criterion, given the optimized parameters. Instead of using gradient-based parameter optimization, we use the parameter estimation approach based on NPE and NPE-PFN (details in App. D.3). The advantage of using NPE and NPE-PFN to estimate $\hat{\theta}$ over gradient-based approaches is to obtain posterior estimates for the parameters without requiring differentiable models. Inspired by the use of a likelihood-based score in Castro et al. (2025), we also use a likelihood-based score in FunSearch+, which we define as

$$s = \log \prod_{j=1}^{M} p_\phi(x_o^j \mid m, c_o^j, \hat{\theta}^j). \tag{E-39}$$

In Eq. E-39, $\hat{\theta}^j$ denotes the parameter estimate for the observed instance $(x_o^j, c_o^j) \in \boldsymbol{x}_o$ and $p_\phi(x|m,c,\theta)$ is a surrogate likelihood (described in App. D.4)for the unknown true likelihood $p(x|m,c,\theta)$.

We base our implementation of FunSearch+ on the openly available FunSearch implementation provided by Zhang (2024). Our main modifications include evolving class methods rather than standalone functions and including our parameter estimation (App. D.3) and scoring schemes (Eq. E-39). Zhang (2024) relies on string-based modifications of functions and is therefore tightly coupled to Python code. As a result, unlike ModelSMC, it cannot be readily extended to other programming languages.

### E.2. ModelSMC $N$=1

This baseline method represents alternative LLM-based approaches for model discovery that explore model space along a single trajectory (Holt et al., 2025; Ma et al., 2024; Li et al., 2024; Shojaee et al., 2025). ModelSMC $N$=1 uses the same implementation as ModelSMC, but employs a population size of one and always propagates the most recently generated particle (i.e. setting $\alpha = 0$ in Eq. 4).

## F. Experimental Details

Here, we provide details on the exact configurations used in the experiments. For a complete list of all settings and details on the implementation, we refer to the accompanying code.

### F.1. Validation of ModelSMC for Model Discovery

The experiment presented in Sec. 4.1 tests the probabilistic inference mechanism underlying ModelSMC without using LLMs to synthesize new models. To implement this, we restrict ModelSMC to select models from a finite model space consisting of 20 predefined candidate models with observation dimension $d_x = 10$.

#### F.1.1. CANDIDATE MODELS

The candidate models used in this experiment are Gaussian Mixture Models (GMMs). Given the number of mixture components $n_{\text{comp}} \in \mathbb{N}$, a set of covariance matrices $\left\{\Sigma^i \in \mathbb{R}^{d_x \times d_x}\right\}_{i=1}^{n_{\text{comp}}}$, mean vectors $\left\{\mu^i \in \mathbb{R}^{d_x}\right\}_{i=1}^{n_{\text{comp}}}$, and weights

$\left\{w^i \in \mathbb{R}_+\right\}_{i=1}^{n_{\text{comp}}}$ with $\sum_i w^i = 1$, we define a GMM as

$$p_{\text{GMM}}(x \mid s, u) = \sum_{i=1}^{n_{\text{comp}}} w^i \cdot \frac{1}{(2\pi)^{d_x/2} \det\left(s^2 \Sigma^i\right)^{1/2}} \exp\left(-\frac{1}{2s^2}\left(x - \left(s\mu^i + u\right)\right)^T \left(\Sigma^i\right)^{-1} \left(x - \left(s\mu^i + u\right)\right)\right). \quad \text{(F-40)}$$

Here, $s$ and $u$ are the external parameters of the model: the scale parameter $s$ is scalar and follows a uniform prior over the interval $[0.1, 2]$. The shift parameter $u$ is a $d_x$-dimensional vector, where all but four components are set to zero. Each of the four non-zero components follows a uniform prior over the interval $[-2, 2]$. The non-zero components are sampled uniformly at random when initializing the candidate model and remain unaltered over the discovery run. The external parameters allow to scale and to shift the GMM along selected components to fit it to the observed data. For each candidate model, the configuration of the underlying GMM is randomly initialized as

$$n_{\text{comp}} \sim \text{Unif}\left(\{1, 2, ..., 10\}\right), \quad \text{(F-41)}$$

$$\tilde{w}^i \sim \text{Unif}\left([0, 1]\right), \qquad w^i = \frac{\tilde{w}^i}{\sum_{j=1}^{n_{\text{comp}}} \tilde{w}^j}, \qquad \forall i = 1, ..., n_{\text{comp}}, \quad \text{(F-42)}$$

$$\mu_j^i \sim \text{Unif}\left([-5, 5]\right), \qquad \forall j = 1, ..., d_x, \qquad \forall i = 1, ..., n_{\text{comp}}, \quad \text{(F-43)}$$

$$\tilde{\Sigma}_{jk}^i \sim \text{Unif}\left([-2, 2]\right), \qquad \forall j, k = 1, ..., d_x, \qquad \forall i = 1, ..., n_{\text{comp}}, \quad \text{(F-44)}$$

$$\Sigma^i = \alpha^i \cdot \tilde{\Sigma}^i \left(\tilde{\Sigma}^i\right)^T, \qquad \alpha^i \sim \text{Unif}\left([1, 2]\right), \qquad \forall i = 1, ..., n_{\text{comp}}. \quad \text{(F-45)}$$

In each discovery run, one of the 20 candidate models is selected as the ground truth model $m^*$, and observed data is drawn from $m^*$ with $s = 1$ and $u = \mathbf{0}_{d_x}$, i.e., no scaling and shifting is applied.

### F.1.2. Experimental configuration

Five different target models are selected from the set of 20 candidate models. For each target model, ten discovery runs are conducted, each with a different random seed. The set of candidate models is the same for all discovery runs.

ModelSMC's resampling weights are estimated using NLE (Papamakarios et al., 2019) with a training set $\mathcal{D}$ of size 5,000 (App. D.4). No additional context is used, i.e., a training instance consists of parameters $\theta = (s, u)$ drawn from the prior and a simulation $m(\theta)$. The observed set $\boldsymbol{x}_o$ is of size 1,000 and is drawn from the target model. To compute the Monte Carlo estimate Eq. D-35, $B = 500$ prior samples are used. We employ sbi's (Boelts et al., 2025) default hyperparameters for NLE but limit the maximum number of training epochs to 500.

All discovery runs proceed for 20 iterations and the probability for copying a particle in the propagation step is $\alpha = 0.8$ (Eq. 4). The LLM-based propagation kernel component is replaced by uniformly sampling a random model from the candidate set. The population contains $N = 100$ particles. At initialization of the population, the resampling weights are computed for all 20 candidate models. The population is then initialized by copying each model five times. In addition to not using an LLM in the propagation step, no text-based performance feedback (App. D.2) is generated.

## F.2. SIR Model

We apply ModelSMC to a synthetic dataset where the ground truth simulator model is known. In particular, we use a simple epidemiological SIR model (Kermack & McKendrick, 1927). The SIR model considers three populations—Susceptible, Infected, Recovered—with interactions controlled by two external parameters, a base transmission rate $\beta$ and a recovery probability $\gamma$. In this experiment, we use a time-discretized adaptation of the classical SIR model from Holt et al. (2025).

### F.2.1. Dataset

We use the implementation of the ground truth model and prior distributions of Holt et al. (2025) to generate observed data. Synthetic data is generated by drawing the initial numbers of susceptible, infected, and recovered individuals from the prior distributions specified as

$$S_0 \sim \text{Unif}\left(\{50, 51, ..., 99, 100\}\right), \qquad I_0 \sim \text{Unif}\left(\{1, 2, ..., 4, 5\}\right), \qquad R_0 = 0. \quad \text{(F-46)}$$

These initial population sizes define the context $c = (S_0, I_0, R_0)$. Given the external parameters $\beta$ and $\gamma$, the population sizes at time step $t \in \{1, ..., T\}$ are computed recursively from those at time $t - 1$. Let $N = S_{t-1} + I_{t-1} + R_{t-1}$ and

$p_{\text{inf}} = 1.0 - \exp(-\beta \cdot I_{t-1}/N)$. New infections and recoveries are drawn as

$$N_{\text{inf}} \sim \text{Binomial}\left(S_{t-1}, p_{\text{inf}}\right), \qquad N_{\text{rec}} \sim \text{Binomial}\left(I_{t-1}, \gamma\right), \tag{F-47}$$

and the state is updated via

$$S_t = S_{t-1} - N_{\text{inf}}, \qquad I_t = I_{t-1} + N_{\text{inf}} - N_{\text{rec}}, \qquad R_t = R_{t-1} + N_{\text{rec}}. \tag{F-48}$$

Because infections and recoveries are sampled from Binomial distributions, the transitions across time steps are stochastic.

In total, the simulation runs for $T = 60$ time steps, yielding three time series of 61 time steps. The prior distributions for the external parameters are given by

$$\gamma \sim \text{Unif}\left([0, 1]\right), \qquad \beta \sim \text{Unif}\left([0, 2]\right). \tag{F-49}$$

In our experiments, these priors are fixed for all generated models and are not treated as part of the model. In contrast to Holt et al. (2025), we do not use an LLM to synthesize the dimensionality of the parameter space and the bounds of the uniform priors. Following their NPE-based parameter estimation scheme, a single training trajectory drawn from the ground truth model is used to estimate $\beta$ and $\gamma$. For computations of evaluation metrics, including the resampling weights, we use $x_o$ containing 100 samples drawn from the same generative process.

### F.2.2. EXPERIMENTAL CONFIGURATION

Parameter estimation for $\beta$ and $\gamma$ is performed using NPE with a training set of 1,000 synthetic instances (App. D.3). Each synthetic training sample consists of a context $c = (S_0, I_0, R_0)$ drawn from the initial condition priors, parameters $\theta = (\gamma, \beta)$ drawn from their uniform priors, and an observation $x = m(\theta, c)$. The observation is a 183-dimensional vector formed by concatenating the simulated time series of susceptible, infected, and recovered populations.

After training, parameter values are inferred using $10^4$ samples drawn from the approximate posterior $p_\psi(\theta|x_o, c_o, m)$ using maximum log-posterior (MAP) criterion (Eq. D-31). Here, $x_o$ denotes the single observed trajectory generated from the ground-truth data-generating model and $c_o$ the corresponding initial condition. The resulting parameter estimate for $\beta$ and $\gamma$ is fixed and reused for all downstream computations. No summary statistics are applied to the observations.

This estimation procedure is inspired by Holt et al. (2025) but differs in multiple aspects: In our implementation, the initial condition is explicitly treated as a condition in NPE. Moreover, we normalize the observations and the initial conditions elementwise by the total population size before passing them to the density estimator. Finally, we use a MAP estimate based on posterior samples, instead of averaged posterior samples, which can lead to poor parameter estimates for multimodal posteriors.

Resampling weights for ModelSMC are computed using NLE (App. D.4). The training set for NLE is identical to the normalized training set used for NPE in the parameter estimation step. We approximate the marginal likelihood (Eq. D-35) using 5,000 prior samples. ModelSMC runs for up to 15 iterations. In the propagation kernel (Eq. 4), existing particles are copied unaltered with a probability of $\alpha = 0.8$. The particle population size is $N=50$. Therefore, in expectation $N \cdot (1 - \alpha) = 10$ new particles are generated per iteration.

The prompts in $c_{\text{task}}$ (App. D.2) are constructed by adapting the prompts used in Holt et al. (2025) to match the specifics of ModelSMC. We follow the same procedure for the program skeleton. Components predicting prior bounds are removed, and the initial simulator model is restricted to return constant population sizes.

For ModelSMC $N=1$ (App. E.2), 150 models are generated in total, matching the expected number generated by ModelSMC across iterations. FunSearch+ (App. E.1) uses the same evaluation pipeline as the other two approaches. Since we find that FunSearch+ requires less LLM tokens per generated model compared to ModelSMC and ModelSMC $N=1$, we increase the total number of generated models for FunSearch+ to 190 in order to match the token budget of the other methods. All methods provide the LLM with up to three previously generated implementations (i.e., ancestors). To compute the results in Table 1, all methods are run with ten different initial random seeds. The two baseline methods use the same parameter estimation scheme as ModelSMC. For implementation details, we refer to the accompanying code.

### F.3. Pharmacological Kidney Model

#### F.3.1. DATASET

The dataset is originally from Dluhy et al. (1972); we use the version provided with the manuscript from Maddah & Hallow (2022). The data contains measured plasma potassium and aldosterone responses to a standardized potassium chloride infusion in healthy humans under controlled dietary conditions. Subjects were assigned to four diet groups defined by low/high potassium (40 vs. 200 mEq/day) and low/high sodium (10 vs. 200 mEq/day) intake for 6–10 days, then received potassium chloride at 0.62 mEq/min for 120 min followed by 180 min recovery. Plasma potassium and aldosterone were sampled at baseline, during infusion (0, 30, 90, 120 min), and after recovery (300 min). This constitutes a total of 40 unique data points. There are no summary statistics used.

As context $c$ is passed to the model, we give the initial condition of the model. The initial condition is derived from steady-state conditions with normal sodium and potassium intake. The model parameters $\theta$ are drawn from uniform priors. For details about the bounds of the prior, we refer to the accompanying code.

#### F.3.2. EXPERIMENTAL CONFIGURATION

ModelSMC is run for 15 iterations with cloning probability $\alpha$=0.8 (Eq. 4). The particle population contains $N$=50 particles. Parameter estimation is performed using NPE-PFN with temperature-scaled posteriors (App. D.3), using temperature $T = 10^{-5}$ and 2,000 simulator calls per evaluation. Resampling weights are estimated using NLE-PFN (App. D.4), and the marginal likelihood estimate (Eq. D-36) is approximated with $B = 500$ posterior samples. In Eq. D-36, NPE-PFN is used for importance weighting and sampling proposals. The in-context dataset used for NPE-PFN is constructed from the same simulations as the in-context dataset of NLE-PFN. In each propagation step, up to three previously discovered models are provided as few-shot context to the LLM (App. D.2).

For ModelSMC $N$=1, the same parameter and weight estimation settings are used. ModelSMC $N$=1 is run for 150 iterations. All experiments are performed using fixed, hand-crafted summary statistics and are evaluated using the negative log marginal likelihood estimated by NLE-PFN.

### F.4. Hodgkin-Huxley Model

#### F.4.1. DATASET

We consider data from the Allen cell types database (Allen Institute for Brain Science, 2015), which consists of 10 in-vitro recordings from a mouse cortex. This data was previously used for parameter inference (Gonçalves et al., 2020; Vetter et al., 2025). We use the same set of seven summary statistics as previously proposed in Gonçalves et al. (2020): Spike count, mean, and standard deviation of the resting potential, and the first four voltage moments, i.e., mean, standard deviation, skew and kurtosis.

The additional context $c$ passed to the model is a two-dimensional vector containing the initial membrane voltage $V_0$ and the input current $I$ to the cell. The prior over the conditions is defined by uniformly drawing the context values from the observed set $\boldsymbol{x}_o$. The two context dimensions are drawn independently, resulting in 100 possible combinations of initial voltage and input current. Model parameters $\theta$ are 10-dimensional vectors with components independently drawn from uniform priors, in the same way as provided in Gonçalves et al. (2020). For details about the bounds of the prior, we refer to the accompanying code.

#### F.4.2. EXPERIMENTAL CONFIGURATION

ModelSMC is run for 15 iterations with cloning probability $\alpha$=0.8 (Eq. 4). The particle population has size of $N$=50. For parameter estimation, we employ NPE-PFN-based parameter estimation utilizing temperature-scaling of the posterior (App. D.3). The temperature used for temperature-scaling (Eq. D-33) is $T = 10^{-5}$. NPE-PFN uses an in-context dataset $\mathcal{D}$ (Eq. D-30) of size 5,000. Resampling weights, i.e. likelihood estimates, are obtained using NLE-PFN (App. D.4), and Eq. D-36 is estimated with $B$ =500 posterior samples. In Eq. D-36, NPE-PFN is used for importance weighting and sampling proposals. NLE-PFN and NPE-PFN are applied to seven standard summary statistics (Gonçalves et al., 2020) rather than to the full voltage traces. In each propagation step, up to five ancestral samples are prompted to the LLM (App. D.2).

For ModelSMC $N$=1 (App. E.2), we use the same configuration for parameter and weight estimation as in ModelSMC. ModelSMC $N$=1 runs for 150 iterations. FunSearch+ (App. E.1) generates 180 models and uses up to three previously synthesized models to construct the prompts used for propagation. Per prompt, four new models are generated. The program database consists of ten islands, and reinitialization of the worst-performing islands is performed every four hours. To estimate the parameters of the generated models and to compute the score (Eq. E-39), NPE-PFN and NLE-PFN are used with the same settings as in ModelSMC and ModelSMC $N$=1. All three methods are run with ten different initial random seeds to obtain the results presented in Table 1.

### F.5. Posterior Analysis of Hodgkin-Huxley Experiment

To characterize posterior mass across all discovered models, we classify all $1,445$ usable particles (those with finite log-weight) from the Hodgkin-Huxley experiment (Sec. 4.4) using the three-step LLM-assisted pipeline described in Sec. 4.5.

**Step 1: Taxonomy construction.** A stratified sample of approximately 15% of all particles (across quality quartiles defined by weight) is used to extract free-form descriptions of the ion channels added beyond the three base Hodgkin-Huxley currents (fast $Na^+$, delayed-rectifier $K^+$, ohmic leak) via an LLM call. A second LLM call consolidates all descriptions into a two-level taxonomy: multiple families ($I_M$, $I_{NaP}$, $I_h$, $I_A$, and multi-channel combinations) with 11 structural subtypes, split by activation gate power, presence of an inactivation gate, and time-constant implementation (voltage-dependent, fixed scalar, log-scaled, or instantaneous).

**Step 2: Classification.** All $1,445$ particles are classified in batches via an LLM call per batch, yielding a `family_id` and `subtype_id` for each particle. The resulting family distribution is: $I_M$ = 1,084 (75.0%), $I_M$+$I_{NaP}$ = 216 (14.9%), $I_M$+$I_h$ = 80 (5.5%), $I_M$+$I_A$ = 35 (2.4%), base only = 10 (0.7%), unknown = 11 (0.8%).

**Step 3: Weight analysis.** The distribution of weights per ion-channel subtype is shown in Fig. 5. Note that we omit showing subtypes with less than 8 models belonging to that subtype, i.e. we result with 8 subtypes in Fig. 5. Figure F-2 shows the within-seed rank of each ion-channel family across all 10 independent seeds. $I_M$ ranks first in 9 out of 10 seeds, confirming that the preference for M-type channel extensions is a stable posterior conclusion and not an artefact of a single run.

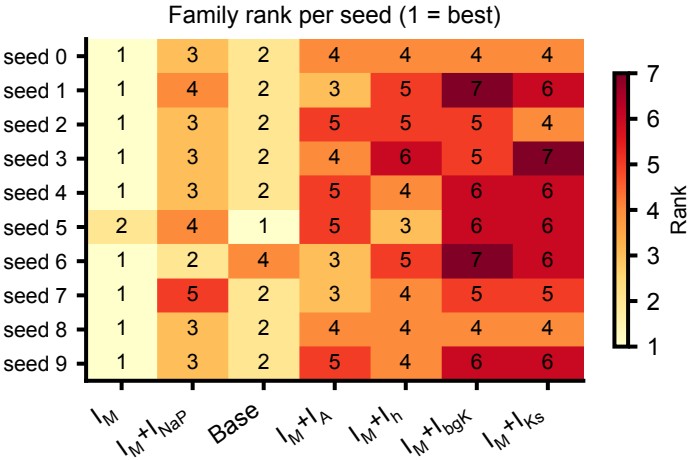

*Figure F-2.* **Cross-seed rank stability of ion-channel families.** Each row is one seed and each column is an ion-channel family; cell colour and number give the within-seed rank of that family (1 = best; darker = better).

### F.6. Ablation Study of Hodgkin-Huxley Experiment

We systematically vary six design axes of ModelSMC on the Hodgkin-Huxley task (Secs. 4.4 and 4.6). The default configuration uses Claude Sonnet 4.6, $N$=50 particles, $K$=15 iterations, 5,000 simulations for posterior and likelihood estimation, marginal-likelihood weights, the full prompt, and LLM-only narrative feedback. Each ablation modifies exactly one component. The default is run with ten independent seeds; all variants use five seeds. The total budget in new particles is $K \cdot (1-\alpha) \cdot N = 15 \cdot 0.2 \cdot 50 = 150$.

**Weighting scheme.** We replace the marginal-likelihood weights with MSE-based weights to test whether the principled Bayesian weighting is necessary for good performance of single models. Note that modifying the weighting removes the interpretatbility of the particles as valid posterior samples.

**Simulation budget.** We reduce the number of simulations used to train the likelihood estimator from 5,000 to $\{1,000, 500, 200\}$ to characterise the compute–quality trade-off.

**Prompt complexity.** We replace the full task description with a minimal prompt to assess how much prompt engineering contributes to discovery performance.

**Feedback format.** We vary how performance information is communicated to the LLM. I.e., LLM-based feedback text only (default), combined LLM feedback + numeric metrics, and numeric metrics only. This tests whether structured metric feedback helps or hurts proposal quality.

**Pool size and number of iterations.** We vary $N$ and $K$ while keeping the total budget of 150 new particles fixed: ($N$=150, $K$=5) favours population diversity, while ($N$=10, $K$=75) favours deep iterative refinement.

**LLM backbone.** We replace Claude Sonnet 4.6 with GPT-5 mini to verify that ModelSMC is not tied to a specific language model.

## G. Additional Experiments

### G.1. Validation of ModelSMC for Model Discovery

In Fig. 2, the median proportion of the target model in the population, i.e., the model from which the data $x_o$ was drawn, converges to an upper bound of $\approx 80\%$ and exhibits a distinct oscillatory behaviour. This oscillatory behaviour becomes even more prominent when evaluating the proportion of the target model in the population for a single run (Fig. G-3 b).

If the ESS (Eq. 2) is below $\tau_{\text{ESS}}$, systematic resampling is applied to rejuvenate the particle population (Sec. 3.3). For candidate models that differ from the target model, the resampling weights are vanishingly small (Fig. G-3 a). Therefore, the population is expected to contain only the target model after resampling with these weights. In the subsequent propagation step, each particle in the population is replaced by a randomly selected candidate model with a probability of $1-\alpha = 20\%$. Given that there are 20 candidate models, in expectation $81\%$ $((1-0.2)+0.2 \cdot \frac{1}{20} = 0.81)$ of the models in the population are the target model in this setting with 20 candidate models and choice of $\alpha$. This matches the observed upper bound of $\approx 80\%$ (Fig. 2).

The oscillatory behaviour can also be explained by the replacement of particles during the propagation step. Starting with $100\%$ target models after the systematic resampling step, the propagation step will in expectation yield $81\%$ target models as outlined in the previous paragraph. If no systematic resampling occurs in the next iteration, i.e., the effective sample size is still sufficiently large, the random replacement of particles during the propagation step will further reduce the proportion of the target model in the population. This reduction continues until the effective sample size is below the specified threshold, which will bring the proportion of the target model back to roughly $80\%$. The repeated interplay between the resampling and propagation steps produces the observed oscillatory behaviour in the proportion of the target model. This relation between effective sample size and proportion of the target model can clearly be observed (Fig. G-3 b,c).

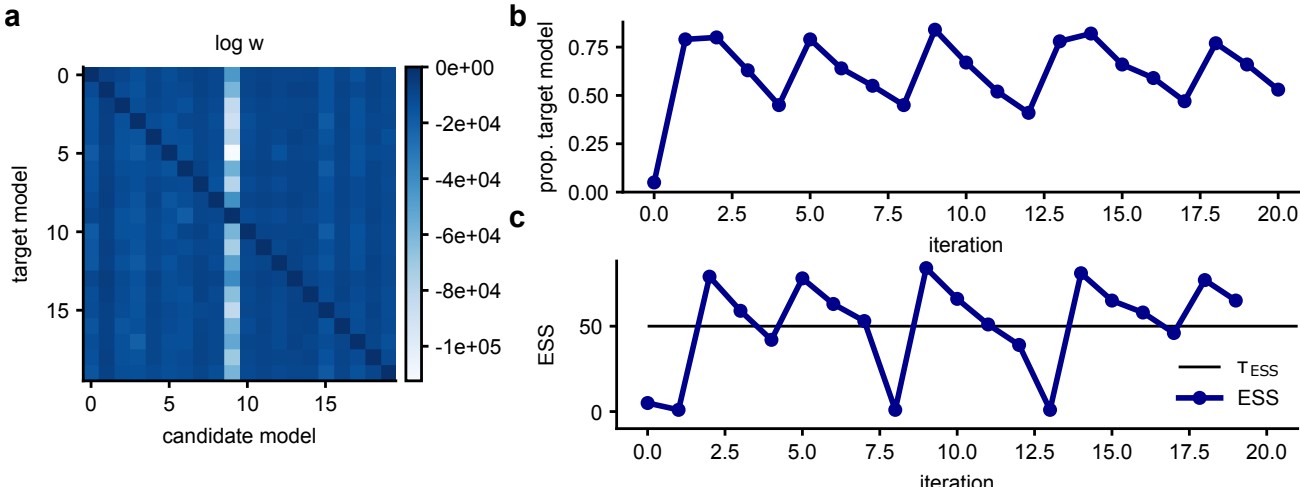

*Figure G-3.* **(a)** Logarithm of normalized resampling weights, averaged over five random seeds, for different combinations of target and candidate models for the LLM-free experiment (Sec. 4.1, App. F.1). Data $x_o$ is drawn from the target model, and weights are evaluated using the ground truth density (Eq. F-40) of the candidate model. **(b)** Proportion of the target model in the population for a selected run from the experiment presented in Fig. 2. **(c)** Effective sample size (Eq. 2) for the run presented in (b). The black horizontal line denotes the threshold $\tau_{\text{ESS}}$ for applying systematic resampling (Sec. 3.3).

## G.2. Extended Version of Table 1

*Table G-1.* **Extended version of Table 1**. Metrics are collected for the best-performing particle from each of the ten discovery runs. Best-performing particles are selected according to the metric which was applied to weight particles during the discovery process. We report the median metric together with a 90% confidence interval. The confidence interval is estimated using a percentile bootstrap: $10^4$ bootstrap samples of size ten are drawn with replacement from the metric values of the selected particles. The median is computed for each sample, and the 5% and 95% quantiles of the resulting distribution are reported.

|  | FunSearch+ | ModelSMC | ModelSMC $N = 1$ |
|---|---|---|---|
|  | $-\log p(x_o\|m)$ (↓) | | |
| SIR | -48794.12 [-51910.18, -48293.60] | -45834.78 [-50184.76, -38023.12] | -57028.36 [-59613.61, -47127.31] |
| HH | 254.58 [250.46, 259.52] | 256.00 [255.05, 260.50] | 243.27 [232.39, 258.08] |
| Kidney | N/A | 42.85 [41.54, 45.62] | 36.54 [34.95, 38.31] |

## G.3. LLM Token Usage

An evaluation criterion to assess the effectiveness of an LLM-based model discovery algorithm is how many tokens are passed to and generated by an LLM to generate a model with a given performance threshold on the observed data. To minimize computational costs, the number of tokens needed to reach a given performance threshold should be as small as possible. In this section, we examine how ModelSMC performs under this evaluation criterion compared to the baseline methods (Apps. E.1 and E.2).

The results in Fig. G-4 are based on the discovery runs evaluated in Sec. 4.2. For each discovery run we compute the running best metric value, i.e., the best value observed up to and including each iteration of the discovery run, and then take the median over all ten discovery runs per combination of task and metric. The 90% confidence intervals are computed by randomly selecting ten of the trajectories with replacement and computing the median of the running best values over the selection at each iteration. This is done 10,000 times for each combination of task and method. The 90% confidence interval is then defined by the 5% and 95% quantiles of the distribution of the bootstrap medians at each iteration. The position on the token axis corresponds to the median number of tokens used up to and including the current iteration, computed over all ten discovery runs.

Fig. G-4 shows that while ModelSMC and ModelSMC $N$=1 generate the same expected number of particles, ModelSMC $N$=1 consistently consumes more tokens across all three tasks. This is because ModelSMC $N$=1 always resamples the

most recently generated particle, so the ancestry chain passed to the LLM grows with each iteration until reaching its maximum length after 3 iterations. In ModelSMC, particles from early iterations can persist into later ones, keeping ancestry chains shorter and reducing expected input tokens. Compared to both ModelSMC variants, FunSearch+ requires significantly fewer tokens per particle. This is most clear on the Hodgkin-Huxley task, where the expected 150 ModelSMC particles consume approximately twice as many tokens as the 180 FunSearch+ particles. This is expected: ModelSMC and ModelSMC $N$=1 pass the task context $c_{\text{task}}$ and text-based performance evaluations (App. D.2) alongside the resampled model ancestry, whereas FunSearch+ only provides the code skeleton and sampled functions from the program database. ModelSMC and ModelSMC $N$=1 also generate more output tokens, as both produce text-based performance feedback that FunSearch+ does not.

The median curve for ModelSMC $N$=1 falls below both other methods for most of the token budget, suggesting higher token efficiency. However, the confidence intervals overlap across all three tasks, so no statistically significant difference can be established from ten discovery runs per method. Formulating model discovery as probabilistic inference therefore does not significantly affect token efficiency relative to the baselines.

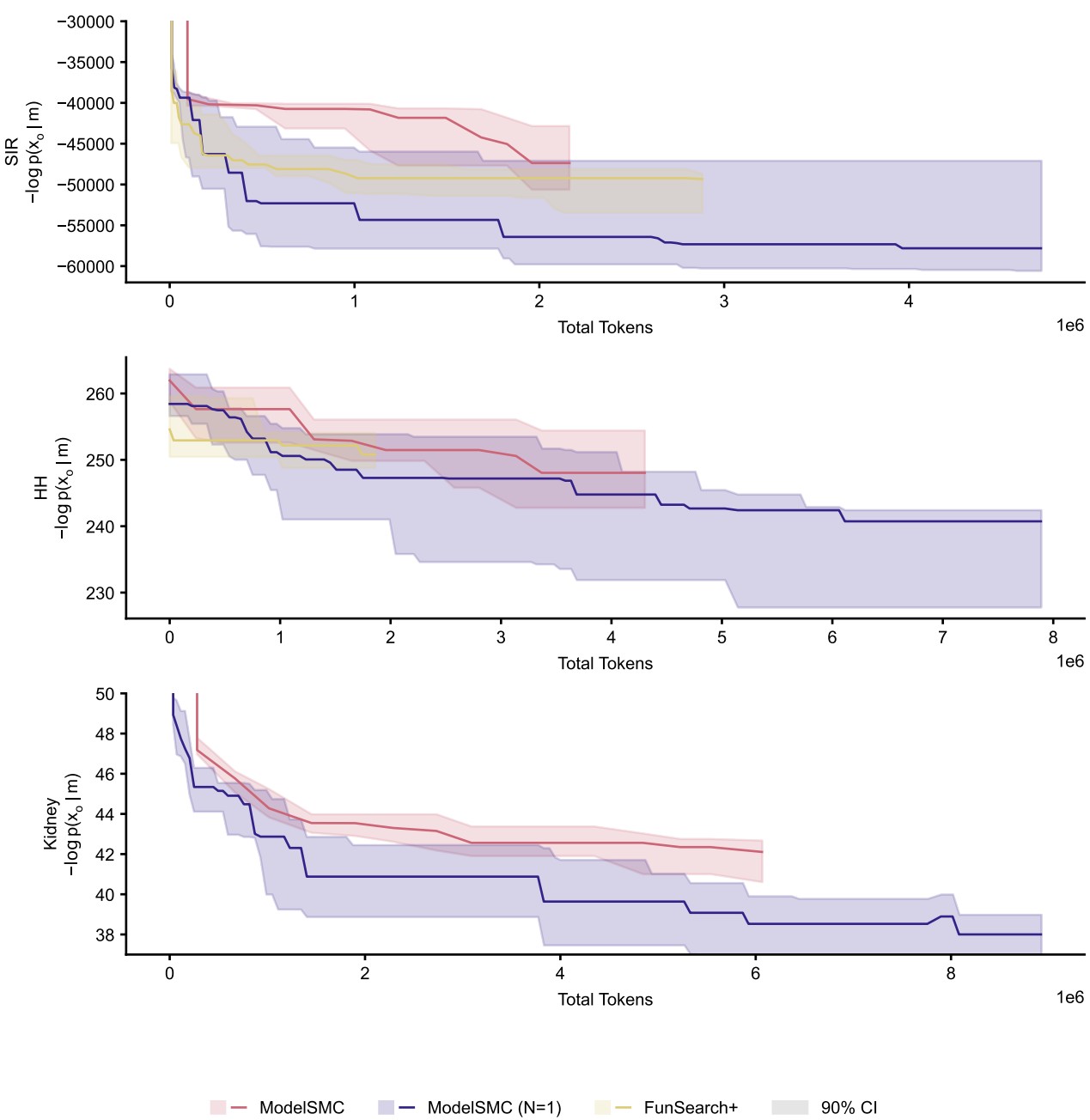

*Figure G-4.* **Performance of the discovered models as a function of token usage**. Red: ModelSMC. Blue: ModelSMC $N$=1. Yellow: FunSearch+. The depicted performance metrics and token counts are the median of the ten discovery runs conducted for the experiments discussed in Sec. 4.2. The depicted metric is the negative log of ModelSMC's resampling weight (Eq. 6). The 90% confidence intervals of the medians are defined by the 5% and 95% quantiles of the empirical distribution of 10,000 bootstrap replicates of the median. First row: Synthetic data from a SIR epidemiological model (App. F.2). Second row: Hodgkin-Huxley Model (Sec. 4.4 and App. F.4). Third row: Pharmacological kidney model (Sec. 4.3 and App. F.3).

# H. Prompt Examples

## H.1. Hodgkin-Huxley Experiment: LLM-History Propagation Step

The prompts passed to the LLM are assembled from different components (see App. D.2) including specified task information and instructions, implementations of ancestral particles, and their LLM-generated performance assessments (App. D.2). In this section, we provide a concrete example of the general interaction scheme visualized in Fig. D-1. Specifically, we exemplify the interactions observed for a particle during the propagation step in a discovery run for the Hodgkin-Huxley experiment (Sec. 4.4, App. F.4). In the following visualization, each text box corresponds to a single message sent to or received from the LLM. The raw messages were reformatted for readability, with malformatted and LaTeX-incompatible characters replaced throughout. The messages sent to the LLM are presented in section 'Prompt', followed by the LLM's response in section 'Response'.

### H.1.1. PROMPT

> **System**
>
> Your input fields are:
> 1. 'signature_description' (str): Simulator I/O description
> 2. 'task_description' (str): Task considerations
> 3. 'base_simulator' (str): Base code structure
> Your output fields are:
> 1. 'scm_definition' (str): Compact SCM summary as a bullet list (max 5 lines): key variables, their causal dependencies, and core assumptions. No equations or verbose descriptions.
> 2. 'simulator_code' (str): Improved mechanistic simulator with substantial structural differences from prior examples.
> 3. 'feedback' (Union[str, NoneType]): Optional feedback. Only generate if requested.
> All interactions will be structured in the following way, with the appropriate values filled in.
>
> [[ ## signature_description ## ]]
> {signature_description}
>
> [[ ## task_description ## ]]
> {task_description}
>
> [[ ## base_simulator ## ]]
> {base_simulator}
>
> [[ ## scm_definition ## ]]
> {scm_definition}
>
> [[ ## simulator_code ## ]]
> {simulator_code}
>
> [[ ## feedback ## ]]
> {feedback} # note: the value you produce must adhere to the JSON schema: {"anyOf": [{"type": "string"}, {"type": "null"}]}
>
> [[ ## completed ## ]]
> In adhering to this structure, your objective is:
> Given the fields 'signature_description', 'task_description', 'base_simulator', produce the fields 'scm_definition', 'simulator_code', 'feedback'.
>
> You are an expert in mechanistic modeling with differential equations and simulator optimization.
>
> Core Requirements:
> - Create a scientific simulator that captures system dynamics accurately
> - Write fully functional, executable code with no placeholders
> - Use tabs for indentation and mathematically correct formulations
> - Add variable shapes in every line of code
> - Achieve optimal validation metrics
>
> Implementation Guidelines:
> - Start with mechanistic components, add stochastic elements where needed
> - Follow physical/biological constraints and include prior knowledge
> - Use any operations (log, exp, power, etc.) and non-differentiable operations
> - Ensure computational efficiency for many simulations
> - Keep the code as simple as possible

Iterative Optimization Rules:
- You are part of an iterative process - make only 1-2 targeted improvements per iteration
- Never copy code verbatim - preserve working components while making precise, localized changes
- Make creative but sensible and simple modifications that are structurally different from prior examples
- Focus on changes that build incrementally toward better validation metrics
- Ensure executable code without syntax or shape errors

System description:
# HODGKIN-HUXLEY NEURON SIMULATOR WITH MISSING ION CHANNELS

## OBJECTIVE
Extend an existing Hodgkin-Huxley neuron simulator by identifying and implementing
missing ion channels necessary to reduce discrepancies between simulations and
experimental voltage recordings. The goal is to improve agreement across multiple
electrophysiological summary statistics while maintaining model parsimony.

## BASE MODEL
Classic Hodgkin-Huxley formulation with standard three-gating-variable structure:
- **Leak current**: conductance g_leak, reversal potential E_leak
- **Sodium ($Na^+$) current**: conductance gNa, gating variables m (activation) and h (inactivation), reversal E_Na
- **Potassium ($K^+$) current**: conductance gK, gating variable n (activation), reversal E_K
- **Dynamics**: First-order gating kinetics with voltage-dependent time constants and steady states
- **Input**: Externally applied current I_inj(t) with optional stochastic components
- **Stochasticity**: Optional random seed for reproducible simulations

## EXTENSIBILITY
Two pre-allocated channel slots (X1, X2) are available for additional mechanisms.
Each slot provides two tunable parameters (see signature_description for details).
Channels beyond these two slots must use fixed parameters, not inferred ones.

## EVALUATION METRICS
Model performance assessed on held-out voltage recordings via these summary statistics:
1. Number of spikes during stimulation
2. Mean resting potential (pre-stimulation)
3. Standard deviation of resting potential
4. Mean voltage during stimulation
5. Variance of voltage
6. Skewness of voltage
7. Kurtosis of voltage

## DATA CONTEXT
Intracellular voltage recordings from a single neuron exhibiting tonic spiking
behavior across different stimulus intensities.

**Important data characteristics**:
- No burst firing patterns (no clusters of rapid spikes separated by quiescence)
- No prolonged or sustained high-frequency firing (no sustained high-frequency firing)
- Regular, tonic spiking activity during periods of input current stimulation
- Non-spiking (quiescent) behavior in the absence of input current
- Evenly-spaced action potentials without burst patterns during the activation period

Therefore, avoid adding channels specifically designed to produce bursting or sustained
high-frequency dynamics.

**User**

This is an example of the task, though some input or output fields are not supplied.

[[ ## signature_description ## ]]
(see current input)

[[ ## task_description ## ]]
(see current input)

[[ ## base_simulator ## ]]
(see current input)

**Assistant**

[[ ## scm_definition ## ]]
None

[[ ## simulator_code ## ]]

```python
import torch
import torch.nn as nn

class DiscoveredSimulator(nn.Module):
  def __init__(self):
    super(DiscoveredSimulator, self).__init__()
    return

  def forward(
    self,
    init_voltage: float,
    input_current: torch.Tensor,
    dt: float,
    t: torch.Tensor,
    params: torch.Tensor,
    seed=None,
  ):
    """
    Simulates a Hodgkin-Huxley neuron for a specified time duration.

    Args:
      init_voltage: torch.Tensor: (batch_size,) # initial voltage
      input_current: torch.Tensor: (batch_size, time_steps) # input current
      dt: float # time step size
      t: torch.Tensor: (time_steps,) # time array
      params: torch.Tensor: (batch_size, n_params) # parameters
      seed: optional random seed

    Returns:
      V: torch.Tensor: (batch_size, time_steps) # voltage traces
    """
    device = params.device

    # Set up random generator
    if seed is not None:
      generator = torch.Generator(device=device)
      generator.manual_seed(seed)
    else:
      generator = torch.Generator(device=device)

    batch_size = params.shape[0]
    time_steps = t.shape[0]

    # Extract parameters
    gbar_Na = params[:, 0].float()   # mS/cm2
    gbar_K = params[:, 1].float() # mS/cm2
    g_leak = params[:, 2].float() # mS/cm2
    E_leak = -params[:, 3].float() # mV
    Vt = -params[:, 4].float() # mV
    nois_fact = params[:, 5].float() # unitless
    # TWO POSSIBLE ADDITIONAL CHANNELS (X1, X2)
    # Each channel has one tunable parameter: conductance gbar_Xi
    # Then there are two additional parameters available: param_i and param_j.
    # ONLY ADD ONE CHANNEL IF NECESSARY. Keep the model as simple as possible.
```

```python
56      gbar_X1 = params[:, 6].float() # mS/cm2 # you can rename X1 to anything you want # in range [1e-4,
    ↪ 10]
57      gbar_X2 = params[:, 7].float() # mS/cm2 # you can rename X2 to anything you want # in range [1e-4,
    ↪ 120]
58      param_i = -params[:, 8].float() # (param are positive values in range [1e-4, 150])
59      param_j = -params[:, 9].float() # (param are positive values in range [1e-4, 3000])
60
61      tstep = float(dt)
62
63      # Parameters
64      nois_fact_obs = 0.0
65      C = 1.0  # uF/cm^2
66      E_Na = 53.0 # mV
67      E_K = -107.0
68
69      ####################################
70      # kinetics
71      def Exp(z):
72        return torch.where(z < -5e2, torch.exp(torch.full_like(z, -5e2)), torch.exp(z))
73
74      def efun(z):
75        return torch.where(torch.abs(z) < 1e-4, 1 - z / 2, z / (Exp(z) - 1))
76
77      # Channel kinetics
78      def alpha_m(x):
79        v1 = x - Vt - 13.0
80        return 0.32 * efun(-0.25 * v1) / 0.25
81
82      def beta_m(x):
83        v1 = x - Vt - 40
84        return 0.28 * efun(0.2 * v1) / 0.2
85
86      def alpha_h(x):
87        v1 = x - Vt - 17.0
88        return 0.128 * Exp(-v1 / 18.0)
89
90      def beta_h(x):
91        v1 = x - Vt - 40.0
92        return 4.0 / (1 + Exp(-0.2 * v1))
93
94      def alpha_n(x):
95        v1 = x - Vt - 15.0
96        return 0.032 * efun(-0.2 * v1) / 0.2
97
98      def beta_n(x):
99        v1 = x - Vt - 10.0
100       return 0.5 * Exp(-v1 / 40)
101
102     def tau_x(alpha, beta):
103       return 1.0 / (alpha + beta)
104
105     def inf_x(alpha, beta):
106       return alpha / (alpha + beta)
107
108     # ===== BEGIN EDITABLE SECTION (only modify within this block) =====
109     # TODO: add the missing kinetics equations for the Hodgkin-Huxley neuron similar to the ones above;
    ↪ ONLY ADD IF NECESSARY
110     # ===== END EDITABLE SECTION =====
111
112     ####################################
113
114     # simulation from initial point
115     V = torch.zeros((batch_size, time_steps), device=device)  # baseline voltage
116     n = torch.zeros((batch_size, time_steps), device=device)
117     m = torch.zeros((batch_size, time_steps), device=device)
118     h = torch.zeros((batch_size, time_steps), device=device)
119     # ===== BEGIN EDITABLE SECTION (only modify within this block) =====
120     # TODO: add the missing state variables for the Hodgkin-Huxley neuron similar to the ones above;
    ↪ ONLY ADD IF NECESSARY
121     # ===== END EDITABLE SECTION =====
122
123     # Initialization
124     V_init = init_voltage.to(device)
125     V[:, 0] = V_init
126     n[:, 0] = inf_x(alpha_n(V[:, 0]), beta_n(V[:, 0]))
127     m[:, 0] = inf_x(alpha_m(V[:, 0]), beta_m(V[:, 0]))
128     h[:, 0] = inf_x(alpha_h(V[:, 0]), beta_h(V[:, 0]))
129     # ===== BEGIN EDITABLE SECTION (only modify within this block) =====
```

```
130       # TODO: add the missing state variable initialization for the Hodgkin-Huxley neuron similar to the
      ↪ ones above; ONLY ADD IF NECESSARY
131       # ===== END EDITABLE SECTION =====
132
133       # Simulation loop
134       for i in range(1, time_steps):
135           # All operations now work on batched tensors (batch_size,)
136           a_m, b_m = alpha_m(V[:, i - 1]), beta_m(V[:, i - 1])
137           a_h, b_h = alpha_h(V[:, i - 1]), beta_h(V[:, i - 1])
138           a_n, b_n = alpha_n(V[:, i - 1]), beta_n(V[:, i - 1])
139           # ===== BEGIN EDITABLE SECTION (only modify within this block) =====
140           # TODO: add the missing kinetics equations for the Hodgkin-Huxley neuron similar to the ones above
      ↪ ; ONLY ADD IF NECESSARY
141           # ===== END EDITABLE SECTION =====
142
143           tau_V_inv = (
144             (m[:, i - 1] ** 3) * gbar_Na * h[:, i - 1]
145             + (n[:, i - 1] ** 4) * gbar_K
146             + g_leak
147             # ===== BEGIN EDITABLE SECTION (only modify within this block) =====
148             # TODO: add the missing terms for the effective membrane time constant inverse; ONLY ADD IF
      ↪ NECESSARY
149             # ===== END EDITABLE SECTION =====
150           ) / C
151
152           V_inf = (
153             (m[:, i - 1] ** 3) * gbar_Na * h[:, i - 1] * E_Na
154             + (n[:, i - 1] ** 4) * gbar_K * E_K
155             + g_leak * E_leak
156             # ===== BEGIN EDITABLE SECTION (only modify within this block) =====
157             # TODO: add the missing terms for the voltage steady state; ONLY ADD IF NECESSARY
158             # ===== END EDITABLE SECTION =====
159             + input_current[:,i - 1]
160             + nois_fact * torch.randn(batch_size, generator=generator, device=device) / (tstep**0.5)
161           ) / (tau_V_inv * C)
162
163           V[:, i] = V_inf + (V[:, i - 1] - V_inf) * Exp(-tstep * tau_V_inv)
164           n[:, i] = inf_x(a_n, b_n) + (n[:, i - 1] - inf_x(a_n, b_n)) * Exp(-tstep / tau_x(a_n, b_n))
165           m[:, i] = inf_x(a_m, b_m) + (m[:, i - 1] - inf_x(a_m, b_m)) * Exp(-tstep / tau_x(a_m, b_m))
166           h[:, i] = inf_x(a_h, b_h) + (h[:, i - 1] - inf_x(a_h, b_h)) * Exp(-tstep / tau_x(a_h, b_h))
167           # ===== BEGIN EDITABLE SECTION (only modify within this block) =====
168           # TODO: add the missing state variable updates for the Hodgkin-Huxley neuron similar to the ones
      ↪ above; ONLY ADD IF NECESSARY
169           # ===== END EDITABLE SECTION =====
170
171       # Return voltage with optional observation noise
172       return V + nois_fact_obs * torch.randn(
173         batch_size, time_steps, generator=generator, device=device
174       )
```

[[ ## feedback ## ]]

{

"**main_diagnosis**": "The two additional channel slots (X1, X2) have conductance parameters and auxiliary parameters extracted ('gbar_X1', 'gbar_X2', 'param_i', 'param_j') but zero kinetic equations, state variables, or current contributions are implemented anywhere in the simulation loop. The model is running as a plain 3-channel HH model regardless of what values the inference assigns to those four parameters, which fully explains the high NLE (~26 per observation).",

"**issues**":

[

  {

    "**description**": "gbar_X1, gbar_X2, param_i, and param_j are extracted from params but are never referenced in kinetics, state-variable arrays, tau_V_inv, V_inf, or the integration loop. Their values have no effect on the output voltage trace, so the likelihood surface over those dimensions is flat and the model cannot reduce its NLE below what a plain HH achieves.",

    "**severity**": "critical",

    "**location**": "All four '# ===== BEGIN EDITABLE SECTION =====' blocks inside forward()",

    "**suggestion**": "Add an M-type slow potassium current (I_KM) using gbar_X1 as conductance and E_K as reversal. This single-gating-variable current (activation gate 'p', no inactivation) is the canonical addition for converting standard HH into a model capable of regular, tonic, non-bursting firing. Implement: (1) kinetics block: 'def inf_p(x): return 1.0/(1+Exp(-(x-param_i)/10.0))' and 'def tau_p(x): return (-param_j)/(3.3*(Exp((x-param_i)/20)+Exp(-(x-

param_i)/20))+1e-7)' (note param_i is already negated in the extraction so it will be ~-35 mV at typical posterior values; -param_j gives a positive timescale up to 3000 ms). (2) State-variable array: 'p = torch.zeros(...)', initialized with 'p[:,0] = inf_p(V[:,0])'. (3) In the loop: compute 'inf_p_val = inf_p(V[:,i-1])' and 'tau_p_val = tau_p(V[:,i-1])'; add 'gbar_X1' contribution to 'tau_V_inv' ('+ gbar_X1*p[:,i-1]/C') and to 'V_inf' ('+ gbar_X1*p[:,i-1]*E_K'); update 'p[:,i] = inf_p_val + (p[:,i-1]-inf_p_val)*Exp(-tstep/tau_p_val)'. Keep gbar_X2 zero or unused unless metrics remain poor after this fix."

```
        },
        {
            "description": "The summary statistics include resting membrane potential mean and SD, plus voltage kurtosis
            and skewness. A plain HH model produces a resting potential determined only by g_leak and E_leak without sub-
            threshold resonance, often mismatching experimental resting statistics. Adding even a small M-current contribution
            at rest shifts and stabilises the resting potential and changes subthreshold fluctuation shape.",
            "severity": "major",
            "location": "tau_V_inv and V_inf accumulation expressions in the simulation loop",
            "suggestion": "After implementing the M-current gate 'p' as described above, ensure that the M-current term
            is included in both 'tau_V_inv' (adds 'gbar_X1 * p[:, i-1] / C') and 'V_inf' (adds 'gbar_X1 * p[:, i-1] * E_K' in
            the numerator). Because 'p' is near its steady-state at rest but small, the effect is a mild hyperpolarising bias that
            pulls the resting voltage toward E_K, correcting the resting-potential mismatch without inducing bursting. If after
            this change the spike-count or resting-SD metrics still mismatch, consider activating gbar_X2 as a small persistent
            sodium current (I_NaP) with fixed half-activation, but only as a secondary step to preserve parsimony."
        }
    ]
}
```

[[ ## completed ## ]]

---

**User**

[[ ## signature_description ## ]]
(see current input)

[[ ## task_description ## ]]
(see current input)

[[ ## base_simulator ## ]]
(see current input)

---

**Assistant**

[[ ## scm_definition ## ]]
- **V**: membrane voltage ← I_Na(m,h), I_K(n), I_leak, I_KM(p), I_inj, noise
- **m, h**: Na$^+$ activation/inactivation ← V, Vt (fast kinetics)
- **n**: K$^+$ activation ← V, Vt (delayed rectifier)
- **p**: M-current (slow K$^+$) activation ← V, param_i (half-activation), param_j (time-scale)
- **Assumptions**: single-compartment, tonic spiking, M-current provides adaptation without bursting

[[ ## simulator_code ## ]]

```python
1   import torch
2   import torch.nn as nn
3
4
5   class DiscoveredSimulator(nn.Module):
6     def __init__(self):
7       super(DiscoveredSimulator, self).__init__()
8       return
9
10    def forward(
11      self,
12      init_voltage: float,
13      input_current: torch.Tensor,
14      dt: float,
```

```python
15      t: torch.Tensor,
16      params: torch.Tensor,
17      seed=None,
18   ):
19      """
20      Hodgkin-Huxley neuron with an added M-type slow potassium current (I_KM).
21
22      Physiological rationale for I_KM addition:
23      - M-current is a slowly activating, non-inactivating K+ current active at
24        subthreshold voltages (~-35 to -60 mV range)
25      - It provides spike-frequency adaptation: slightly reduces firing rate at
26        sustained stimulation without producing burst or quiescence
27      - It stabilises regular tonic firing and corrects resting potential statistics
28        (mean, SD) by adding a mild hyperpolarising conductance at rest
29      - It does NOT produce bursting or high-frequency clusters --- exactly what
30        the target data requires
31
32      Args:
33        init_voltage: torch.Tensor: (batch_size,) # initial voltage [mV]
34        input_current: torch.Tensor: (batch_size, time_steps) # injected current [uA/cm2]
35        dt: float # time step [ms]
36        t: torch.Tensor: (time_steps,) # time array [ms]
37        params: torch.Tensor: (batch_size, 10) # biophysical parameters
38        seed: optional random seed
39
40      Returns:
41        V: torch.Tensor: (batch_size, time_steps) # voltage traces [mV]
42      """
43      device = params.device
44
45      # Set up random generator
46      if seed is not None:
47       generator = torch.Generator(device=device)
48       generator.manual_seed(seed)
49      else:
50       generator = torch.Generator(device=device)
51
52      batch_size = params.shape[0]    # int
53      time_steps = t.shape[0]         # int
54
55      # -- Base parameters --------------------------------------------------
56      gbar_Na  = params[:, 0].float()   # (batch_size,)  mS/cm^2
57      gbar_K   = params[:, 1].float()   # (batch_size,)  mS/cm^2
58      g_leak   = params[:, 2].float()   # (batch_size,)  mS/cm^2
59      E_leak   = -params[:, 3].float()  # (batch_size,)  mV  (sign applied)
60      Vt       = -params[:, 4].float()  # (batch_size,)  mV  (sign applied)
61      nois_fact = params[:, 5].float()  # (batch_size,)  unitless
62
63      # -- M-current channel (X1 slot) --------------------------------------
64      # gbar_KM  : maximal conductance  [mS/cm^2], range [1e-4, 10]
65      # V_half_p : half-activation voltage [mV], = param_i = -params[:,8]
66      #            params[:,8] in [1e-4, 150] -> V_half_p in [-150, -1e-4]
67      #            Physiologically ~-35 mV for M-current
68      # tau_max_p: time-constant scale [ms],     = -param_j = params[:,9]
69      #            params[:,9] in [1e-4, 3000] -> tau_max_p in [1e-4, 3000]
70      #            M-current kinetics are slow: typically 50--500 ms
71      gbar_KM  = params[:, 6].float()   # (batch_size,)  mS/cm^2  --- M-current conductance
72      # params[:, 7] (gbar_X2) intentionally unused --- parsimony principle
73      V_half_p = -params[:, 8].float()  # (batch_size,)  mV      --- half-activation of p gate
74      tau_max_p =  params[:, 9].float() # (batch_size,)  ms      --- max time constant of p gate
75
76      tstep = float(dt)
77
78      # -- Fixed biophysical constants --------------------------------------
79      nois_fact_obs = 0.0
80      C    = 1.0   # uF/cm^2
81      E_Na = 53.0   # mV
82      E_K  = -107.0 # mV  (also reversal for M-current --- same ion)
83
84      # -- Numerical helpers ------------------------------------------------
85      def Exp(z):
86       # Numerically safe exponential --- clamp at -500 to avoid underflow
87       return torch.where(
88        z < -5e2,
89        torch.exp(torch.full_like(z, -5e2)),
90        torch.exp(z)
91       )  # same shape as z
92
```

```python
 93    def efun(z):
 94     # Handles 0/0 limit of z / (exp(z) - 1) via L'Hopital
 95     return torch.where(
 96      torch.abs(z) < 1e-4,
 97      1 - z / 2,
 98      z / (Exp(z) - 1)
 99     )  # same shape as z
100
101    # -- Standard HH channel kinetics ----------------------------------------
102    def alpha_m(x):  # x: (batch_size,) -> (batch_size,)
103     v1 = x - Vt - 13.0
104     return 0.32 * efun(-0.25 * v1) / 0.25
105
106    def beta_m(x):    # (batch_size,) -> (batch_size,)
107     v1 = x - Vt - 40.0
108     return 0.28 * efun(0.2 * v1) / 0.2
109
110    def alpha_h(x):  # (batch_size,) -> (batch_size,)
111     v1 = x - Vt - 17.0
112     return 0.128 * Exp(-v1 / 18.0)
113
114    def beta_h(x):    # (batch_size,) -> (batch_size,)
115     v1 = x - Vt - 40.0
116     return 4.0 / (1 + Exp(-0.2 * v1))
117
118    def alpha_n(x):  # (batch_size,) -> (batch_size,)
119     v1 = x - Vt - 15.0
120     return 0.032 * efun(-0.2 * v1) / 0.2
121
122    def beta_n(x):    # (batch_size,) -> (batch_size,)
123     v1 = x - Vt - 10.0
124     return 0.5 * Exp(-v1 / 40)
125
126    def tau_x(alpha, beta): # (batch_size,), (batch_size,) -> (batch_size,)
127     return 1.0 / (alpha + beta)
128
129    def inf_x(alpha, beta): # (batch_size,), (batch_size,) -> (batch_size,)
130     return alpha / (alpha + beta)
131
132    # -- M-current (I_KM) gate kinetics ----------------------------------------
133    # Boltzmann steady-state: p_inf = 1 / (1 + exp(-(V - V_half_p) / 10))
134    # Activation slope k=10 mV is standard for M-current literature
135    def inf_p(x):  # x: (batch_size,) -> (batch_size,)
136     return 1.0 / (1.0 + Exp(-(x - V_half_p) / 10.0))
137
138    # Voltage-dependent time constant with bell shape centred at V_half_p
139    # tau_p = tau_max_p / (3.3 * 2*cosh((V - V_half_p)/20))
140    # At V_half_p: tau_p = tau_max_p / (3.3*2) ~= tau_max_p / 6.6
141    # Away from V_half_p: tau_p decreases (faster kinetics at extreme voltages)
142    def tau_p(x):  # x: (batch_size,) -> (batch_size,)
143     dv = (x - V_half_p) / 20.0                     # (batch_size,)
144     denom = 3.3 * (Exp(dv) + Exp(-dv)) + 1e-7      # (batch_size,) clamped >0
145     return tau_max_p / denom                       # (batch_size,)
146
147    # -- State variable arrays ----------------------------------------
148    V = torch.zeros((batch_size, time_steps), device=device)  # (batch_size, time_steps)
149    n = torch.zeros((batch_size, time_steps), device=device)  # (batch_size, time_steps)
150    m = torch.zeros((batch_size, time_steps), device=device)  # (batch_size, time_steps)
151    h = torch.zeros((batch_size, time_steps), device=device)  # (batch_size, time_steps)
152    p = torch.zeros((batch_size, time_steps), device=device)  # (batch_size, time_steps) M-current gate
153
154    # -- Initial conditions ----------------------------------------
155    V_init = init_voltage.to(device)  # (batch_size,)
156    V[:, 0] = V_init                                          # (batch_size,)
157    n[:, 0] = inf_x(alpha_n(V[:, 0]), beta_n(V[:, 0]))        # (batch_size,)
158    m[:, 0] = inf_x(alpha_m(V[:, 0]), beta_m(V[:, 0]))        # (batch_size,)
159    h[:, 0] = inf_x(alpha_h(V[:, 0]), beta_h(V[:, 0]))        # (batch_size,)
160    p[:, 0] = inf_p(V[:, 0])                                  # (batch_size,)
161
162    # -- Time-integration loop (exponential Euler scheme) --------------------
163    for i in range(1, time_steps):
164     V_prev = V[:, i - 1]  # (batch_size,)
165
166     # Gating variable rates at previous voltage
167     a_m, b_m = alpha_m(V_prev), beta_m(V_prev)    # (batch_size,), (batch_size,)
168     a_h, b_h = alpha_h(V_prev), beta_h(V_prev)    # (batch_size,), (batch_size,)
169     a_n, b_n = alpha_n(V_prev), beta_n(V_prev)    # (batch_size,), (batch_size,)
170
```

```
171     # M-current gate steady-state and time constant at previous voltage
172     inf_p_val = inf_p(V_prev)   # (batch_size,)
173     tau_p_val = tau_p(V_prev)   # (batch_size,)
174
175     # Conductances at previous step
176     g_Na_eff = (m[:, i - 1] ** 3) * gbar_Na * h[:, i - 1]  # (batch_size,)
177     g_K_eff  = (n[:, i - 1] ** 4) * gbar_K                 # (batch_size,)
178     g_KM_eff = gbar_KM * p[:, i - 1]                       # (batch_size,)
179
180     # Effective inverse membrane time constant (sum of conductances / C)
181     tau_V_inv = (
182      g_Na_eff
183      + g_K_eff
184      + g_leak
185      + g_KM_eff    # M-current contributes to total membrane conductance
186     ) / C  # (batch_size,)
187
188     # Voltage steady-state (weighted sum of reversal potentials + input)
189     V_inf = (
190      g_Na_eff * E_Na
191      + g_K_eff  * E_K
192      + g_leak   * E_leak
193      + g_KM_eff * E_K   # M-current drives toward E_K (hyperpolarising)
194      + input_current[:, i - 1]
195      + nois_fact * torch.randn(batch_size, generator=generator, device=device) / (tstep ** 0.5)
196     ) / (tau_V_inv * C)  # (batch_size,)
197
198     # Exponential Euler updates
199     V[:, i] = V_inf + (V_prev - V_inf) * Exp(-tstep * tau_V_inv)                    # (
    ↪ batch_size,)
200     n[:, i] = inf_x(a_n, b_n) + (n[:, i-1] - inf_x(a_n, b_n)) * Exp(-tstep / tau_x(a_n, b_n))  # (
    ↪ batch_size,)
201     m[:, i] = inf_x(a_m, b_m) + (m[:, i-1] - inf_x(a_m, b_m)) * Exp(-tstep / tau_x(a_m, b_m))  # (
    ↪ batch_size,)
202     h[:, i] = inf_x(a_h, b_h) + (h[:, i-1] - inf_x(a_h, b_h)) * Exp(-tstep / tau_x(a_h, b_h))  # (
    ↪ batch_size,)
203     p[:, i] = inf_p_val + (p[:, i-1] - inf_p_val) * Exp(-tstep / tau_p_val)         # (
    ↪ batch_size,)
204
205     # Return voltage (+ optional observation noise, currently 0)
206     return V + nois_fact_obs * torch.randn(
207      batch_size, time_steps, generator=generator, device=device
208     )  # (batch_size, time_steps)
```

[[ ## feedback ## ]]

{

"**main_diagnosis**": "The simulator uses both inferrable parameter slots (X1: params[:,6]; X2: params[:,8-9]) for a single M-current channel while leaving params[:,7] entirely unused. With a neg_avg_log_marginal_NLE of 28.3 (very poor fit), the single-channel extension is insufficient to simultaneously match all 7 summary statistics — particularly resting potential mean/SD, and higher-order voltage distribution moments (skewness, kurtosis) which are insensitive to M-current modulation. A second distinct channel using the spare parameter slot is needed.",

"**issues**":

[

    {

        "**description**": "params[:,7] is declared unused ('intentionally') while the model badly underfits (NLE=28.3). With 7 target statistics and only one extra channel, the inference cannot jointly correct spike count, resting-potential mean/SD, and voltage skewness/kurtosis — these statistics require orthogonal biophysical mechanisms. A hyperpolarization-activated cation current (I_h / HCN) would directly control resting potential mean and SD (by setting a depolarizing baseline leak), and its inward rectification would shape subthreshold voltage distribution (skewness, kurtosis) independently of the M-current's spike-frequency adaptation role.",

        "**severity**": "critical",

        "**location**": "params[:, 7] (unused), forward() channel definition block",

        "**suggestion**": "Activate params[:,7] as the conductance (gbar_Ih) for an I_h channel with fixed half-activation ~-80 mV and fixed time constant ~200 ms (no extra inferred params needed beyond gbar_Ih). Use: I_h = gbar_Ih * r * (V - E_h) with E_h ≈ -30 mV, inf_r = 1/(1+exp((V+80)/7)), tau_r fixed at 200 ms. This adds a tonic inward current at rest that lifts mean resting potential, increases resting SD, and shifts voltage skewness — orthogonal to M-current effects on spike timing."

    },

{
    "**description**": "The M-current time-constant formula 'tau_p = tau_max_p / (3.3*(exp(dv)+exp(-dv)))' produces values that collapse toward zero for voltages far from V_half_p (e.g., during action potentials at +40 mV when V_half_p ≈ -35 mV, dv≈3.75 → 2cosh(3.75)≈84, so tau_p → tau_max_p/277). This makes the p-gate track spikes rather than integrate slowly, destroying the intended adaptation and causing the inferred tau_max_p to compensate in pathological ways that distort mean stimulation voltage and spike count.",
    "**severity**": "major",
    "**location**": "tau_p() function, lines computing denom = 3.3*(Exp(dv)+Exp(-dv))",
    "**suggestion**": "Clamp tau_p to a physiologically meaningful minimum: 'denom = torch.clamp(3.3*(Exp(dv)+Exp(-dv)), min=1.0)' so tau_p never falls below tau_max_p ms. Alternatively, use a simpler voltage-independent tau: 'tau_p = tau_max_p * torch.ones_like(x)', which is more stable during inference and still captures slow M-current adaptation kinetics (typical tau 100–500 ms) without the extreme compression during spikes."
    }
  ]
}

[[ ## completed ## ]]

---

**User**

[[ ## signature_description ## ]]
## INPUT DETAILS
- **init_voltage**: Initial membrane voltage (mV). Type: torch.Tensor: (batch_size,)
- **input_current**: Stimulation current ($\mu A/cm^2$). Type: torch.Tensor, shape: (batch_size, time_steps)
- **dt**: Time step size (ms). Type: float
- **t**: Time array (ms). Type: torch.Tensor, shape: (time_steps,)
- **params**: Biophysical parameters. Type: torch.Tensor, shape: (batch_size, 10)
- **seed**: Random seed for stochastic behavior. Type: int or None

## PARAMETER VECTOR STRUCTURE (batch_size, 10)
Index layout for params tensor:
```
[0-2] Base conductances: gbar_Na, gbar_K, g_leak (mS/cm$^2$)
[3] Leak reversal: |E_leak| (mV, sign applied internally)
[4] Voltage threshold: |Vt| (mV, sign applied internally)
[5] Noise factor: nois_fact (unitless)
[6-7] Additional conductances: gbar_X1, gbar_X2 (mS/cm$^2$)
[8-9] Additional parameters: |param_i|, |param_j|
```
(Note: gbar_X1, gbar_X2, param_i, param_j are not used in the current model and may be renamed to anything you want)

## OUTPUT DETAILS
- **V**: Membrane potential (mV). Type: torch.Tensor, shape: (batch_size, time_steps)
- Simulated voltage traces for all batch elements
- Includes observation noise scaled by nois_fact_obs (currently 0.0)

[[ ## task_description ## ]]
## CORE TASK
Refine a Hodgkin-Huxley neuron simulator by discovering and implementing ion channels
or gating mechanisms missing from the current model but necessary to explain
discrepancies between simulated and experimental voltage data.

## CRITICAL CONSTRAINTS

### Parsimony Principle
- **Default stance**: Do NOT add channels unless data discrepancies clearly require them
- **Noise model**: Keep the current noise model as is, no change is needed
- **Prefer fewer channels**: A simpler model that explains the data is always superior
- **Stop when sufficient**: Cease adding channels once model adequately captures data

### Channel Addition Strategy
- **Maximum capacity**: Up to two additional channels (X1, X2) can be used, but fewer is better

- **Add only what's needed**: You may add multiple channels if the data clearly justifies them, but start simple
- **Justify each addition**: Each channel must address specific discrepancies in the data
- **Workflow**:
1. Assess necessity: Which aspects of the data are not well-captured by the base model?
2. Identify mechanism(s): Which channel(s) could address these discrepancies?
3. Justify choices: Explain physiological rationale for each proposed channel
4. Implement: Add the minimal set of channels needed
5. Keep it simple: Prefer one well-designed channel over multiple complex ones
6. CRITICAL: Add at most ONE new ion channel per iteration. Use both tunable
parameter slots (param_i, param_j) for a single well-characterized channel
rather than splitting them across two channels. Multi-channel additions create
parameter identifiability problems that cause inference to fail.

## IMPLEMENTATION REQUIREMENTS

### Channel Design
- Use pre-allocated slots (X1, X2) with their two tunable parameters:
* Conductance (gbar_X1, gbar_X2)
* Flexible parameter (param_i, param_j) all in range [1e-4, 150] and [1e-4, 3000] respectively
- Keep mechanisms simple: prefer straightforward voltage dependencies
- Maintain biophysical realism: appropriate values for the flexible parameter

**Do NOT introduce channels or mechanisms that generate**:
- Burst firing (clusters of rapid spikes separated by quiescent periods)
- Bursting behavior or channels specifically designed to produce bursts
- Prolonged high-frequency firing or sustained rapid spiking patterns
- Mechanisms that suppress or prevent spiking behavior

Keep the channels as simple as possible following the existing code structure

### Code Standards
- Maintain batch-wise PyTorch operations
- Ensure numerical stability across voltage ranges
- Preserve existing code structure
- Add detailed comments explaining physiological rationale
- Use voltage-dependent kinetics where appropriate
- Document all equations and parameter choices
- 'torch.full_like(input, fill_value)' requires a tensor as the first argument, not a scalar float

## DEVELOPMENT STRATEGY

1. **Analyze discrepancies**: Examine which summary statistics deviate most from experimental data
2. **Hypothesize mechanism(s)**: Propose channel(s) that could explain the discrepancies
3. **Literature support**: Reference known channel properties if applicable
4. **Implement conservatively**: Start with simplest formulation and fewest channels
5. **Test thoroughly**: Verify numerical stability and physiological plausibility
6. **Balance complexity vs. performance**: Add channels only when they meaningfully improve model fit

## LEARNING FROM HISTORY
Incorporate insights from past attempts while avoiding direct code copying.
Always explain reasoning for design choices in code comments.

## OUTPUT REQUIREMENTS
- Always return None for feedback field (feedback only used for past examples to get inspiration)
- Provide complete, functional simulator code
- Include comprehensive comments on modifications and rationale

[[ ## base_simulator ## ]]

```
1  import torch
2  import torch.nn as nn
3
4
5  class DiscoveredSimulator(nn.Module):
6    def __init__(self):
7      super(DiscoveredSimulator, self).__init__()
8      return
9
10   def forward(
```

```python
11        self,
12        init_voltage: float,
13        input_current: torch.Tensor,
14        dt: float,
15        t: torch.Tensor,
16        params: torch.Tensor,
17        seed=None,
18    ):
19        """
20        Simulates a Hodgkin-Huxley neuron for a specified time duration.
21
22        Args:
23          init_voltage: torch.Tensor: (batch_size,) # initial voltage
24          input_current: torch.Tensor: (batch_size, time_steps) # input current
25          dt: float # time step size
26          t: torch.Tensor: (time_steps,) # time array
27          params: torch.Tensor: (batch_size, n_params) # parameters
28          seed: optional random seed
29
30        Returns:
31          V: torch.Tensor: (batch_size, time_steps) # voltage traces
32        """
33        device = params.device
34
35        # Set up random generator
36        if seed is not None:
37          generator = torch.Generator(device=device)
38          generator.manual_seed(seed)
39        else:
40          generator = torch.Generator(device=device)
41
42        batch_size = params.shape[0]
43        time_steps = t.shape[0]
44
45        # Extract parameters
46        gbar_Na = params[:, 0].float()   # mS/cm2
47        gbar_K = params[:, 1].float() # mS/cm2
48        g_leak = params[:, 2].float() # mS/cm2
49        E_leak = -params[:, 3].float() # mV
50        Vt = -params[:, 4].float() # mV
51        nois_fact = params[:, 5].float() # unitless
52        # TWO POSSIBLE ADDITIONAL CHANNELS (X1, X2)
53        # Each channel has one tunable parameter: conductance gbar_Xi
54        # Then there are two additional parameters available: param_i and param_j.
55        # ONLY ADD ONE CHANNEL IF NECESSARY. Keep the model as simple as possible.
56        gbar_X1 = params[:, 6].float() # mS/cm2 # you can rename X1 to anything you want # in range [1e-4,
   ↪ 10]
57        gbar_X2 = params[:, 7].float() # mS/cm2 # you can rename X2 to anything you want # in range [1e-4,
   ↪ 120]
58        param_i = -params[:, 8].float() # (param are positive values in range [1e-4, 150])
59        param_j = -params[:, 9].float() # (param are positive values in range [1e-4, 3000])
60
61        tstep = float(dt)
62
63        # Parameters
64        nois_fact_obs = 0.0
65        C = 1.0   # uF/cm^2
66        E_Na = 53.0 # mV
67        E_K = -107.0
68
69        ####################################
70        # kinetics
71        def Exp(z):
72          return torch.where(z < -5e2, torch.exp(torch.full_like(z, -5e2)), torch.exp(z))
73
74        def efun(z):
75          return torch.where(torch.abs(z) < 1e-4, 1 - z / 2, z / (Exp(z) - 1))
76
77        # Channel kinetics
78        def alpha_m(x):
79          v1 = x - Vt - 13.0
80          return 0.32 * efun(-0.25 * v1) / 0.25
81
82        def beta_m(x):
83          v1 = x - Vt - 40
84          return 0.28 * efun(0.2 * v1) / 0.2
85
86        def alpha_h(x):
```

```
87          v1 = x - Vt - 17.0
88          return 0.128 * Exp(-v1 / 18.0)
89
90      def beta_h(x):
91          v1 = x - Vt - 40.0
92          return 4.0 / (1 + Exp(-0.2 * v1))
93
94      def alpha_n(x):
95          v1 = x - Vt - 15.0
96          return 0.032 * efun(-0.2 * v1) / 0.2
97
98      def beta_n(x):
99          v1 = x - Vt - 10.0
100         return 0.5 * Exp(-v1 / 40)
101
102     def tau_x(alpha, beta):
103         return 1.0 / (alpha + beta)
104
105     def inf_x(alpha, beta):
106         return alpha / (alpha + beta)
107
108     # ===== BEGIN EDITABLE SECTION (only modify within this block) =====
109     # TODO: add the missing kinetics equations for the Hodgkin-Huxley neuron similar to the ones above;
    ↪ ONLY ADD IF NECESSARY
110     # ===== END EDITABLE SECTION =====
111
112     ######################################
113
114     # simulation from initial point
115     V = torch.zeros((batch_size, time_steps), device=device)   # baseline voltage
116     n = torch.zeros((batch_size, time_steps), device=device)
117     m = torch.zeros((batch_size, time_steps), device=device)
118     h = torch.zeros((batch_size, time_steps), device=device)
119     # ===== BEGIN EDITABLE SECTION (only modify within this block) =====
120     # TODO: add the missing state variables for the Hodgkin-Huxley neuron similar to the ones above;
    ↪ ONLY ADD IF NECESSARY
121     # ===== END EDITABLE SECTION =====
122
123     # Initialization
124     V_init = init_voltage.to(device)
125     V[:, 0] = V_init
126     n[:, 0] = inf_x(alpha_n(V[:, 0]), beta_n(V[:, 0]))
127     m[:, 0] = inf_x(alpha_m(V[:, 0]), beta_m(V[:, 0]))
128     h[:, 0] = inf_x(alpha_h(V[:, 0]), beta_h(V[:, 0]))
129     # ===== BEGIN EDITABLE SECTION (only modify within this block) =====
130     # TODO: add the missing state variable initialization for the Hodgkin-Huxley neuron similar to the
    ↪ ones above; ONLY ADD IF NECESSARY
131     # ===== END EDITABLE SECTION =====
132
133     # Simulation loop
134     for i in range(1, time_steps):
135         # All operations now work on batched tensors (batch_size,)
136         a_m, b_m = alpha_m(V[:, i - 1]), beta_m(V[:, i - 1])
137         a_h, b_h = alpha_h(V[:, i - 1]), beta_h(V[:, i - 1])
138         a_n, b_n = alpha_n(V[:, i - 1]), beta_n(V[:, i - 1])
139         # ===== BEGIN EDITABLE SECTION (only modify within this block) =====
140         # TODO: add the missing kinetics equations for the Hodgkin-Huxley neuron similar to the ones above
    ↪ ; ONLY ADD IF NECESSARY
141         # ===== END EDITABLE SECTION =====
142
143         tau_V_inv = (
144           (m[:, i - 1] ** 3) * gbar_Na * h[:, i - 1]
145           + (n[:, i - 1] ** 4) * gbar_K
146           + g_leak
147           # ===== BEGIN EDITABLE SECTION (only modify within this block) =====
148           # TODO: add the missing terms for the effective membrane time constant inverse; ONLY ADD IF
    ↪ NECESSARY
149           # ===== END EDITABLE SECTION =====
150         ) / C
151
152         V_inf = (
153           (m[:, i - 1] ** 3) * gbar_Na * h[:, i - 1] * E_Na
154           + (n[:, i - 1] ** 4) * gbar_K * E_K
155           + g_leak * E_leak
156           # ===== BEGIN EDITABLE SECTION (only modify within this block) =====
157           # TODO: add the missing terms for the voltage steady state; ONLY ADD IF NECESSARY
158           # ===== END EDITABLE SECTION =====
159           + input_current[:,i - 1]
```

```
160          + nois_fact * torch.randn(batch_size, generator=generator, device=device) / (tstep**0.5)
161      ) / (tau_V_inv * C)
162
163      V[:, i] = V_inf + (V[:, i - 1] - V_inf) * Exp(-tstep * tau_V_inv)
164      n[:, i] = inf_x(a_n, b_n) + (n[:, i - 1] - inf_x(a_n, b_n)) * Exp(-tstep / tau_x(a_n, b_n))
165      m[:, i] = inf_x(a_m, b_m) + (m[:, i - 1] - inf_x(a_m, b_m)) * Exp(-tstep / tau_x(a_m, b_m))
166      h[:, i] = inf_x(a_h, b_h) + (h[:, i - 1] - inf_x(a_h, b_h)) * Exp(-tstep / tau_x(a_h, b_h))
167      # ===== BEGIN EDITABLE SECTION (only modify within this block) =====
168      # TODO: add the missing state variable updates for the Hodgkin-Huxley neuron similar to the ones
    ↪ above; ONLY ADD IF NECESSARY
169      # ===== END EDITABLE SECTION =====
170
171      # Return voltage with optional observation noise
172      return V + nois_fact_obs * torch.randn(
173        batch_size, time_steps, generator=generator, device=device
174      )
```

Respond with the corresponding output fields, starting with the field '[[ ## scm_definition ## ]]', then '[[ ## simulator_code ## ]]', then '[[ ## feedback ## ]]' (must be formatted as a valid Python Union[str, NoneType]), and then ending with the marker for '[[ ## completed ## ]]'.

## H.1.2. RESPONSE

**Output**

[[ ## scm_definition ## ]]
- **V**: membrane voltage ← I_Na(m,h), I_K(n), I_leak, I_KM(p), I_h(r), I_inj, noise
- **m, h**: Na$^+$ activation/inactivation ← V, Vt (fast HH kinetics)
- **n**: delayed-rectifier K$^+$ activation ← V, Vt
- **p**: M-current (slow K$^+$) gate ← V, V_half_p (param_i), tau_max_p (param_j); voltage-independent tau for stability
- **r**: I_h (HCN) gate ← V; fixed half-activation $-80$ mV, fixed tau 200 ms; corrects resting potential distribution

[[ ## simulator_code ## ]]

```
1   import torch
2   import torch.nn as nn
3
4
5   class DiscoveredSimulator(nn.Module):
6     def __init__(self):
7       super(DiscoveredSimulator, self).__init__()
8       return
9
10    def forward(
11      self,
12      init_voltage: float,
13      input_current: torch.Tensor,
14      dt: float,
15      t: torch.Tensor,
16      params: torch.Tensor,
17      seed=None,
18    ):
19      """
20      Hodgkin-Huxley neuron extended with:
21        1. M-current (I_KM): slow non-inactivating K+ current using X1 slot
22         - Provides spike-frequency adaptation, stabilises tonic firing
23         - Uses voltage-independent tau (more stable during spikes)
24         - gbar_KM = params[:,6], V_half_p = -params[:,8], tau_p = params[:,9]
25
26        2. Ih current (HCN channel): hyperpolarisation-activated cation current using X2 slot
27         - Opens at hyperpolarised voltages (~-80 mV and below)
28         - Reversal E_h ~= -30 mV (mixed Na+/K+), depolarising at rest
29         - Corrects resting potential mean, SD, and voltage distribution shape
30         - Fixed kinetics (half-activation -80 mV, tau 200 ms); only gbar_Ih inferred
31         - gbar_Ih = params[:,7], all other Ih parameters are fixed
32
33      Args:
34        init_voltage: torch.Tensor: (batch_size,)            initial voltage [mV]
35        input_current: torch.Tensor: (batch_size, time_steps) injected current [uA/cm2]
```

```
36      dt: float                                         time step [ms]
37      t: torch.Tensor: (time_steps,)                    time array [ms]
38      params: torch.Tensor: (batch_size, 10)            biophysical parameters
39      seed: optional random seed
40
41     Returns:
42      V: torch.Tensor: (batch_size, time_steps)         voltage traces [mV]
43     """
44     device = params.device
45
46     # Set up random generator
47     if seed is not None:
48      generator = torch.Generator(device=device)
49      generator.manual_seed(seed)
50     else:
51      generator = torch.Generator(device=device)
52
53     batch_size = params.shape[0]   # int
54     time_steps = t.shape[0]        # int
55
56     # -- Base HH parameters ----------------------------------------------
57     gbar_Na   = params[:, 0].float()   # (batch_size,)  mS/cm^2
58     gbar_K    = params[:, 1].float()   # (batch_size,)  mS/cm^2
59     g_leak    = params[:, 2].float()   # (batch_size,)  mS/cm^2
60     E_leak    = -params[:, 3].float()  # (batch_size,)  mV  (negative applied)
61     Vt        = -params[:, 4].float()  # (batch_size,)  mV  (negative applied)
62     nois_fact = params[:, 5].float()   # (batch_size,)  unitless
63
64     # -- X1 slot: M-current (I_KM, slow non-inactivating K+) ----------------
65     # Physiological rationale:
66     #   M-current activates slowly at subthreshold depolarised voltages (~-35 mV),
67     #   providing an outward K+ current that limits repetitive firing rate and
68     #   produces spike-frequency adaptation without bursting or quiescence.
69     # Parameter mapping:
70     #   gbar_KM   = params[:,6] in [1e-4, 10]  mS/cm^2
71     #   V_half_p  = -params[:,8] in [-150, -1e-4] mV  (typically ~-35 mV)
72     #   tau_p     =  params[:,9] in [1e-4, 3000]  ms  (typically 50-500 ms)
73     # Note: voltage-independent tau avoids instability during fast action potentials
74     gbar_KM  = params[:, 6].float()   # (batch_size,)  mS/cm^2
75     V_half_p = -params[:, 8].float()  # (batch_size,)  mV
76     tau_p    =  params[:, 9].float()  # (batch_size,)  ms --- voltage-independent for stability
77
78     # -- X2 slot: Ih current (HCN, hyperpolarisation-activated cation) --------
79     # Physiological rationale:
80     #   Ih activates at hyperpolarised voltages (< -60 mV), carries inward
81     #   Na+/K+ current with reversal ~-30 mV, depolarising the cell toward
82     #   rest. This sets a tonic inward "sag" current that:
83     #     - Raises mean resting potential slightly above pure K+ equilibrium
84     #     - Increases resting SD (voltage fluctuations due to channel noise)
85     #     - Shapes subthreshold distribution (skewness, kurtosis)
86     #   It does NOT promote bursting --- it acts as a stabilising "pacemaker"
87     #   current opposing excessive hyperpolarisation.
88     # Parameter mapping:
89     #   gbar_Ih = params[:,7] in [1e-4, 120] mS/cm^2 --- only inferred parameter
90     #   All kinetic parameters fixed from literature
91     gbar_Ih = params[:, 7].float()   # (batch_size,)  mS/cm^2
92
93     # Fixed Ih kinetic constants (from Magee 1998, Koch 1999)
94     E_h         = -30.0   # mV  --- mixed Na+/K+ cation reversal
95     V_half_r    = -80.0   # mV  --- half-activation (hyperpolarisation-activated)
96     k_r         =  7.0    # mV  --- activation slope (negative: opens on hyperpol.)
97     tau_r_fixed = 200.0   # ms  --- slow, voltage-independent time constant
98
99     tstep = float(dt)
100
101    # -- Fixed biophysical constants ------------------------------------------
102    nois_fact_obs = 0.0
103    C    = 1.0    # uF/cm^2
104    E_Na = 53.0   # mV
105    E_K  = -107.0 # mV
106
107    # -- Numerical helpers ----------------------------------------------------
108    def Exp(z):
109     # Numerically safe exponential --- prevent overflow at very negative z
110     return torch.where(
111      z < -5e2,
112      torch.exp(torch.full_like(z, -5e2)),
113      torch.exp(z)
```

```python
114      )  # same shape as z
115
116    def efun(z):
117     # Handles 0/0 limit of z / (exp(z) - 1) near z=0
118     return torch.where(
119      torch.abs(z) < 1e-4,
120      1 - z / 2,
121      z / (Exp(z) - 1)
122     )  # same shape as z
123
124    # -- Standard HH channel kinetics ----------------------------------------
125    def alpha_m(x):  # (batch_size,) -> (batch_size,)
126     v1 = x - Vt - 13.0
127     return 0.32 * efun(-0.25 * v1) / 0.25
128
129    def beta_m(x):   # (batch_size,) -> (batch_size,)
130     v1 = x - Vt - 40.0
131     return 0.28 * efun(0.2 * v1) / 0.2
132
133    def alpha_h(x):  # (batch_size,) -> (batch_size,)
134     v1 = x - Vt - 17.0
135     return 0.128 * Exp(-v1 / 18.0)
136
137    def beta_h(x):   # (batch_size,) -> (batch_size,)
138     v1 = x - Vt - 40.0
139     return 4.0 / (1 + Exp(-0.2 * v1))
140
141    def alpha_n(x):  # (batch_size,) -> (batch_size,)
142     v1 = x - Vt - 15.0
143     return 0.032 * efun(-0.2 * v1) / 0.2
144
145    def beta_n(x):   # (batch_size,) -> (batch_size,)
146     v1 = x - Vt - 10.0
147     return 0.5 * Exp(-v1 / 40)
148
149    def tau_x(alpha, beta):  # (batch_size,), (batch_size,) -> (batch_size,)
150     return 1.0 / (alpha + beta)
151
152    def inf_x(alpha, beta):  # (batch_size,), (batch_size,) -> (batch_size,)
153     return alpha / (alpha + beta)
154
155    # -- M-current (I_KM) gating: Boltzmann steady-state, constant tau --------
156    # p_inf = 1 / (1 + exp(-(V - V_half_p) / 10))
157    # Slope k=10 mV is standard for cortical M-current (Wang 1998)
158    def inf_p(x):  # (batch_size,) -> (batch_size,)
159     return 1.0 / (1.0 + Exp(-(x - V_half_p) / 10.0))
160
161    # Voltage-independent tau_p avoids spurious fast tracking during spikes.
162    # tau_p is directly the inferred parameter (batch_size,), already positive.
163
164    # -- Ih gating: hyperpolarisation-activated Boltzmann, fixed tau ----------
165    # r_inf = 1 / (1 + exp((V - V_half_r) / k_r))
166    # Note positive sign in exponent: more open at V << V_half_r = -80 mV
167    def inf_r(x):  # (batch_size,) -> (batch_size,)
168     return 1.0 / (1.0 + Exp((x - V_half_r) / k_r))
169
170    # tau_r is fixed scalar (200 ms) --- no per-batch variation needed
171
172    # -- State variable arrays -----------------------------------------------
173    V = torch.zeros((batch_size, time_steps), device=device)  # (batch_size, time_steps)
174    n = torch.zeros((batch_size, time_steps), device=device)  # (batch_size, time_steps)
175    m = torch.zeros((batch_size, time_steps), device=device)  # (batch_size, time_steps)
176    h = torch.zeros((batch_size, time_steps), device=device)  # (batch_size, time_steps)
177    p = torch.zeros((batch_size, time_steps), device=device)  # (batch_size, time_steps) M-current gate
178    r = torch.zeros((batch_size, time_steps), device=device)  # (batch_size, time_steps) Ih gate
179
180    # -- Initial conditions (steady state at initial voltage) ----------------
181    V_init = init_voltage.to(device)  # (batch_size,)
182    V[:, 0] = V_init                                        # (batch_size,)
183    n[:, 0] = inf_x(alpha_n(V[:, 0]), beta_n(V[:, 0]))      # (batch_size,)
184    m[:, 0] = inf_x(alpha_m(V[:, 0]), beta_m(V[:, 0]))      # (batch_size,)
185    h[:, 0] = inf_x(alpha_h(V[:, 0]), beta_h(V[:, 0]))      # (batch_size,)
186    p[:, 0] = inf_p(V[:, 0])                                # (batch_size,)
187    r[:, 0] = inf_r(V[:, 0])                                # (batch_size,)
188
189    # -- Exponential Euler time-integration loop -----------------------------
190    for i in range(1, time_steps):
191     V_prev = V[:, i - 1]  # (batch_size,)
```

```
192
193      # Standard HH gating rates
194      a_m, b_m = alpha_m(V_prev), beta_m(V_prev)   # (batch_size,), (batch_size,)
195      a_h, b_h = alpha_h(V_prev), beta_h(V_prev)   # (batch_size,), (batch_size,)
196      a_n, b_n = alpha_n(V_prev), beta_n(V_prev)   # (batch_size,), (batch_size,)
197
198      # M-current gate steady-state (tau is voltage-independent -> use tau_p directly)
199      inf_p_val = inf_p(V_prev)    # (batch_size,)
200
201      # Ih gate steady-state (tau is fixed scalar)
202      inf_r_val = inf_r(V_prev)    # (batch_size,)
203
204      # Effective conductances at previous time step
205      g_Na_eff  = (m[:, i-1] ** 3) * gbar_Na * h[:, i-1]  # (batch_size,)
206      g_K_eff   = (n[:, i-1] ** 4) * gbar_K              # (batch_size,)
207      g_KM_eff  = gbar_KM * p[:, i-1]                     # (batch_size,)
208      g_Ih_eff  = gbar_Ih * r[:, i-1]                     # (batch_size,)
209
210      # Effective inverse membrane time constant (total conductance / C)
211      tau_V_inv = (
212       g_Na_eff
213       + g_K_eff
214       + g_leak
215       + g_KM_eff    # M-current: outward K+, raises conductance
216       + g_Ih_eff    # Ih: inward mixed cation, raises conductance
217      ) / C  # (batch_size,)
218
219      # Voltage steady-state numerator (weighted reversal potentials + input)
220      V_inf = (
221       g_Na_eff * E_Na
222       + g_K_eff  * E_K
223       + g_leak   * E_leak
224       + g_KM_eff * E_K    # M-current drives V toward E_K (hyperpolarising)
225       + g_Ih_eff * E_h    # Ih drives V toward E_h ~ -30 mV (depolarising at rest)
226       + input_current[:, i-1]
227       + nois_fact * torch.randn(batch_size, generator=generator, device=device) / (tstep ** 0.5)
228      ) / (tau_V_inv * C)  # (batch_size,)
229
230      # Exponential Euler update for voltage
231      V[:, i] = V_inf + (V_prev - V_inf) * Exp(-tstep * tau_V_inv)        # (batch_size,)
232
233      # Standard HH gating variable updates
234      n[:, i] = inf_x(a_n, b_n) + (n[:, i-1] - inf_x(a_n, b_n)) * Exp(-tstep / tau_x(a_n, b_n))  # (
    ↪ batch_size,)
235      m[:, i] = inf_x(a_m, b_m) + (m[:, i-1] - inf_x(a_m, b_m)) * Exp(-tstep / tau_x(a_m, b_m))  # (
    ↪ batch_size,)
236      h[:, i] = inf_x(a_h, b_h) + (h[:, i-1] - inf_x(a_h, b_h)) * Exp(-tstep / tau_x(a_h, b_h))  # (
    ↪ batch_size,)
237
238      # M-current gate update (voltage-independent tau -> stable during spikes)
239      p[:, i] = inf_p_val + (p[:, i-1] - inf_p_val) * Exp(-tstep / tau_p)  # (batch_size,)
240
241      # Ih gate update (fixed scalar tau)
242      r[:, i] = inf_r_val + (r[:, i-1] - inf_r_val) * Exp(
243       torch.full_like(V_prev, -tstep / tau_r_fixed)
244      )  # (batch_size,)
245
246     # Return voltage traces (+ optional observation noise, currently zero)
247     return V + nois_fact_obs * torch.randn(
248      batch_size, time_steps, generator=generator, device=device
249     )  # (batch_size, time_steps)
```

[[ ## feedback ## ]]
null

[[ ## completed ## ]]

## H.2. Hodgkin-Huxley Experiment: LLM-History Feedback Step

This section presents a specific example of the LLM interactions in the feedback step. The messages sent to and received from the LLM are presented in the same format as the propagation step in App. H.1.

## H.3. Prompt

> **System**
>
> Your input fields are:
> 1. 'system_description' (str): System description
> 2. 'simulator_code' (str): Simulator code
> 3. 'performance_metrics' (str): Performance metrics
> Your output fields are:
> 1. 'feedback' (ProvideFeedbackOutput): Improvement feedback
> All interactions will be structured in the following way, with the appropriate values filled in.
>
> [[ ## system_description ## ]]
> {system_description}
>
> [[ ## simulator_code ## ]]
> {simulator_code}
>
> [[ ## performance_metrics ## ]]
> {performance_metrics}
>
> [[ ## feedback ## ]]
> {feedback} # note: the value you produce must adhere to the JSON schema: {"type": "object", "$defs": {"Issue": {"type": "object", "properties": {"description": {"type": "string", "description": "Issue description", "title": "Description"}, "location": {"anyOf": [{"type": "string"}, {"type": "null"}], "default": null, "description": "Code location", "title": "Location"}, "severity": {"type": "string", "description": "Severity level", "enum": ["critical", "major", "minor", "suggestion"], "title": "Severity"}, "suggestion": {"type": "string", "description": "Fix suggestion", "title": "Suggestion"}}, "required": ["description", "severity", "suggestion"], "title": "Issue"}}, "properties": {"issues": {"type": "array", "description": "Issues with fixes", "items": {"$ref": "#/$defs/Issue"}, "maxItems": 2, "title": "Issues"}, "main_diagnosis": {"type": "string", "description": "Main issue and location", "title": "Main Diagnosis"}}, "required": ["main_diagnosis", "issues"], "title": "ProvideFeedbackOutput"}
>
> [[ ## completed ## ]]
> In adhering to this structure, your objective is:
> Analyze evaluation metrics to diagnose failure modes, then provide code improvement feedback.

> **User**
>
> [[ ## system_description ## ]]
> # HODGKIN-HUXLEY NEURON SIMULATOR WITH MISSING ION CHANNELS
>
> ## OBJECTIVE
> Extend an existing Hodgkin-Huxley neuron simulator by identifying and implementing missing ion channels necessary to reduce discrepancies between simulations and experimental voltage recordings. The goal is to improve agreement across multiple electrophysiological summary statistics while maintaining model parsimony.
>
> ## BASE MODEL
> Classic Hodgkin-Huxley formulation with standard three-gating-variable structure:
> - **Leak current**: conductance g_leak, reversal potential E_leak
> - **Sodium ($Na^+$) current**: conductance gNa, gating variables m (activation) and h (inactivation), reversal E_Na
> - **Potassium ($K^+$) current**: conductance gK, gating variable n (activation), reversal E_K
> - **Dynamics**: First-order gating kinetics with voltage-dependent time constants and steady states
> - **Input**: Externally applied current I_inj(t) with optional stochastic components
> - **Stochasticity**: Optional random seed for reproducible simulations
>
> ## EXTENSIBILITY
> Two pre-allocated channel slots (X1, X2) are available for additional mechanisms.
> Each slot provides two tunable parameters (see signature_description for details).
> Channels beyond these two slots must use fixed parameters, not inferred ones.

## EVALUATION METRICS
Model performance assessed on held-out voltage recordings via these summary statistics:
1. Number of spikes during stimulation
2. Mean resting potential (pre-stimulation)
3. Standard deviation of resting potential
4. Mean voltage during stimulation
5. Variance of voltage
6. Skewness of voltage
7. Kurtosis of voltage

## DATA CONTEXT
Intracellular voltage recordings from a single neuron exhibiting tonic spiking
behavior across different stimulus intensities.

**Important data characteristics**:
- No burst firing patterns (no clusters of rapid spikes separated by quiescence)
- No prolonged or sustained high-frequency firing (no sustained high-frequency firing)
- Regular, tonic spiking activity during periods of input current stimulation
- Non-spiking (quiescent) behavior in the absence of input current
- Evenly-spaced action potentials without burst patterns during the activation period

Therefore, avoid adding channels specifically designed to produce bursting or sustained
high-frequency dynamics.

[[ ## simulator_code ## ]]

```python
import torch
import torch.nn as nn

class DiscoveredSimulator(nn.Module):
  def __init__(self):
   super(DiscoveredSimulator, self).__init__()
   return

  def forward(
   self,
   init_voltage: float,
   input_current: torch.Tensor,
   dt: float,
   t: torch.Tensor,
   params: torch.Tensor,
   seed=None,
  ):
   """
   Hodgkin-Huxley neuron extended with:
    1. M-current (I_KM): slow non-inactivating K+ current using X1 slot
      - Provides spike-frequency adaptation, stabilises tonic firing
      - Uses voltage-independent tau (more stable during spikes)
      - gbar_KM = params[:,6], V_half_p = -params[:,8], tau_p = params[:,9]

    2. Ih current (HCN channel): hyperpolarisation-activated cation current using X2 slot
      - Opens at hyperpolarised voltages (~-80 mV and below)
      - Reversal E_h ~= -30 mV (mixed Na+/K+), depolarising at rest
      - Corrects resting potential mean, SD, and voltage distribution shape
      - Fixed kinetics (half-activation -80 mV, tau 200 ms); only gbar_Ih inferred
      - gbar_Ih = params[:,7], all other Ih parameters are fixed

   Args:
    init_voltage: torch.Tensor: (batch_size,)           initial voltage [mV]
    input_current: torch.Tensor: (batch_size, time_steps) injected current [uA/cm2]
    dt: float                                            time step [ms]
    t: torch.Tensor: (time_steps,)                       time array [ms]
    params: torch.Tensor: (batch_size, 10)               biophysical parameters
    seed: optional random seed

   Returns:
    V: torch.Tensor: (batch_size, time_steps)            voltage traces [mV]
   """
   device = params.device

   # Set up random generator
   if seed is not None:
    generator = torch.Generator(device=device)
```

```
 49     generator.manual_seed(seed)
 50   else:
 51     generator = torch.Generator(device=device)
 52
 53   batch_size = params.shape[0]    # int
 54   time_steps = t.shape[0]         # int
 55
 56   # -- Base HH parameters -----------------------------------------------
 57   gbar_Na  = params[:, 0].float()    # (batch_size,)  mS/cm^2
 58   gbar_K   = params[:, 1].float()    # (batch_size,)  mS/cm^2
 59   g_leak   = params[:, 2].float()    # (batch_size,)  mS/cm^2
 60   E_leak   = -params[:, 3].float()   # (batch_size,)  mV  (negative applied)
 61   Vt       = -params[:, 4].float()   # (batch_size,)  mV  (negative applied)
 62   nois_fact = params[:, 5].float()   # (batch_size,)  unitless
 63
 64   # -- X1 slot: M-current (I_KM, slow non-inactivating K+) ----------------
 65   # Physiological rationale:
 66   #   M-current activates slowly at subthreshold depolarised voltages (~-35 mV),
 67   #   providing an outward K+ current that limits repetitive firing rate and
 68   #   produces spike-frequency adaptation without bursting or quiescence.
 69   # Parameter mapping:
 70   #   gbar_KM   = params[:,6] in [1e-4, 10] mS/cm^2
 71   #   V_half_p  = -params[:,8] in [-150, -1e-4] mV  (typically ~-35 mV)
 72   #   tau_p     = params[:,9] in [1e-4, 3000] ms  (typically 50-500 ms)
 73   # Note: voltage-independent tau avoids instability during fast action potentials
 74   gbar_KM = params[:, 6].float()    # (batch_size,)  mS/cm^2
 75   V_half_p = -params[:, 8].float()  # (batch_size,)  mV
 76   tau_p    = params[:, 9].float()   # (batch_size,)  ms --- voltage-independent for stability
 77
 78   # -- X2 slot: Ih current (HCN, hyperpolarisation-activated cation) --------
 79   # Physiological rationale:
 80   #   Ih activates at hyperpolarised voltages (< -60 mV), carries inward
 81   #   Na+/K+ current with reversal ~-30 mV, depolarising the cell toward
 82   #   rest. This sets a tonic inward "sag" current that:
 83   #     - Raises mean resting potential slightly above pure K+ equilibrium
 84   #     - Increases resting SD (voltage fluctuations due to channel noise)
 85   #     - Shapes subthreshold distribution (skewness, kurtosis)
 86   #   It does NOT promote bursting --- it acts as a stabilising "pacemaker"
 87   #   current opposing excessive hyperpolarisation.
 88   # Parameter mapping:
 89   #   gbar_Ih = params[:,7] in [1e-4, 120] mS/cm^2 --- only inferred parameter
 90   #   All kinetic parameters fixed from literature
 91   gbar_Ih = params[:, 7].float()    # (batch_size,)  mS/cm^2
 92
 93   # Fixed Ih kinetic constants (from Magee 1998, Koch 1999)
 94   E_h        = -30.0   # mV  --- mixed Na+/K+ cation reversal
 95   V_half_r   = -80.0   # mV  --- half-activation (hyperpolarisation-activated)
 96   k_r        = 7.0     # mV  --- activation slope (negative: opens on hyperpol.)
 97   tau_r_fixed = 200.0  # ms  --- slow, voltage-independent time constant
 98
 99   tstep = float(dt)
100
101   # -- Fixed biophysical constants ------------------------------------------
102   nois_fact_obs = 0.0
103   C    = 1.0   # uF/cm^2
104   E_Na = 53.0  # mV
105   E_K  = -107.0 # mV
106
107   # -- Numerical helpers ----------------------------------------------------
108   def Exp(z):
109     # Numerically safe exponential --- prevent overflow at very negative z
110     return torch.where(
111       z < -5e2,
112       torch.exp(torch.full_like(z, -5e2)),
113       torch.exp(z)
114     )  # same shape as z
115
116   def efun(z):
117     # Handles 0/0 limit of z / (exp(z) - 1) near z=0
118     return torch.where(
119       torch.abs(z) < 1e-4,
120       1 - z / 2,
121       z / (Exp(z) - 1)
122     )  # same shape as z
123
124   # -- Standard HH channel kinetics -----------------------------------------
125   def alpha_m(x):  # (batch_size,) -> (batch_size,)
126     v1 = x - Vt - 13.0
```

```
127      return 0.32 * efun(-0.25 * v1) / 0.25
128
129    def beta_m(x):    # (batch_size,) -> (batch_size,)
130      v1 = x - Vt - 40.0
131      return 0.28 * efun(0.2 * v1) / 0.2
132
133    def alpha_h(x):   # (batch_size,) -> (batch_size,)
134      v1 = x - Vt - 17.0
135      return 0.128 * Exp(-v1 / 18.0)
136
137    def beta_h(x):    # (batch_size,) -> (batch_size,)
138      v1 = x - Vt - 40.0
139      return 4.0 / (1 + Exp(-0.2 * v1))
140
141    def alpha_n(x):   # (batch_size,) -> (batch_size,)
142      v1 = x - Vt - 15.0
143      return 0.032 * efun(-0.2 * v1) / 0.2
144
145    def beta_n(x):    # (batch_size,) -> (batch_size,)
146      v1 = x - Vt - 10.0
147      return 0.5 * Exp(-v1 / 40)
148
149    def tau_x(alpha, beta):  # (batch_size,), (batch_size,) -> (batch_size,)
150      return 1.0 / (alpha + beta)
151
152    def inf_x(alpha, beta):  # (batch_size,), (batch_size,) -> (batch_size,)
153      return alpha / (alpha + beta)
154
155    # -- M-current (I_KM) gating: Boltzmann steady-state, constant tau --------
156    # p_inf = 1 / (1 + exp(-(V - V_half_p) / 10))
157    # Slope k=10 mV is standard for cortical M-current (Wang 1998)
158    def inf_p(x):   # (batch_size,) -> (batch_size,)
159      return 1.0 / (1.0 + Exp(-(x - V_half_p) / 10.0))
160
161    # Voltage-independent tau_p avoids spurious fast tracking during spikes.
162    # tau_p is directly the inferred parameter (batch_size,), already positive.
163
164    # -- Ih gating: hyperpolarisation-activated Boltzmann, fixed tau ----------
165    # r_inf = 1 / (1 + exp((V - V_half_r) / k_r))
166    # Note positive sign in exponent: more open at V << V_half_r = -80 mV
167    def inf_r(x):   # (batch_size,) -> (batch_size,)
168      return 1.0 / (1.0 + Exp((x - V_half_r) / k_r))
169
170    # tau_r is fixed scalar (200 ms) --- no per-batch variation needed
171
172    # -- State variable arrays -------------------------------------------
173    V = torch.zeros((batch_size, time_steps), device=device)  # (batch_size, time_steps)
174    n = torch.zeros((batch_size, time_steps), device=device)  # (batch_size, time_steps)
175    m = torch.zeros((batch_size, time_steps), device=device)  # (batch_size, time_steps)
176    h = torch.zeros((batch_size, time_steps), device=device)  # (batch_size, time_steps)
177    p = torch.zeros((batch_size, time_steps), device=device)  # (batch_size, time_steps) M-current gate
178    r = torch.zeros((batch_size, time_steps), device=device)  # (batch_size, time_steps) Ih gate
179
180    # -- Initial conditions (steady state at initial voltage) ------------------
181    V_init = init_voltage.to(device)  # (batch_size,)
182    V[:, 0] = V_init                                             # (batch_size,)
183    n[:, 0] = inf_x(alpha_n(V[:, 0]), beta_n(V[:, 0]))          # (batch_size,)
184    m[:, 0] = inf_x(alpha_m(V[:, 0]), beta_m(V[:, 0]))          # (batch_size,)
185    h[:, 0] = inf_x(alpha_h(V[:, 0]), beta_h(V[:, 0]))          # (batch_size,)
186    p[:, 0] = inf_p(V[:, 0])                                     # (batch_size,)
187    r[:, 0] = inf_r(V[:, 0])                                     # (batch_size,)
188
189    # -- Exponential Euler time-integration loop -------------------------------
190    for i in range(1, time_steps):
191      V_prev = V[:, i - 1]  # (batch_size,)
192
193      # Standard HH gating rates
194      a_m, b_m = alpha_m(V_prev), beta_m(V_prev)   # (batch_size,), (batch_size,)
195      a_h, b_h = alpha_h(V_prev), beta_h(V_prev)   # (batch_size,), (batch_size,)
196      a_n, b_n = alpha_n(V_prev), beta_n(V_prev)   # (batch_size,), (batch_size,)
197
198      # M-current gate steady-state (tau is voltage-independent -> use tau_p directly)
199      inf_p_val = inf_p(V_prev)    # (batch_size,)
200
201      # Ih gate steady-state (tau is fixed scalar)
202      inf_r_val = inf_r(V_prev)    # (batch_size,)
203
204      # Effective conductances at previous time step
```

```
205    g_Na_eff  = (m[:, i-1] ** 3) * gbar_Na * h[:, i-1]   # (batch_size,)
206    g_K_eff   = (n[:, i-1] ** 4) * gbar_K                # (batch_size,)
207    g_KM_eff  = gbar_KM * p[:, i-1]                      # (batch_size,)
208    g_Ih_eff  = gbar_Ih * r[:, i-1]                      # (batch_size,)
209
210    # Effective inverse membrane time constant (total conductance / C)
211    tau_V_inv = (
212     g_Na_eff
213     + g_K_eff
214     + g_leak
215     + g_KM_eff   # M-current: outward K+, raises conductance
216     + g_Ih_eff   # Ih: inward mixed cation, raises conductance
217    ) / C  # (batch_size,)
218
219    # Voltage steady-state numerator (weighted reversal potentials + input)
220    V_inf = (
221     g_Na_eff * E_Na
222     + g_K_eff  * E_K
223     + g_leak   * E_leak
224     + g_KM_eff * E_K     # M-current drives V toward E_K (hyperpolarising)
225     + g_Ih_eff * E_h     # Ih drives V toward E_h ~ -30 mV (depolarising at rest)
226     + input_current[:, i-1]
227     + nois_fact * torch.randn(batch_size, generator=generator, device=device) / (tstep ** 0.5)
228    ) / (tau_V_inv * C)  # (batch_size,)
229
230    # Exponential Euler update for voltage
231    V[:, i] = V_inf + (V_prev - V_inf) * Exp(-tstep * tau_V_inv)        # (batch_size,)
232
233    # Standard HH gating variable updates
234    n[:, i] = inf_x(a_n, b_n) + (n[:, i-1] - inf_x(a_n, b_n)) * Exp(-tstep / tau_x(a_n, b_n))  # (
     ↪ batch_size,)
235    m[:, i] = inf_x(a_m, b_m) + (m[:, i-1] - inf_x(a_m, b_m)) * Exp(-tstep / tau_x(a_m, b_m))  # (
     ↪ batch_size,)
236    h[:, i] = inf_x(a_h, b_h) + (h[:, i-1] - inf_x(a_h, b_h)) * Exp(-tstep / tau_x(a_h, b_h))  # (
     ↪ batch_size,)
237
238    # M-current gate update (voltage-independent tau -> stable during spikes)
239    p[:, i] = inf_p_val + (p[:, i-1] - inf_p_val) * Exp(-tstep / tau_p)  # (batch_size,)
240
241    # Ih gate update (fixed scalar tau)
242    r[:, i] = inf_r_val + (r[:, i-1] - inf_r_val) * Exp(
243     torch.full_like(V_prev, -tstep / tau_r_fixed)
244    )  # (batch_size,)
245
246   # Return voltage traces (+ optional observation noise, currently zero)
247   return V + nois_fact_obs * torch.randn(
248    batch_size, time_steps, generator=generator, device=device
249   )  # (batch_size, time_steps)
```

[[ ## performance_metrics ## ]]
Metrics: neg_avg_log_marginal_NLE: 35.3, neg_log_marginal_NLE: 353, timeout_evidence_estimation: 0

Respond with the corresponding output fields, starting with the field '[[ ## feedback ## ]]' (must be formatted as a valid Python ProvideFeedbackOutput), and then ending with the marker for '[[ ## completed ## ]]'.

## H.3.1. RESPONSE

**Output**

[[ ## feedback ## ]]

{

    "**main_diagnosis**": "The primary failure is a systematically over-hyperpolarized potassium reversal potential (E_K = -107 mV, hardcoded), which is far below the physiological range for cortical neurons (~-90 mV). This single constant distorts the after-hyperpolarization depth for both the standard K+ current and the M-current, biasing spike count, mean stimulation voltage, skewness, and kurtosis simultaneously. A secondary issue is that the Ih gating parameters allow unrealistically strong activation near rest: with V_half_r = -80 mV and k_r = 7 mV, ~19% of Ih channels are open at -70 mV rest, and the prior permits gbar_Ih up to 120 mS/cm$^2$ (physiological range is 0.05–5 mS/cm$^2$), causing gbar_Ih to dominate the conductance balance and push mean resting potential and voltage variance far from target.",

"**issues**":
[
    {
        "**description**": "E_K = -107.0 mV is too hyperpolarized. Physiological E_K for cortical neurons is approximately -90 mV. The excessively negative value deepens the AHP after every spike, reduces firing rate by prolonging inter-spike intervals, pulls mean stimulation-period voltage too negative (metric 4), and skews the voltage distribution toward hyperpolarized values (metrics 6–7). Because E_K also sets the M-current reversal, this error is amplified once gbar_KM is inferred.",
        "**location**": "Line: 'E_K = -107.0 # mV'",
        "**severity**": "critical",
        "**suggestion**": "Change E_K to -90.0 mV, which is the standard physiological value for cortical/hippocampal neurons used in most HH extensions (e.g., Pospischil et al. 2008). If the target data suggests a slightly different value, keep it as a fixed constant in the range [-80, -95] mV rather than leaving it at -107."
    },
    {
        "**description**": "The Ih current is over-powered at rest. Two compounding problems: (1) k_r = 7 mV with V_half_r = -80 mV means ~19% of Ih channels are already open at a typical resting voltage of -70 mV, providing significant tonic depolarizing current even without strong hyperpolarisation. (2) The prior upper bound for gbar_Ih (120 mS/cm$^2$) is ~25–100× the physiological range (0.05–5 mS/cm$^2$), so the inference can select values that make Ih the dominant conductance, dramatically raising resting potential SD and variance (metrics 3, 5) to unrealistic levels while also distorting skewness and kurtosis.",
        "**location**": "Lines: 'gbar_Ih = params[:, 7]', 'k_r = 7.0', 'V_half_r = -80.0'",
        "**severity**": "major",
        "**suggestion**": "Shift the half-activation to V_half_r = -90 mV (deeper hyperpolarisation required to open channels), which reduces Ih activation at rest from ~19% to ~4%. Also reduce the prior upper bound for gbar_Ih from 120 to 5 mS/cm$^2$ to keep Ih in the physiological regime. Alternatively, increase k_r to 10–12 mV to soften the transition and reduce resting activation. These changes together will prevent Ih from dominating resting conductance while preserving its role in correcting subthreshold distribution shape."
    }
]
}
[[ ## completed ## ]]

