# OpenReview forum: "A Probabilistic Framework for LLM-Based Model Discovery"
_ICML.cc/2026/Conference — ICML 2026 regular_

### Official Review · Reviewer_zDTX · 2026-03-11

**Soundness:** 3
**Presentation:** 2
**Significance:** 3
**Originality:** 3
**Overall Recommendation:** 5
**Confidence:** 4

**Summary:**

The paper proposes a framework for automated scientific model discovery using LLMs. It formulates the problem as Bayesian inference over mechanistic models, targeting the posterior over model programs $p(m|x_{0})$. The prior over the space of model programs is implicitly defined as the LLMs generative process. The proposed algorithm uses Sequential Monte Carlo where candidate models are proposed by an LLM and weighted using an approximate marginal likelihood. This allows the system to maintain a population of plausible mechanistic models rather than selecting a single best model.

**Compliance With Llm Reviewing Policy:**

Affirmed.

**Final Justification:**

The authors have addressed most of my questions through the ablation experiments, better understanding of the compute and exploration of the algorithm from a seed-model.

**Key Questions For Authors:**

1. The paper would benefit from a concrete end-to-end example to aid understanding. I think this should be fairly simple to provide.

the initial prompt -> the initial model -> an LLM-generated modification -> the resulting evaluation feedback -> the next iteration of the prompt.

2. Because the algorithm starts from one seed model, exploration of model space depends strongly on the initial model (m_{0}), and it may be useful to understand how far the LLM edits go in terms of proposing modifications to this initial model. The authors say that the posterior mass quickly concentrate on plausible models but it is important to understand how far the seed model was and the ability of the LLM proposals to deviate from the seed model.

3. it would be helpful to understand how robust the algorithm is to errors in the likelihood approximation, as the weights are tied to it.

**Limitations:**

Yes, limitations have been discussed.

**Strengths And Weaknesses:**

Strengths:

- The idea of fashioning model discovery as posterior inference over a space of model programs is statistically well-reasoned and superior to several other heuristic approaches.
- It addresses a topic of substantial interest. LLMs are increasingly being used to generate code, propose hypotheses, and interact with simulation environments, developing  frameworks for integrating LLM proposals with statistical inference is a worthwhile direction to pursue. Automated model discovery is going to be even more topical in the near future.
- The experiments shown impress upon the key fact that in these examples, the posterior quickly concentrates on a small number of plausible models, which supports the claim that the weighting and resampling procedure effectively filters out poor model proposals.

Weaknesses:
- Missing clear-cut examples of LLM interaction. The propagation step relies on querying an LLM to generate a revised model if I understand correctly; this is conditioned on previous models and feedback. However this exact interaction protocol is not specified..making the whole algorithm opaque. I guess the authors want to position the paper more in-line with a probabilistic framework of interacting with LLMs, but providing a clear example of the prompt, the resulting LLM-generated modification, and how the feedback is incorporated in subsequent iterations would greatly improve the reproducibility of the method.

- Unclear separation between framework and prompt engineering. It's difficult to disentangle improvements due to the SMC formulation vs. the prompt engineering.

- The likelihood estimation step (for weighting) is extremely heavy, for each candidate model you must simulate dataset from the model particles, train or evaluate a density estimator as a likelihood surrogate, so the computational cost is roughly like O(particles×simulations)..this can explode quickly.

- Rewrite, too confusing to understand (lines 212-216):

"The context $c_{k-1}^{a^{n}_{k}}$ summarizes the recent ancestry of the particle and its performance and is generated via an auxiliary LLM call based on the particle’s weight. To control computational cost and prompt length, only a limited number of the most recent ancestry elements are included"

---

> ### Author Rebuttal · Authors · 2026-03-31
>
> We appreciate the reviewer noting the formulation is “statistically well-reasoned and superior to heuristic approaches”, that it “addresses a topic of substantial interest”, and that “the experiments shown impress” on key aspects of the proposed inference procedure. We address concerns (W1-4) and questions (Q1-3) below. We refer to new tables and figures at [link](https://tinyurl.com/modelsmc).
>
> **W1/Q1. End-to-end LLM prompting example.**
> A concrete end-to-end example is indeed missing. Starting from a zero-shot prompt (system instruction, task description, base simulator), the LLM generates an initial simulator, which is weighted using likelihood estimation to obtain performance metrics. A second LLM call diagnoses these metrics and returns structured feedback, which is then incorporated as a few-shot demonstration in the next iteration's prompt, closing the loop (Fig. 6). We will add the figure illustrating this flow to the manuscript, include representative prompt examples, and release prompts alongside our code upon acceptance.
>
> **W2/Q3. Ablation study: likelihood robustness and prompt engineering.**
> We address both concerns through ablations. We conducted the following ablation on the HH tasks (3 seeds each): N=1 particles (no ModelSMC framework), MSE vs. marginal likelihood weighting, LLM backends, simulation budgets, feedback variants, minimal prompts.
>
> Most directly relevant (Tab. 2):
> - the simulation budget analysis degrades the likelihood approximation via fewer simulations. We observe that reducing the from 5000 simulations to as low as 200 yields comparable final performance, indicating that the framework is reasonably robust to errors in likelihood estimation.
> - the minimal prompt variant removes all details while keeping the full SMC framework, directly isolating prompt engineering. Results indicate that even a minimal prompt produces reasonable results. Crucially, any improvements can be incorporated into ModelSMC while preserving its probabilistic posterior interpretation.
> We will include the full ablation study in the final manuscript.
>
> **W3. Exploration from seed model.**
> The seed model in ModelSMC can range from an empty skeleton (SIR) to an informed starting point reflecting existing domain knowledge (kidney, HH). In the latter case, the scientist’s prior knowledge implicitly constrains the explored regions of model space through conditioning on the base model in the LLM. We consider this a feature rather than a limitation since it allows domain expertise to guide exploration without hard-coding model structure.
>
> Quantifying exploration via a program-space distance metric is difficult, there exists no agreed-upon metric. Instead, we provide a posterior-level analysis on the HH task. Specifically, we used an LLM to extract a taxonomy of ion-channel model variants from the discovered simulators, then classified all 1,285 particles (across 10 seeds) into subtypes (e.g. I_M, I_A, I_NaP families) (Fig. 3, 4). I_M variants systematically achieve better posterior weights than I_A or I_NaP variants in 7/10 seed (Fig. 5) . This demonstrates that ModelSMC meaningfully explores model space and arrives at reproducible conclusions to rule in or out specific modelling hypotheses. This would not be possible if proposals remained close to the seed. We will enlarge this discussion in Section 4.
>
> **Q2. Computational cost.**
> The total simulation budget for a naive sequential implementation is $\mathcal{O}(N\times S\times K)$ for $N$ particles, $S$ simulations and $K$ iterations. Since particles are independent, simulation and likelihood weighting can be parallelized across particles, reducing wall-clock time to $\mathcal{O}(S\times K)$ with sufficient compute.  We collect the duration of different steps of ModelSMC for the HH and SIR task (Tab. 3) and observe that while the time needed for the likelihood estimation and sampling from the discovered model indeed dominate, the purely LLM-based contributions are also non-negligible. Simulation cost for particle weighting is not unique to ModelSMC. Any LLM-based discovery method that evaluates a candidate model score must estimate parameters. ModelSMC uses the simulation budget to construct a principled posterior rather than just a scalar score, yielding richer scientific conclusions at similar cost.
>
> **W4. Presentation fix.**
> We will improve the description of context generation (lines 212-216). This paragraph concerns the textual LLM feedback used in future LLM calls improving the model via few-shot learning. This will be clearer with the end-to-end example (Fig. 6).
>
> We hope these clarifications and experiments address the concerns, and would be grateful if you would consider raising your score.

---

> > ### Author Rebuttal · Reviewer_zDTX · 2026-04-04
> >
> > Thank you for the response. I am happy to raise my score 4->5 in light of the supplementary experiments and the promise of an end-to-end example in the final paper.

---

### Official Review · Reviewer_rd29 · 2026-03-12

**Soundness:** 3
**Presentation:** 2
**Significance:** 2
**Originality:** 3
**Overall Recommendation:** 4
**Confidence:** 3

**Summary:**

This paper discusses the model discovery problem, which was cast into a Bayesian probabilistic inference. SMC is used to approximate the target distributions using particles. Specifically, the bootstrap particle filter is employed. Each candidate model works as a particle, so people may also (although not necessarily) read this paper from the perspective of Bayesian model averaging. LLMs are suggested to propagate the models, that is, to generate new candidate models using the existing candidate models (i.e., acting as model transition). The NLE technique is employed to obtain an estimate of the unknown likelihood (i.e., acting as model “measurement” or model “evaluation”).

**Compliance With Llm Reviewing Policy:**

Affirmed.

**Key Questions For Authors:**

I have the following specific questions or comments.

1.	In Introduction (Right column, Page 1), model $m$ is introduced without clarification. It was not mathematically defined until Subsection 3.2 (Right column, Page 3).
2.	It is not clear why the bootstrap particle filter is preferred without justification, as many practical particle filters exist. For example, in some particle filters, people do propagate first, and then do re-weighting using Bayes’ rule with priors and likelihoods, and then resample using systematic or stratified resampling techniques, and finally weight normalization. I am not sure if the bootstrap particle filter is the only choice for some reasons.
3.	NPE is used before its first definition (Left column, Page 5).
4.	In Theorem 3.1, $q(m’|m)$ is mentioned but not used later. Also, as the paper admitted, the exact likelihood evaluation is not available, so the NLE is used. But in Theorem 3.1, an exact likelihood evaluation is assumed. To me, Theorem 3.1 is not necessarily for this practical paper. In the theoretical literature of particle filters, weak convergence of particle filters is indeed studied in the sense of (7). However, as a practical paper, such results are not really instructive to the contents of this paper: the first concern is whether the likelihood is exact; the second concern is whether N can really be infinite. To me, Theorem 3.1 (which introduces a convergence result of particle filters in the asymptotic regime and an idealized scenario) brings no much insights to the contents of this practical paper. When a theorem is used in a practical paper, it usually justifies that the assumptions or approximations in the paper are reasonable or good enough. If the theorem can claim something even under some errors of likelihood estimation and N is finite, then the theorem would be insightful to the contents of this paper, in my opinion.
5.	In Algorithm 1, does it mean at each iteration, the model size increases with N’? In other words, at the end of the final round K, you have in total N + K * N’ candidate models? If so, is computation a practical issue if K is large? Please justify why not keeping the number of candidate models unchanged (e.g., by using systematic or stratified resampling techniques). In other words, what benefits can we have if we increase the model size at each iteration?
6.	In practice, I am not sure whether the marginal distributions $p(theta)$ and $p(c)$ are always available, so we can safely and easily construct a synthetic dataset for NLE training; see texts below (6). Please justify for various applications.

**Limitations:**

Yes

**Strengths And Weaknesses:**

Strengths: Casting model discovery as a Bayesian probabilistic inference is new to this area.

Weakness: But the embedding is trivial, the authors might need to deepen their thoughts

---

> ### Author Rebuttal · Authors · 2026-03-31
>
> We thank the reviewer for their positive evaluation, recognizing that "casting model discovery as a Bayesian probabilistic inference is new to this area" and address the raised questions (Q1-6) below. We refer to new tables and figures found in this [link](https://tinyurl.com/modelsmc).
>
> **Q2. Bootstrap particle filter.**
> We agree that the specific choice of the PF is not the main part of the paper, rather the probabilistic framework for model discovery. The bootstrap PF is the simplest possible choice and well-suited as a proof of concept for our probabilistic framework. More advanced variants are of course possible. We have additionally switched to systematic resampling (Doucet & Johansen 2008), which is the most widely adopted resampling strategy. A comprehensive survey of which PF variant is best suited for model discovery is an interesting direction but beyond the scope of this paper.
>
> **Q4. Theorem 3.1 relevance and assumptions.**
> We agree that in a practical paper, a theorem limited to exact likelihoods as $N\to\infty$ cannot serve as a performance guarantee. That is not our intent.
>
> The theorem is a proof of concept: it shows that, in an idealized setting, the probabilistic framing of model discovery admits formal theoretical analysis at all. Prior approaches to automated model discovery are more heuristic pipelines with no natural extension for such analysis. By recasting model discovery as inference, we inherit the theoretical properties of the SMC literature (Del Moral 2004), and Theorem 3.1 establishes the idealized reference distribution against which practical approximation can be characterized. Finite-sample bounds under NLE approximation error are a natural extension that our framing uniquely enables.
>
> We will revise Section 3.4 to clarify this role. Specifically, to characterize the idealized reference sampler, to state clearly how the practical algorithm departs from it, and to frame gaps as directions for future work.
>
> **Q5. Growing particle pool.**
> The original Algorithm 1 accumulates particles across iterations, resulting in a growing pool size of $K$ per round, up to a maximum of 50 particles. To resolve this issue and more closely align ModelSMC with standard SMC algorithms, we have now reformulated ModelSMC to maintain a fixed pool of $N$ particles throughout. This includes the following modifications:
> - Resampling. Compute the effective sample size (ESS). Apply systematic resampling when ESS falls below a threshold.
> - Propagation. Each particle is either copied (with probability $\alpha$) or replaced by an LLM-propagated version (with probability $1-\alpha$)
> - Weighting. Updated via the likelihood, consistent with standard importance weighting.
>
> We have rerun all experiments under the revised algorithm. Results are consistent with the original submission (Tab. 4), and will update the manuscript accordingly. In practice, the difference in the algorithm is not substantial resulting in the same conclusions from the results.
>
> **Q6. Availability of marginals $p(\theta)$ and $p(c)$.**
> $p(\theta)$ is the prior distribution over the parameter of the discovered models. Since parameters vary across models, this prior is model- and task-specific, and is defined by the scientist using domain knowledge to identify meaningful parameter ranges, defaulting to uniform priors over physically plausible bounds, or incorporating knowledge from literature. Concrete definitions for all three tasks are known from previous work and defined in Appendix F.2-F.4. We will make this explicit in the main text and add a short practical guide on prior choice.
>
> For $p(c)$: the experimental conditions $c$ (e.g., the stimulus current and initial voltage in Hodgkin-Huxley) are observable by definition. Therefore, a categorical distribution over observed training value is always available as a minimal choice. Additional information about the experimental setting can provide more sophisticated distributions. We will add a brief discussion of both points to the manuscript.
>
> **Q1, Q3. Notation and presentation fix.**
> We thank the reviewer for catching the presentation issues. We will ensure that all abbreviations and notations are formally introduced before it is referenced in the revised manuscript.
>
> We hope these clarifications and revisions address your concerns. We would be happy to answer any further questions or provide additional clarification, and would appreciate the reviewer considering raising their score.

---

> > ### Author Rebuttal · Reviewer_rd29 · 2026-04-05
> >
> > Thank you for your detailed responses. I will keep my score.

---

### Official Review · Reviewer_4Td6 · 2026-03-13

**Soundness:** 3
**Presentation:** 3
**Significance:** 2
**Originality:** 2
**Overall Recommendation:** 4
**Confidence:** 4

**Summary:**

This work formulates LLM-based scientific model (in the form of executable code) discovery as probabilistic inference. The proposed ModelSMC uses a SMC style for model evolution, candidate models are treated as particles and the proposal distribution of SMC is an LLM.  The method is tested empirically on an LLM-free synthetic validation task, a kidney pharmacology simulator, and a Hodgkin–Huxley neuron model, with comparisons to few competing algorithms.

**Compliance With Llm Reviewing Policy:**

Affirmed.

**Key Questions For Authors:**

1. Is it possible for the authors to provide a clearer empirical decomposition of where the gains come from? For example, how much benefit comes from population-based inference versus likelihood-based weighting versus text-feedback ancestry conditioning?

2. Given that FunSearch+ is unavailable for the kidney task and the single-particle version is closely related to the proposed method, could the authors justify more explicitly why the present baselines are enough to support the paper’s broader claims?

3. How stable are posterior conclusions across seeds and surrogate-likelihood choices? - the paper reports ten runs and confidence intervals for some metrics, but how consistent are the posterior high-weight regions across surrogate estimators, simulation budgets, or prompt variants?

**Limitations:**

The authors discuss several technical limitations, including computational cost, surrogate-likelihood bias, dependence on LLM proposal quality, and the lack of an explicit geometry over model space. That part is constructive and appropriate.

**Strengths And Weaknesses:**

Soundness.
The paper has a clear methodological core. Recasting model discovery as posterior inference in coding space is a nice idea, and the algorithm is described under standard SMC steps, which were well-established in the past two decades. The paper surrogate likelihoods and an implicit, context-dependent LLM proposal in actual implementation, that is very practical consideration. However, my main concern is: while the true proposal density is implicit and context-dependent, the exact importance correction is unavailable as the surrogate likelihood is used. This means the formal consistency result is more like a motivating analogy than a guarantee for the actual implemented algorithm. A second concern is that the competing algorithm comparisons remain relatively narrow, which makes the result as somewhat limited empirical picture. - it is a bit difficult to evaluate the proposed method amongst others in the field.

Presentation.
The paper is generally well written and well structured, easy to follow.


Significance.
The problem is important and should be of interests to a certain group of audience. The attempt to move from heuristic LLM-agent systems to a more systematic one is valuable. There are lots of open issues remain unsolved, the probabilistic framing itself could influence future work on uncertainty, posterior analysis, proposal steps and propagation. Regarding this paper, its practical impact is still somewhat uncertain because of computational expense and dependence on the LLM’s proposal quality and the form of models.

Originality.
The paper's idea is certainly contains originality, but more from reframing LLM-based proposal distribution within an SMC lens. So the novelty is more in synthesis and formalization than in a wholly new algorithmic mechanism.

---

> ### Author Rebuttal · Authors · 2026-03-31
>
> We thank the reviewer for their positive evaluation, recognizing the idea containing "originality", the problem as "important", the paper as "well written” and ”easy to follow”, and the move toward systematic model inference as "valuable". We address the raised concerns below (W1-2) and answer questions (Q1-3). We refer to new tables and figures found in this [link](https://tinyurl.com/modelsmc).
>
> **W1. Relevance of Theorem 3.1.**
> The theorem characterizes an idealized ModelSMC sampler, with exact likelihoods and a prior-matching proposal. Thus, it serves as a precise reference point against which the practical approximations can be understood. We document both approximate in more detail in the appendix:
> - Approximate prior matching (Appendix B, Remarks). The LLM proposal is conditioned on feedback and context, making exact importance weights intractable (Eq. B-11). Instead ModelSMC uses the likelihood alone as a practical approximation.
> - Surrogate likelihood (Appendix C, Remarks). Likelihoods are estimated with NLE rather than evaluated exactly, as discussed in Assumption (iv), Appendix C.
> We agree these points should be more prominent in the main text, and will revise Section 3.4 to state explicitly how the implemented algorithm departs from the idealized setting and what this implies for interpreting the theorem.
>
> **Q1. Ablation study.**
> We appreciate the precise decomposition suggested by the reviewer. Though we note that ModelSMC’s primary contribution is a probabilistic formulation of model discovery. This enables scientifically relevant posterior over model programs rather than point estimates maximising a held-out score. As a concrete illustration of what this posterior enables, Fig. 3 shows an analysis of the posterior to rule in or out certain modelling hypotheses. This conclusion is only possible because of ModelSMC’s posterior over programs rather than point estimates.
> We conducted the following ablation on the HH tasks (3 seeds each): N=1 particles (no ModelSMC framework), MSE vs. marginal likelihood weighting, LLM backends, simulation budgets, feedback variants, minimal prompts.
> Overall, our configuration for ModelSMC provides the strongest results (Tab. 2). Most directly relevant to your decomposition: The N=1 baseline isolates the population benefit, MSE vs marginal likelihood weighting isolates the benefit of principled likelihood weighting, and the feedback and minimal-prompt variants address the ancestry conditioning. Importantly, any performance improvement can be incorporated into ModelSMC as a drop-in enhancement, while the framework uniquely retains the ability to interpret results as a posterior over model programs to draw meaningful scientific conclusions.
> We will include the ablation study in the final manuscript.
>
> **W2/Q2. Baseline method justification.**
> We emphasize that, as far as we know, no existing model discovery method directly targets the problem that ModelSCM addresses, namely probabilistic inference over model programs and parameters. Thus, a direct and fair benchmark is unavailable. The two baselines were chosen to represent the two dominant paradigms for LLM-based model discovery: FunSearch+ represents evolutionary search, and $N=1$ particles represents sequential, non-probabilistic heuristic methods, the paradigm underlying G-Sim, SGA, D3, and others. Crucially, $N=1$ differs from ModelSMC only in removing the SMC framework, making it a well-defined representation of the non-probabilistic set of methods. FunSearch+ could not be used for the kidney task because it is Python-specific and does not support the R-based simulator, as noted in the main text. We consider HH the primary benchmark for cross-method conclusions. We argue the present baselines are sufficient to evaluate ModelSMC performs in line with existing point estimate model discovery methods, while additionally providing a probabilistic interpretation of posteriors over model programs.
>
> **Q3. Posterior conclusion stability.**
> To assess whether model posterior conclusions are reproducible, we conducted an analysis on the Hodgkin-Huxley model (Fig. 3, 4). Specifically, we used an LLM to extract a taxonomy of ion-channel model variants from the discovered simulators, then classified all 1,285 particles (across 10 seeds) into subtypes (e.g. I_M, I_A, I_NaP families). Comparing per-seed weights in each channel class across 10 independent seeds shows that the ranking of channel-type families is consistent: I_M variants systematically achieve better weights than I_A or I_NaP variants in 7/10 seeds (Fig. 5).
>
> We hope these clarifications address the reviewer's concerns and are happy to provide further details on any of the points raised. We would be grateful if the reviewer considers revising their score.

---

> > ### Author Rebuttal · Reviewer_4Td6 · 2026-04-03
> >
> > The rebuttal addressed most of my concerns, thanks.

---

### Official Review · Reviewer_ZVwo · 2026-03-13

**Soundness:** 3
**Presentation:** 3
**Significance:** 3
**Originality:** 3
**Overall Recommendation:** 5
**Confidence:** 4

**Summary:**

The paper proposes to perform model discovery by leveraging a particle filter in the space of models, where the dynamics are generated by an LLM rewriting the code for a simulator and where the likelihood is computed based on Neural Likelihood Estimation (NLE). Since the problem is hierarchical in nature, the likelihood for a model $m$ is a marginal likelihood which involves marginalisation over parameters $\theta$ and contexts $c$. The method is then compared against FunSearch on three problems, one simulated and two real problems.

**Compliance With Llm Reviewing Policy:**

Affirmed.

**Final Justification:**

The rebuttal addressed my main concerns.

**Key Questions For Authors:**

1. Which part of Theorem 3.1 is novel?
2. In (4), the marginal likelihood is obtained by integrating against a generic distribution $p(\theta)$ which does not seem to be explicitly defined. How is it defined?
3. How would the performance change when varying the amount of information provided in the context $c$? Would the performance be close to FunSearch with less information?

**Limitations:**

Yes

**Strengths And Weaknesses:**

# Strengths

1. The particular combination of techniques used seems to be novel, although individual elements are quite well understood.
2. If all approximations and the LLM-based dynamics are to be trusted, then the obtained posterior over model should have good statistical properties.
3. The method is quite general

# Weaknesses

1. Although there is some theory, I could not see what part of the proposed theorem is different from the usual setting for sequential Monte Carlo. If this is indeed a standard theorem then this should be made much clearer in the text. The proof in the appendix mentions that the standard argument is "adapted to the ModelSMC setting", but it is unclear what needs to be adapted.
2. Holt et al. (2024) talks about simulation based inference, so the statement that it relies on "heuristic system designs" seems partially misleading
3. The paper justifies not comparing against "black-box predictive models, such as transformers, because
they lack mechanistic interpretability." However it would insightful to understand the cost of mechanistic interpretability in terms of performance. Such comparison is crucial for practitioners to understand the trade off between performance and interpretability.
4. The performance assessment, in its current form, does not clearly indicate which component of the proposed algorithm brings the most performance. An ablation study would be insightful.

---

> ### Author Rebuttal · Authors · 2026-03-31
>
> We appreciate that the reviewer recognizes that the method is “quite general” and yields posteriors with “good statistical properties”. We address the raised concerns below (W1-4) and answer questions (Q1-3). We refer to new tables and figures found in this [link](https://tinyurl.com/modelsmc).
>
> **W1/Q1. Novelty of Theorem 3.1.**
> The theorem applies a standard SMC consistency result, intentionally so, as stated in Section 3.4. The novelty lies in what it is instantiated over, i.e., a space of executable model programs where particles are simulator implementations and the proposal is an LLM. Two adaptations are needed:
> - State space. Classical SMC operates over continuous finite-dimensional states. In our work, Assumptions on support coverage (i) and bounded importance weights (ii) must be reinterpreted for a discrete, combinatorially large program space. E.g. support coverage becomes a statement about LLM expressivity over model programs.
> - Proposal. The proposal kernel $q(m’|m)$ is induced by an LLM and is implicit, so there is no closed-form density. The path-space argument (Appendix C, Step 7) is therefore applied without requiring a tractable proposal density.
> We acknowledge that the current explanation in Section 3.4 is too brief and will clarify the necessary adaptations from standard SMC theory.
>
> **W2. Clarification on Holt et al. comparison.**
> Holt et al. perform simulation-based inference (SBI) for parameter inference conditional on a model, but the model discovery itself is not part of the Bayesian inference formulation. In contrast, ModelSMC performs joint Bayesian inference over both model programs and parameters. Our use of “heuristic” therefore refers specifically to model-level weighting criteria (e.g., Wasserstein distance, MMD) used in prior model discovery methods, which are not part of the Bayesian inference formulation. ModelSMC’s importance weights, instead, are grounded in a probabilistic objective (Theorem 3.1). We will clarify this in the revision.
>
> **W3. Black-box comparison.**
> Black-box models provide a useful reference point, but their performance should be interpreted as a lower bound. By construction, they can exploit arbitrary statistical patterns without being constrained to physically plausible dynamics. We trained a set of GRUs on the Hodkin-Huxley task and evaluated on-step-ahead prediction MSE. The GRUs achieve slightly lower MSE than the models discovered by ModelSMC (Tab. 1, Fig. 1). Crucially, this advantage does not carry to longer-horizon rollouts, where the GRUs fail to produce stable voltage traces (Fig. 2), while the discovered mechanistic models remain physically consistent.
>
> More broadly, this comparison is inherently limited because the two approaches are designed for different goals. Black-box models optimize predictive accuracy and ModelSMC produces a posterior over interpretable, executable simulators that can yield mechanistic insights. We will add this discussion to the manuscript
>
> **W4/Q3. Ablation study.**
> We agree that understanding individual design choices is valuable. Though we note that ModelSMC’s primary contribution is a *probabilistic formulation* of model discovery. This enables scientifically relevant posteriors over model programs rather than point estimates maximising a held-out score. As a concrete illustration of what this posterior enables, Fig. 3 shows an analysis of the posterior to rule in or our certain modelling hypotheses. This conclusion is only possible because of ModelSMCs posterior over programs.
> We conducted the following ablations on the HH tasks (3 seeds each): N=1 particles (no ModelSMC framework), MSE vs marginal likelihood weighting, LLM backends, simulation budgets, feedback variants, minimal prompts.
> Overall, our configuration for ModelSMC provides the strongest results (Tab. 2). The N=1 baseline is most directly relevant. Removing the population structure reduces overall performance and eliminates the sample-based posterior analysis. This confirms that maintaining a particle population meaningfully contributes to model space exploration. Crucially, any performance improvements, e.g. stronger LLM backends or richer feedback, can be incorporated into ModelSMC while preserving its probabilistic posterior interpretation. We will include the ablation study in the final manuscript.
>
> **Q2. Definition of $p(\theta)$ in (4).**
> $p(\theta)$ is the prior distribution of the parameters of the discovered models. Since parameters vary across models, this prior is model- and task-specific. Priors use domain knowledge by the scientists, default to uniform over physically plausible bounds, or informative priors from literature. The concrete definitions for all three tasks (SIR, kidney, Hodekin-Huxley) are given in Appendices F.2-F.4. We will clarify in the main text with a short guide on prior choice.
>
> We hope these clarifications and experiments address the concerns. We welcome further questions and would appreciate raising the score.

---

> > ### Author Rebuttal · Reviewer_ZVwo · 2026-04-01
> >
> > The rebuttal addresses my main concerns, thank you.

---

### Decision · Program_Chairs · 2026-04-30

**Decision:**

Accept (regular)

**Comment:**

This paper presents a probabilistic framework for automated scientific model discovery using LLMs.  Approximate Bayesian inference over mechanistic models is conducted via a variation of Sequential Monte Carlo where candidate models are proposed by an LLM and weighted using Neural Likelihood Estimation. The paper received favourable reviews, and the discussion was instrumental in clarifying several points, notably touching upon the LMM model proposal mechanisms and to ablation studies. While I concur with 4Td6 that the novelty is more in synthesis and formalization than in a wholly new algorithmic mechanism, I think that the work will make a valuable contribution to ICML 2026 and I am looking forward to reading the revised version in the proceedings.